# Modelling rock glacier ice content based on InSAR-derived velocity, Khumbu and Lhotse Valleys, Nepal

Yan Hu[1,2,3], Stephan Harrison[2], Lin Liu[1,3], Joanne Laura Wood[2]

[1]Earth System Science Programme, Faculty of Science, The Chinese University of Hong Kong, Hong Kong SAR, China
[2]College of Life and Environmental Sciences, University of Exeter, Penryn, Cornwall, TR10 9EZ, UK
[3]Institute of Environment, Energy and Sustainability, The Chinese University of Hong Kong, The Chinese University of Hong Kong, Hong Kong SAR, China

*Correspondence to*: Yan Hu (huyan@link.cuhk.edu.hk)

**Abstract.** Rock glaciers contain significant amounts of ground ice and serve as potential freshwater reservoirs as mountain glaciers melt in response to climate warming. However, current knowledge about ice content in rock glaciers has been acquired mainly from in situ investigations in limited study areas, which hinders a comprehensive understanding of ice storage in rock glaciers situated in remote mountains over local to regional scales. In this study, we develop an empirical rheological model to infer ice content of rock glaciers using readily available input data, including rock glacier planar shape, surface slope angle, active layer thickness, and surface velocity. The model is calibrated and validated using observational data from the Chilean Andes and the Swiss Alps. We apply the model to five rock glaciers in Khumbu and Lhotse Valleys, north-eastern Nepal. The velocity constraints applied to the model are derived from Interferometric Synthetic Aperture Radar (InSAR) measurements. The inferred volumetric ice fraction in Khumbu and Lhotse Valleys ranges from 70±8% to 74±8%; and the water volume equivalents lie between 1.4±0.2 to 5.9±0.6 million m$^3$ for the coherently moving parts of individual landforms. Based on previous inventories and extrapolating our findings to the entire Nepalese Himalaya, the total amount of water stored in rock glaciers would be in the magnitude of 10 billion m$^3$, equivalent to a ratio of 1:17 between rock glacier and glacier reservoirs. Due to the accessibility of model inputs, our approach is easily applicable to permafrost regions where observational data are lacking, and is thus valuable for estimating the water storage potential of rock glaciers in remote areas.

## 1 Introduction

Rock glaciers are valley-floor and valley-side landforms that occur in the periglacial realm. Intact rock glaciers contain ground ice and are common in the cold mountain regions (Ballantyne, 2018; Berthling, 2011; Brenning, 2005a). Recent research has suggested that they represent important hydrological reservoirs in areas where glaciers are undergoing recession in the face of climate change (Azócar and Brenning, 2010; Jones et al., 2018a; Munroe, 2018; Rangecroft et al., 2014). Corte (1976) first proposed the potential hydrological value of rock glaciers, yet research on the role of rock glaciers in maintaining hydrological stores in mountain catchments remains limited.

In regions such as the Himalaya, recent research has argued that rock glaciers might represent the end member of an evolutionary process where some glaciers transition to debris-covered glaciers, a proportion of which will then undergo further transition to rock glaciers (Jones et al., 2019a; Knight et al., 2019). The paraglacial response of high mountain slopes would contribute to this process, as glaciers undergo downwasting, which triggers rock slope failures such as rockslides and rock

avalanches and increases the flux of rock debris to glacier surfaces (Jarman and Harrison, 2019). Depending on the debris cover thickness, this would be expected to limit ice melting and increase the resilience of glaciers to climate change (e.g., Reznichenko et al., 2010).

Jones et al. (2021) were the first to show that around 25,000 rock glaciers exist in the Himalayas, covering 3747 km² and containing $51.80 \pm 10.36$ km³ of water volume equivalent. The ratio between rock glacier ice content and that in glaciers in

the region was 1:25, ranging from 1:42 to 1:17 in the Eastern and Central Himalaya and falling to 1:9 in Nepal. Importantly, we expect these existing ratios to reduce significantly as glaciers melt and/or undergo transitions to rock glaciers. Few studies have investigated the hydrological contribution of rock glaciers to surface runoffs at annual or seasonal timescale (e.g., Geiger et al., 2014; Harrington et al., 2018; Krainer and Mostler, 2002; Winkler et al., 2016), and little evidence has shown that rock glacier discharge is a prominent water source at present due to the insulation effect produced by their blocky surfaces (Duguay

et al., 2015; Jones et al., 2019b; Pruessner et al., 2021). Yet, on multi-annual to centennial and millennial timescales, we expect rock glaciers with high ice content to serve as water reservoirs long after glaciers have melted.

To date, we have little quantitative information concerning the ice content of rock glaciers, which hinders our understanding of the potential future hydrological role of rock glaciers. Currently, estimates of ice content in rock glaciers have focused on empirical information from drilling cores and boreholes (Hausmann et al., 2007; Monnier and Kinnard, 2013, 2015a, b; Fukui

et al., 2007; Arenson et al., 2002; Berthling et al., 2000; Croce and Milana, 2002; Florentine et al., 2014; Fukui et al., 2008; Guglielmin et al., 2004; Guglielmin et al., 2018; Haeberli et al., 1998; Haeberli et al., 1999; Krainer et al., 2015; Leopold et al., 2011; Steig et al., 1998), and from geophysical surveys (e.g., for reviews see: Hauck, 2013; Kneisel et al., 2008; Scott et al., 1990). However, these approaches to estimate the likely ice content are costly, time-consuming, and labour-intensive to apply to rock glaciers at high altitudes and in remote mountains. It is therefore desirable to develop alternative approaches to

understanding the likely ice content of rock glaciers, especially for regional scale estimates.

Ice content is one factor controlling the movement of rock glaciers by influencing the driving force and the rheological properties of materials which constitute the permafrost core (Arenson and Springman, 2005a; Cicoira et al., 2020), thus it is feasible to infer ice content using rheological modelling and observed kinematic data. Here we adapt an empirical model by integrating rheological properties of rock glaciers derived from laboratory experiments (Arenson and Springman, 2005a), and

parameterise the rheological model based on the structure and composition data of Las Liebres rock glacier (Monnier and Kinnard, 2015b; Monnier and Kinnard, 2016). We then apply the model to simulate surface velocities of three rock glaciers with known ice content in the Swiss Alps and evaluate the modelling results to determine a suitable parameterisation scheme. Finally, we apply the calibrated model for five rock glaciers in the study area of north-eastern Nepal and model their ice contents based on remote-sensing-derived downslope velocities as constraints. The proposed approach aims to estimate the

current amount of ground ice stored in Nepalese rock glaciers and to assess the hydrological importance of rock glaciers as freshwater reservoirs in the long term.

## 2 Study area

Our study area comprises the Khumbu and Lhotse valleys in north-eastern Nepal (Fig. 1a). Among the highest in the world, the Khumbu and Lhotse glaciers draining Everest have well defined debris-covered snouts. The tributary valleys contain a variety of rock glaciers and composite landforms where glaciers are transitioning to rock glaciers (Jones et al., 2019; Knight et al., 2019). There are five rock glaciers in the study area, namely Kala-Patthar, Kongma, Lingten, Nuptse, and Tobuche (Fig. 1b). The five rock glaciers examined in this study are situated at 4900–5090 m a.s.l., near the lower limit of permafrost in the region. Previous seismic refraction surveys conducted on active rock glaciers indicate that the lower limit of permafrost occurrence in this region to be ~5000–5300 m a.s.l. (Jakob, 1992), which is consistent with an earlier estimate of 4900 m a.s.l. based on ground temperature measurements (Fujii and Higuchi, 1976).

Meteorological data provided by the Pyramid Observatory Laboratory near Lobuche village on the western side of the Khumbu Glacier (5050 m a.s.l.) reveal that the dominating climate of this area is the South Asian Summer Monsoon. For the period of 1994–2013, recorded accumulated annual precipitation was 449 mm yr$^{-1}$, with 90% of the precipitation concentrated during June–September (Salerno et al., 2015). The mean annual air temperature is –2.4 °C (Salerno et al., 2015).

Measurements of ground temperature in the study area are scarce in general. However, we infer that these rock glaciers develop in a warm permafrost environment for the following reasons: (1) the landforms are located near or below the altitudinal limit of permafrost distribution in Nepal (Fujii and Higuchi, 1976; Jakob, 1992), indicating that the local environment is at the critical limit of permafrost occurrence; (2) based on empirical relationships between mean annual ground temperature (MAGT), mean annual air temperature, latitude, and altitude, the estimated MAGT is >0.5°C, which suggests that permafrost in this area is in a warm and unstable state (Nan et al., 2002; Zhao and Sheng, 2015).

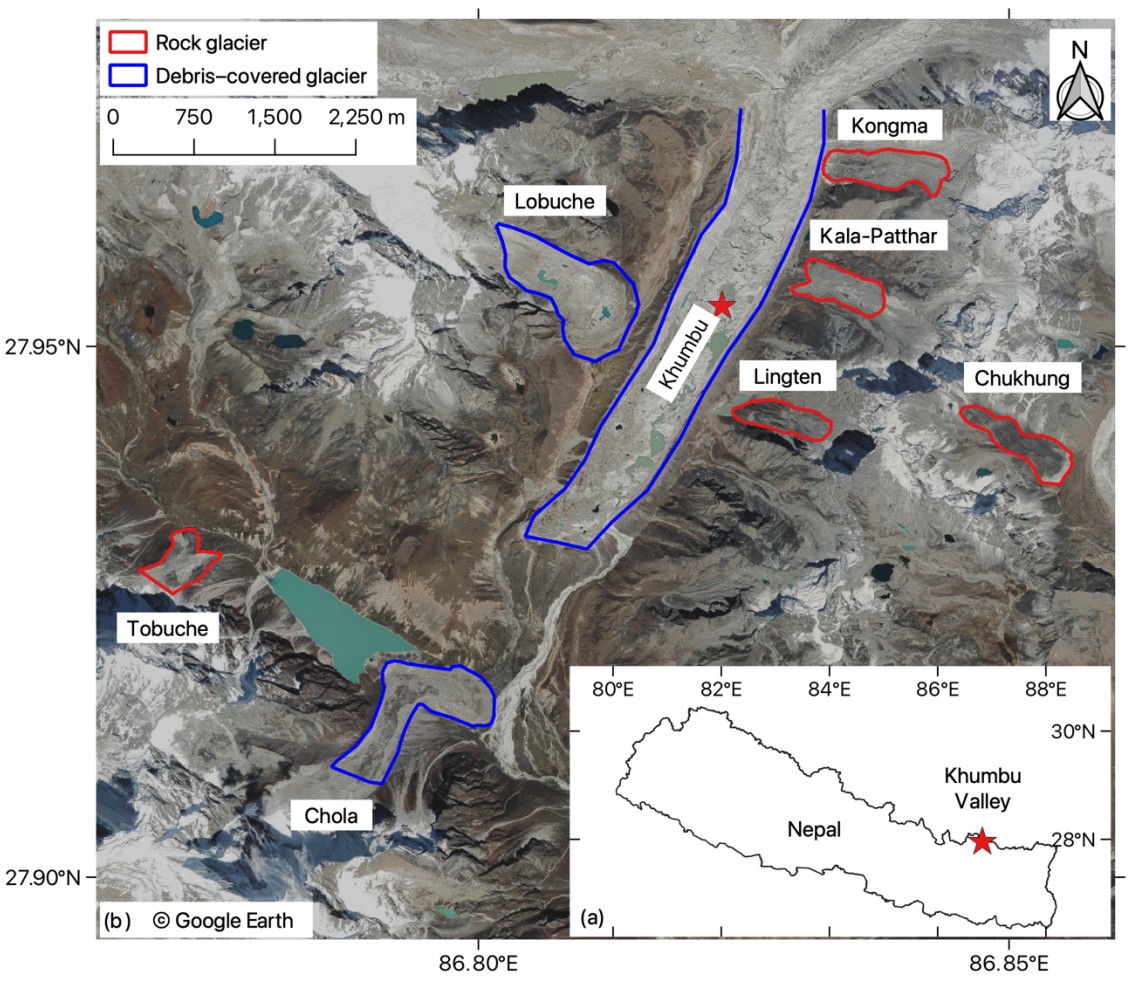

**Figure 1: (a)** Location of the study site; **(b)** Google Earth images (taken in 2019) showing the spatial distribution of the active ice–debris landforms, including rock glaciers (RG) in red outlines and debris-covered glaciers (DCG) in blue boundaries. The RGs are delineated by Jones et al. (2018) and the DCGs by the authors based on Google Earth images.

## 3 Methods

The main workflow of our method is illustrated in Fig.2. In this section, we first introduce the model design and basic assumptions we adopted (Sect. 3.1). Then we present the following development steps in sequence: model calibration (Sect. 3.2), validation (Sect. 3.3), and sensitivity test (Sect. 3.4). Finally, we describe the model application based on InSAR (Sect. 3.5) and the method to extrapolate the results at the regional scale (Sect. 3.6).

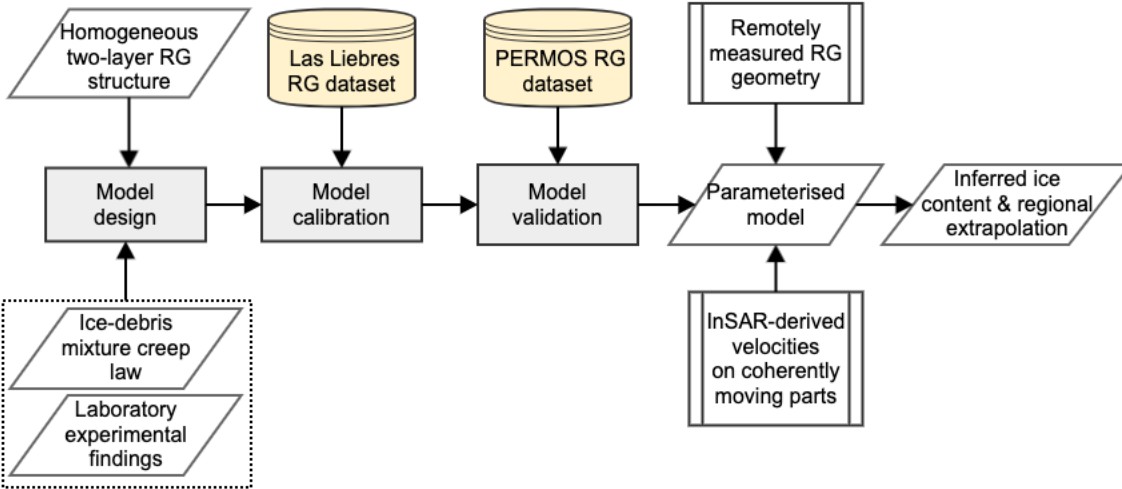

**Figure 2: Diagram of the workflow conducted in this study to develop and apply a modelling approach for inferring ice content of rock glaciers (RG).**

### 3.1 Model design and assumptions

Active rock glaciers are viscous flow features embodying ice-rich permafrost (Ballantyne, 2018; Berthling, 2011; Haeberli, 2000). Many previous modelling studies depict the deformation mechanism of rock glaciers based on Glen's flow law (e.g., Arenson and Springman, 2005a; Cicoira et al., 2020; Whalley and Azizi, 1994), which essentially relates strain rate ($\dot{\varepsilon}$) with effective shear stress ($\tau$) and describes the rheology of ice flow (Glen, 1955):

$$\dot{\varepsilon} = A\tau^n , \tag{1}$$

where $A$ and $n$ are creep parameters reflecting variations in environmental conditions (mainly temperature and pressure), material properties (such as composition, structure, and texture), and operating creep mechanisms (e.g., diffusion and dislocation).

In this study, we primarily adopted a creep model of ice–debris mixture, proposed by Moore (2014), based on Glen's flow law:

$$\dot{\varepsilon} = EA[(\tau - \tau_{th})\Gamma]^n , \tag{2}$$

where $E$ is a strain enhancement factor; $\Gamma$ is a parameter reflecting the strength of the ice–debris mixture, associated with the volumetric debris content ($\theta_d$). We assumed the rock glacier has an ice-rich permafrost core. When $\theta_d$ is less than a critical volumetric debris content ($\theta_{dc}$), ice creep dominates the behaviour of the mixture , and the value of $\Gamma$ equals one. Theoretically, $\theta_{dc}$ is around 0.52 (Moore, 2014). $\tau$ is the driving stress and $\tau_{th}$ is a threshold stress imparted by the frictional strength between debris particles, also depending upon the volumetric debris content ($\theta_d$).

Assuming that $\tau_{th} \ll \tau$, $\theta_d < \theta_{dc}$, and $\Gamma = 1$, Eq. 2 can be reduced to the following form (Monnier and Kinnard, 2016):

$$\dot{\varepsilon} = \left(\frac{\tau}{B}\right)^n , \tag{3}$$

where $B$ is the effective viscosity and is equal to $\left(\frac{1}{EA}\right)^{-\frac{1}{n}}$. We introduced the effective viscosity ($B$) to absorb the intricate effects of strain enhancement factor ($E$), threshold stress ($\tau_{th}$), and most importantly, the creep parameter ($A$), which is primarily affected by ground temperatures (Mellor and Testa, 1969). Previous research (e.g., Arenson and Springman, 2005a; Azizi and Whalley, 1996; Kääb et al., 2007; Ladanyi, 2003) considered this factor by implementing a heat diffusion model (proposed by Carslaw and Jaeger, 1959). In this study, we used a constant effective viscosity ($B$) to describe the deformation behaviour of rock glaciers in a warm permafrost environment ($> -3°C$). The empirical formula was developed based on existing observational data and laboratory findings. This warm ground condition is likely to be realistic in our study area (Sect. 2).

We assume a homogeneous structure and consider each rock glacier as a slab with uniform width and thickness and a semi-elliptical cross-section, resting on a bed of constant slope, which is a common setup in glaciology (Cuffey and Paterson, 2010). It consists of two layers: an active layer and a permafrost core. The active layer is a mixture of debris and air, and the permafrost core consists of ice, water, debris and air. Both layers are assumed as homogeneous. Movement of rock glaciers is caused by the steady creep of the permafrost core in the plane parallel to the bed slope. The active layer moves passively along with the inner core, which has been validated by observations (Arenson et al., 2002; Haeberli, 2000).

Here we neglected the presence of shear horizon where deformation is enhanced and ground ice content is high, as discovered from borehole investigations (Arenson et al., 2002; Buchli et al., 2018; Haeberli et al., 1998). Field observations and numerical modelling suggest that unfrozen water within the shear horizon plays an important role in controlling the seasonal variations in rock glacier creep (Buchli et al., 2018; Cicoira et al., 2019b; Kenner et al., 2019). However, the short-term rock glacier kinematic patterns are irrelevant to this study focusing on modelling the relationship between ice content and multi-annual average movement velocity in our study.

From Eq. 3 and the structure and geometry illustrated in Fig. 3, we have:

$$\frac{du}{dz} = 2\left(\frac{\tau}{B}\right)^n , \tag{4}$$

where $\frac{du}{dz}$ is the velocity derivative relative to the depth $z$ in the permafrost core.

At a given depth $z$, the driving stress $\tau$ is imparted, taking into account the loading of the above material and the effect of frictional drag occurring between the lateral margins and surrounding bedrocks, which is represented by a shape factor $S_f$ (Cuffey and Paterson, 2010):

$$\tau(z) = S_f \sin\alpha(\rho_{al}gh_{al} + \rho_{core}gz) , \tag{5}$$

where $\alpha$ is the slope angle; $g$ is the gravitational acceleration; $\rho_{al}$ and $\rho_{core}$ are the densities of the active layer and the permafrost core, respectively; $h_{al}$ is the active layer thickness.

The shape factor is expressed as (Oerlemans, 2001):

$$S_f = \frac{\pi}{2}\arctan\left(\frac{W}{2T}\right) , \tag{6}$$

where $W$ and $T$ are the width and thickness of the rock glacier, respectively.

The integration of the velocity profile (Eq. 4 and 5) is expressed as:

$$\int_0^z du = -2 \left(\frac{S_{fg} \sin \alpha}{B}\right)^n \int_0^z (\rho_{al} h_{al} + \rho_{core} z)^n dz, \tag{7}$$

$\quad u(z) = u_s - \frac{2(\rho_{al} h_{al} + \rho_{core} z)^{n+1}}{\rho_{core}(n+1)} \left(\frac{S_{fg} \sin \alpha}{B}\right)^n, \tag{8}$

where $u_s$ is the surface velocity as illustrated in Fig. 3. When $z$ is set as the thickness of the ice core ($h_{core}$) and basal sliding is assumed to be absent, $u_s$ is then expressed as:

$$u_s = \frac{2(\rho_{al} h_{al} + \rho_{core} h_{core})^{n+1}}{\rho_{core}(n+1)} \left(\frac{S_{fg} \sin \alpha}{B}\right)^n, \tag{9}$$

The densities of the active layer ($\rho_{al}$) and the permafrost core ($\rho_{core}$) are given as:

$\quad \rho_{al} = \theta_{d,al} \rho_d + \theta_{a,al} \rho_a, \tag{10}$

$\rho_{core} = \theta_{d,core} \rho_d + \theta_{a,core} \rho_a + \theta_{i,core} \rho_i + \theta_{w,core} \rho_w, \tag{11}$

where $\theta_{d,al}$ and $\theta_{a,al}$ are the volumetric contents of debris and air in the active layer, respectively. The volumetric contents of the components in the inner core, namely debris, air, ice, and water, are expressed as $\theta_{d,core}$, $\theta_{a,core}$, $\theta_{i,core}$, and $\theta_{w,core}$, respectively. $\rho_d$, $\rho_a$, $\rho_i$, and $\rho_w$ are the densities of debris, air, ice, and water, respectively.

We fixed the air content in the permafrost core as 7.5%, which is a mean value of the air fraction in ice-rich permafrost samples (Arenson and Springman, 2005b). At near 0 °C, the volumetric content of water ($\theta_{w,core}$) displays a positive correlation with the debris fraction ($\theta_{d,core}$) (Monnier and Kinnard, 2016). Thus, we determined the $\theta_{d,core} - \theta_{w,core}$ relationship based on the data published in Monnier and Kinnard (2015b) and assumed the constitution of the selected rock glaciers for model validation and application followed the same linear relationship (Fig. S1). The debris density ($\rho_d$) was given as 2450 kg/m³ (Monnier

and Kinnard, 2016). The density of air ($\rho_a$) is determined by the elevation of each rock glacier: for instance, rock glaciers situated between 2500 m and 3500 m have an air density of 1.007 kg/m³. The ice density ($\rho_i$) is 916 kg/m³ and the water density ($\rho_w$) is 1000 kg/m³.

For the flow law exponent ($n$), we first used an empirical average value as assumed in modelling pure ice creep:

$n = 3, \tag{12}$

We also adopted a linear relationship between $n$ and the volumetric ice content ($\theta_{i,core}$) based on laboratory experiments undertaken on borehole samples from two rock glaciers (Arenson and Springman, 2005a):

$n = 3\theta_{i,core}, \tag{13}$

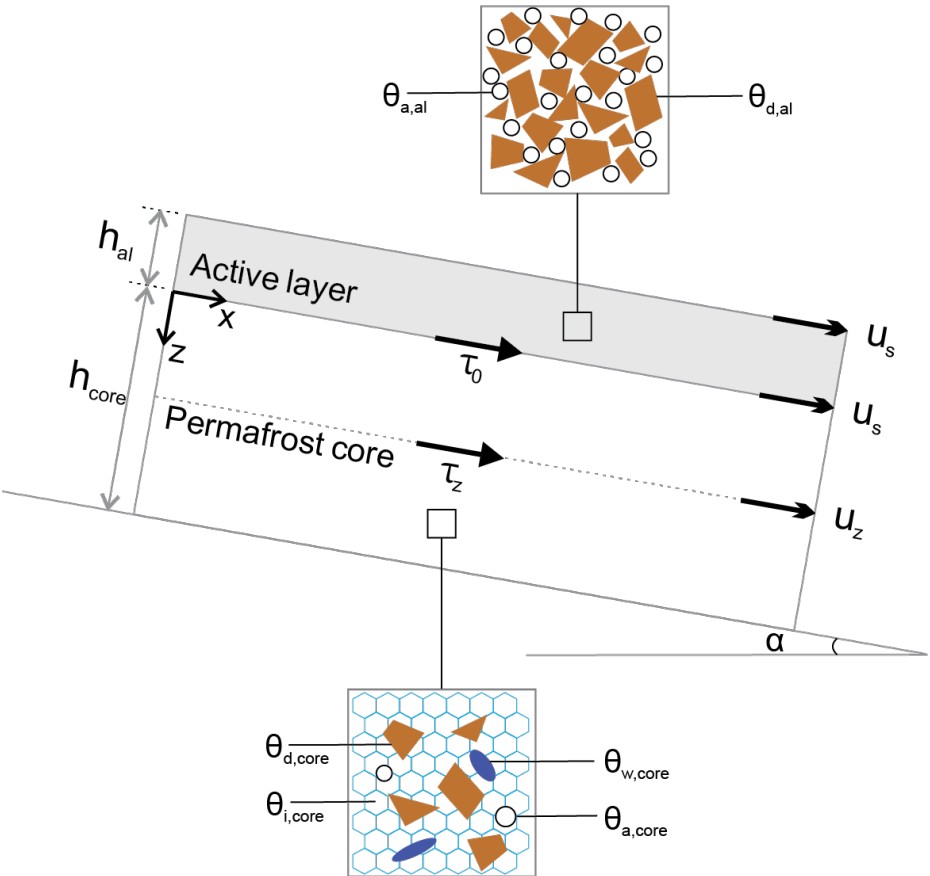

Figure 3: Schematic geometry, structure, stress status, and composition of rock glaciers. The rock glacier consists of a permafrost core underlying the active layer. Parameters involved in the model include surface slope ($\alpha$), active layer thickness ($h_{al}$), thickness of permafrost core ($h_{core}$), driving stress at the base of the active layer ($\tau_0$), driving stress at depth z ($\tau_z$), surface velocity ($u_s$), velocity at depth z ($u_z$). $\theta_{d,al}$ and $\theta_{a,al}$ refer to the debris fraction and air fraction of the active layer. $\theta_{d,core}$, $\theta_{i,core}$, $\theta_{w,core}$, and $\theta_{a,core}$ are the fractions of debris, ice, water, and air in the permafrost core, respectively.

## 3.2 Model calibration

Combining Eq. 9–11 with Eq. 12 or 13, we formulated several expressions depicting the relationship between the surface velocity and properties of rock glaciers, including their composition, structure, and geometry. We then calibrated the model by using observational data of Las Liebres rock glacier in Central Chilean Andes (Monnier and Kinnard, 2015b) to determine the curve of best fit between the effective viscosity ($B$) and the volumetric ice content ($\theta_{i,core}$). The calibration dataset includes information of structure ($h_{core}$ and $h_{al}$), geometry ($\alpha$ and $S_f$), and composition ($\theta_{d,core}$, $\theta_{a,core}$, $\theta_{i,core}$, and $\theta_{w,core}$), all of which were derived from Ground Penetrating Radar (GPR) measurements. Surface velocities ($u_s$) were provided by a Differential Global Positioning System (DGPS) along the central creep line at 14 locations on Las Liebres rock glacier (Monnier and Kinnard, 2015b & 2016).

First, we adopted the exponential $B-\theta_{i,core}$ relationship estimated by Monnier & Kinnard (2016) with the same dataset and a constant creep parameter $n$ (Eq. 12). Then by integrating the relationship between $n$ and ice content (Eq. 13), we applied both a 2$^{nd}$-degree polynomial regression model and an exponential regression model to determine the $B-\theta_{i,core}$ relationship. The polynomial regression model is used to capture the subtle increase in effective viscosity when the ice fraction increases. This trend was also shown by Arenson and Springman (2005a) who suggested a parabolic relationship between the minimum axial creep strain rate and the volumetric ice content.

## 3.3 Model validation

The calibrated parameterisation schemes were validated using observational data from three rock glaciers in the Swiss Alps, namely Murtèl-Corvatsch, Muragl, and Schafberg (Cicoira et al., 2019a; Arenson et al., 2002; Hoelzle et al., 1998). We simulated the surface velocity ($u_s$) of each rock glacier by varying volumetric ice content ($\theta_{i,core}$) of the permafrost core. Then we compared the modelled velocity with the measured velocity from Terrestrial Geodetic Surveys (PERMOS, 2019). We then referred to the previously estimated ice content of the selected rock glaciers to validate our predicted results.

To derive the input parameters, we first outlined the boundaries of the three rock glaciers from Google Earth images (September of 2018), from which their shapes and areal extents can be extracted using Geographic Information System tools. As Muragl and Schafberg rock glaciers consist of multiple or overlapping lobes, we focus on a single active lobe of each rock glacier where the borehole is present and composition data are available. The three rock glaciers for validation have a tongue-shaped typology. An empirical relationship established by Brenning (2005b) was then applied to calculate the rock glacier thickness ($T$) from its areal extent ($A_{rg}$):

$$T = 50 A_{rg}^{0.2} , \tag{14}$$

where the area ($A_{rg}$) is in km$^2$. We assigned a relative uncertainty of 40% to the area parameter and considered the propagated error to the final modelling result. The width of each glacier was quantified as the width of its minimum envelop rectangle. We took the mean value of the active layer thickness obtained from borehole measurements in the PERMOS network as the input parameter $h_{al}$ for each rock glacier. The surface slope ($\alpha$) was calculated based on the SRTM DEM with a spatial resolution of ~30 m. Table 1 lists the values of the above parameters. The permafrost core thickness ($h_{core}$) can be obtained by subtracting $h_{al}$ from the total thickness $T$ calculated using Eq. 14.

We assumed the volumetric ice content ($\theta_{i,core}$) of the permafrost core to be between 40% and 100%, considering the prerequisites of the modified ice–debris mixture flow law (Eq. 3) that the debris fraction ($\theta_{d,core}$) should be less than the threshold ($\theta_{dc}$) (Sect. 3.1). We varied the ice content ($\theta_{i,core}$) by 1% in each step to model the corresponding surface velocities ($u_s$).

**Table 1. Summary of the geometric and structural parameters used in the validation.**

| Rock glacier | Area ($\mathbf{A_{rg}}$) (km$^2$) | Width (W) (m) | Active layer thickness ($\mathbf{h_{al}}$) (m) | Surface slope ($\boldsymbol{\alpha}$) (°) |
| --- | --- | --- | --- | --- |

| | | | | |
|---|---|---|---|---|
| Murtèl-Corvatsch | 0.06487 | 29 | 3.0 | 16 |
| Muragl | 0.02666 | 24 | 4.5 | 12 |
| Schafberg | 0.02715 | 24 | 4.8 | 16 |

## 3.4 Sensitivity analysis

To explore how uncertainties of the input parameters contribute to the final output of the developed approach, we tested the response of the model to varying input parameters by performing a series of synthetic sensitivity experiments. For these experiments, we simulated surface velocities of the rock glacier with variable ice fractions and inferred the current ice content from the velocity constraint. A reference scenario is set up with the parameters of Murtèl-Corvatsch rock glacier and labelled as Sc-1.0. We designed eight scenarios extending from Sc-1.0, naming each scenario after a multiplication factor which indicates the ratio between the applied parameter and the reference scenario; For two parameters, namely debris density ($\rho_d$) and debris fraction in the active layer ($\theta_{d,al}$), we applied a value range according to the known natural variability based on observations ($\rho_d$: 1450–3450 kg/m$^3$; $\theta_{d,al}$: 13–93%). A full list of the parameters used in the sensitivity test is presented in Table S1 in the supplementary materials. We performed the sensitivity experiments by varying one parameter at a time while keeping the other variables constant.

## 3.5 Model application

The validated model with the optimal parameterisation scheme was applied to estimate ice content of rock glaciers with remotely sensed input data. In this subsection, we present our method of measuring surface velocities of rock glaciers with InSAR for constraining the model (Sect. 3.5.1) and deriving geometric and structural parameters from remote sensing products (Sect. 3.5.2).

### 3.5.1 Deriving surface velocity constraints with Differential InSAR

InSAR has been widely applied to quantifying surface velocities of rock glaciers (e.g., Bertone et al. 2022; Reinosch et al. 2021; Rouyet et al. 2019; Zhang et al. 2021). In this study, we adopted the conventional two-pass Differential InSAR method to derive the surface velocities by assuming rock glaciers creep along the slope direction (Brencher et al., 2021; Hu et al., 2021; Liu et al., 2013; Wang et al., 2017). We identified the coherently moving part of the rock glacier and determined the surface velocity for constraining the model.

Step 1: Interferometric processing

Nineteen L-band ALOS PALSAR images and twenty-one ALOS-2 PALSAR-2 images acquired during 2006–2010 and 2015–2020, respectively, were used to form more than fifty interferograms to measure the surface displacements of the landforms in the study area (Table 2). We selected interferograms to achieve high interferometric coherence by following the criteria such

as: (1) short temporal spans (less than 92 days for ALOS pairs and 70 days for ALOS-2 pairs); (2) short perpendicular baselines (smaller than 800 m for ALOS pairs and 400 m for ALOS-2 pairs). We estimated and removed the topographic phase with the 1-arcsec digital elevation models (DEM) produced by the Shuttle Radar Topography Mission (SRTM) (spatial resolution ~30 m). Multi-looking operation and adaptive Goldstein filter (8×8 pixels) were applied to the interferometric processing, which was implemented by the open-source software ISCE version 2.4.2 (available at https://github.com/isce-framework/isce2). The final georeferenced interferogram has a ground resolution of ~30 m according to the DEM. The interferograms were unwrapped using the SNAPHU software (Chen and Zebker, 2002). We randomly selected three pixels at places supposed to be stable near each ice–debris landform (within 300m) and averaged their phase values to re-reference the unwrapped phases measured within the landforms. By doing so, atmospheric delays can be effectively removed because these lead to long-wavelength artefacts and can be assumed as constant within the range of our study objects (Hanssen, 2001).

Step 2: Calculating downslope velocities from high-quality interferograms

We then derived the surface velocities along the SAR satellite line-of-sight (LOS) direction from the unwrapped interferograms and projected the LOS velocities onto the downslope direction of the landforms. The projection was conducted considering the satellite's flight direction, the local incidence angle, and the landform topographic parameters including the aspect and slope angles (Massonnet and Feigl, 1998; Bechor and Zebker, 2006). We considered the propagation of errors introduced by the InSAR measurements and DEM data which were used to determine the associated topographic parameters (Hu et al., 2021). For each pixel, we found the velocity error is < 10 cm yr$^{-1}$.

To ensure high data quality, we selected the InSAR observations meeting the following criteria as valid results for further analyses: (1) the pixels showing acceptable coherence (>0.3) are kept before velocity statistics; (2) the remaining pixels cover more than 40% of the landform surface; (3) the mean velocity of the landform is larger than 5 cm yr$^{-1}$ (Wang et al., 2017). We set this empirical threshold considering the typical noise level (5 cm yr$^{-1}$) as we observed in most interferograms.

Step 3: Determining the velocities of the coherently moving parts as the model constraint

Field observations have revealed that multiple areas moving differentially can occur on rock glaciers and exhibit complex kinematic patterns (e.g., Buchli et al., 2018), which violates the assumption of a continuously moving body (Sect. 3.1, Fig. 3). Therefore, we aim to identify the coherently moving part of the landform that corresponds with our assumption and is thus suitable for model application.

After the data-refining procedure in Step 2, for each landform, the remaining interferograms constituted a series of observations spanning multiple years. Then we defined and outlined the "coherently moving part" of each landform by considering the time series of downslope velocity of each pixel acquired during the observational periods. If the InSAR-measured velocity is higher than 5 cm yr$^{-1}$ in more than half of the interferograms at a given pixel, it was included in the coherently moving part of the

landform. Otherwise, the pixel cannot be regarded as actively in motion with the coherently moving area but in an inactive or transitional kinematic status.

Then, we analysed the velocity values of all pixels within the coherently moving part of the landform and selected the mean, median, and maximum values for each observation to characterise the surface kinematics of the landforms. The error of the mean velocity can be derived by error propagation of all the pixels taken into account, which is limited to $< 1$cm yr$^{-1}$.

Finally, we take the range of the spatial mean velocities of the coherently moving parts over the observational period as the velocity constraint for modelling ice content. By doing so, the short-term feature of rock glacier kinematics is neglected in our study.

**Table 2. List of ALOS PALSAR and ALOS-2 PALSAR-2 interferograms used in the study.**

| Satellite | Acquisition interval (days) | Period | Path/frame | Orbit direction | No. of interferograms |
|-----------|-----------------------------|--------|------------|-----------------|-----------------------|
| ALOS | 46 | Dec 2007 to Feb 2010 | 507/540 | Ascending | 8 |
| ALOS | 46 | Dec 2007 to Feb 2010 | 507/550 | Ascending | 6 |
| ALOS | 46 | Jun 2007 to Feb 2010 | 508/540 | Ascending | 4 |
| ALOS | 46 | May 2006 to Jul 2006 | 511/540 | Ascending | 1 |
| ALOS-2 | 14 | Mar 2015 | 48/3050 | Descending | 1 |
| ALOS-2 | 14 | Jun 2015 to Feb 2020 | 156/550 | Ascending | 20 |

**3.5.2 Deriving geometric and structural parameters from remote sensing products**

Area, width, and slope angle are quantified using the same method as described in Sect. 3.3. Active layer thickness was determined as the mean value over the extent of each rock glacier, based on the 2006–2017 estimate from the European Space Agency Permafrost Climate Change Initiative Product (ESA CCI) (Obu et al., 2020). The empirical relation for calculating rock glacier thickness used in the validation procedure (Sect. 3.3) was applied here to obtain the thickness parameter. The

surface velocity constraint is the range of InSAR-derived downslope velocity during the observed period; except for Tobuche rock glacier where the abnormal value in 2015 is removed from the range (see Sect. 4.4.1 for details).

**3.6 Regional extrapolation**

We calculated the water equivalents of the five rock glaciers by considering the modelled ice contents and the volumetric extents of the coherently moving parts. Then we used average water equivalents to represent the water storage in a typical rock

glacier in this region. We referred to a published inventory compiled by Jones et al. (2018) that reported 4226 intact rock glaciers over the Nepalese Himalaya. By multiplying the average water storage and the number of landforms, we extrapolated our findings from the Khumbu and Lhotse valleys to estimate the potential water storage across the mountain range. Finally, we compared the estimated water storage in rock glaciers with the glacier reservoir at the regional scale.

## 4 Results

In this section we first present the results of our model development including the calibrated parameterisation schemes (Sect. 4.1), model validation (Sect. 4.2), and model sensitivity (Sect. 4.3). Then we report the modelled ice content in Khumbu and Lhotse valleys (Sect. 4.4). Finally, we show the extrapolated results of the potential water storage in rock glaciers in the Nepalese Himalaya (Sect. 4.5).

### 4.1 Calibrated parameterisation schemes

By applying the different regression models to depict the $B$–$\theta_{i,core}$ relationship (Fig. 4a–c), we obtained three candidate parameterisation schemes expressed as:

Scheme 1: $\quad u_s = \dfrac{2(\rho_{al}h_{al} + \rho_{core}h_{core})^4}{\rho_{core}(n+1)}\left(\dfrac{S_f g \sin\alpha}{35300 e^{2.01\theta_{i,core}}}\right)^3,$ (15)

Scheme 2: $\quad u_s = \dfrac{2(\rho_{al}h_{al} + \rho_{core}h_{core})^{3\theta_{i,core}+1}}{\rho_{core}(3\theta_{i,core}+1)}\left(\dfrac{S_f g \sin\alpha}{7183435\theta_{i,core}{}^2 - 9543596\theta_{i,core} + 3322637}\right)^{3\theta_{i,core}},$ (16)

Scheme 3: $\quad u_s = \dfrac{2(\rho_{al}h_{al} + \rho_{core}h_{core})^{3\theta_{i,core}+1}}{\rho_{core}(3\theta_{i,core}+1)}\left(\dfrac{S_f g \sin\alpha}{5217905 e^{-5.26\theta_{i,core}}}\right)^{3\theta_{i,core}},$ (17)

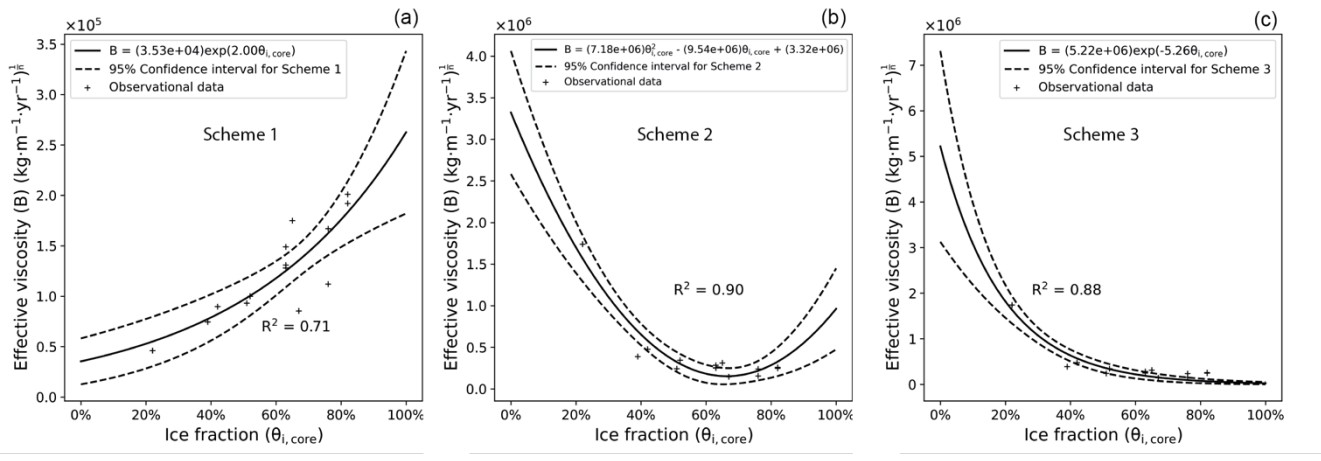


**Figure 4: (a)–(c) Relationships between the ice fraction ($\theta_{i,core}$) and the effective viscosity (B) estimated from the three regression equations and parameterisation schemes (Eq. 15, 16, and 17, respectively). The observational data are derived from the GPR and DGPS measurements in Monnier and Kinnard (2015b & 2016).**

### 4.2 Model validation

We simulated the surface velocities ($u_s$) of the three rock glaciers using Schemes 1–3. Uncertainties from the statistical analysis (dashed lines in Fig. 4) and area delineation (Sect. 3.3) have been considered in the simulation. We used the mean annual surface velocities, calculated from the Terrestrial Ground Survey data (PERMOS, 2019), as the constraint for inferring the ice content.

For each rock glacier, an inferred ice content range is derived based on the velocity constraint and modelled $u_s - \theta_{i,core}$

relationship. The median of the range is selected as the inferred ice content and compared with the reference ice content, i.e., the average value of the estimated ice content based on previous field measurements (Cicoira et al., 2019a; Arenson et al., 2002; Hoelzle et al., 1998).

Comparing the observed and modelled ice content from the three schemes, we see that Scheme 2 is the optimal model for the following two reasons: (1) the reference ice content is within the range inferred from Scheme 2 (Fig. 5, Fig S2 and S3); (2)

Scheme 2 gives the smallest root mean square error (RMSE) (8%) compared with Scheme 1 (9%) and Scheme 3 (12%) (Table 3). We used the RMSE (8%) derived from Scheme 2 to represent the uncertainty of our approach.

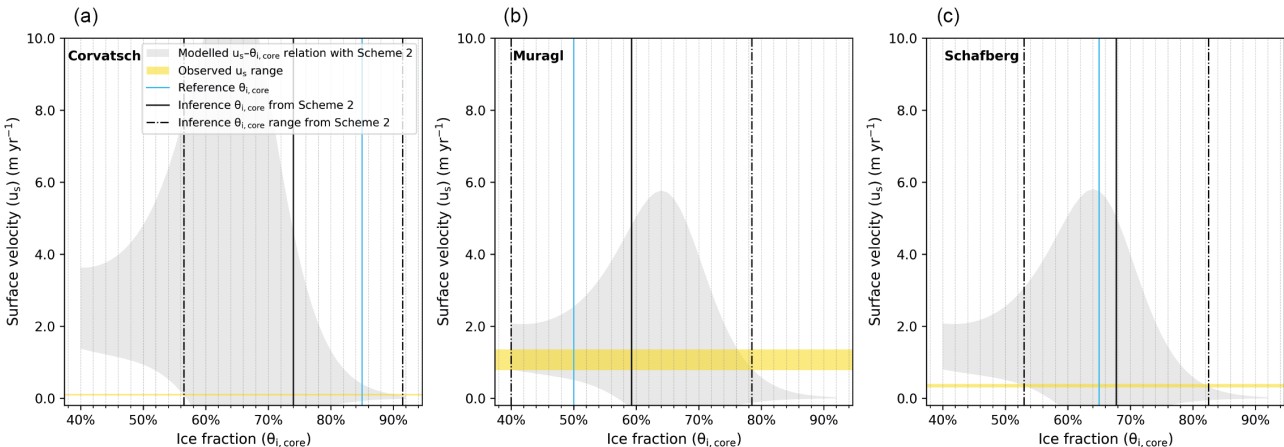

**Figure 5: Modelled relationships (grey shaded areas) between the ice fraction ($\theta_{i,core}$) and the surface velocity ($u_s$) of 95% confidence intervals for the three RGs monitored in the PERMOS network with model parameterisation Scheme 2. The yellow bands show the**
**observed surface velocities, and the blue lines denote the reference ice contents. For each rock glacier, the intersection between the simulated $\theta_{i,core}$- $u_s$ relationship (grey shaded area) and the observed velocity (yellow band) gives the estimated range of ice content, as marked by the dash-dotted black lines. We take the estimated average as the inferred ice content and show the value by the solid black line.**

**Table 3. Summary of the reference and inference ice contents derived from the three model parameterisation schemes. The values**
**in brackets following the inference ice contents give the corresponding bias from the reference ice contents. The last row presents the root mean square error (RMSE) of the schemes.**

| Rock glacier | Reference (%) | Inference and bias | | |
| --- | --- | --- | --- | --- |
| | | Scheme 1 (%) | Scheme 2 (%) | Scheme 3 (%) |
| Murtèl-Corvatsch | 85 | 91 (6) | 74 (–11) | 79 (–6) |
| Muragl | 50 | 56 (6) | 59 (9) | 66 (16) |
| Schafberg | 65 | 79 (14) | 68 (3) | 76 (11) |
| RMSE | – | 9 | 8 | 12 |

## 4.3 Model sensitivity

The results of sensitivity experiments are normalised to the corresponding values of the reference scenario (Scn-1.0, Fig. 6). We observe that the inference result remains stable in response to most varying parameters, with a bias of less than 5%, relative to the reference scenario (Scn-1.0). The model has a higher sensitivity to the surface slope angle. In the extreme scenario (Scn-0.2), the inferred ice content can be altered by 15%. In non-extreme cases (e.g., Scn-0.8, Scn-0.6), the influences of varying slope angles can be well constrained within the 5% range. In general, the model is mostly insensitive to the uncertainties of any single input parameter.

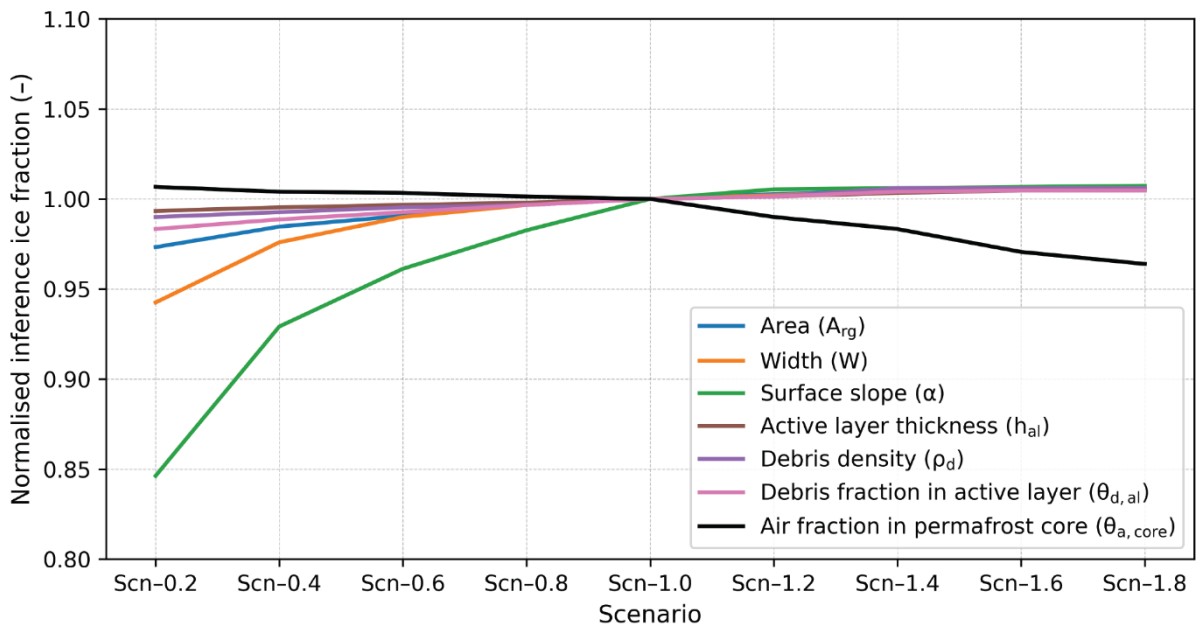

Figure 6: Normalised inference ice fractions from sensitivity experiments with different parameter scenarios. The varying parameters include rock glacier area (blue line), width (orange line), surface slope (green line), active layer thickness (brown line), debris density (purple line), debris fraction in the active layer (pink line), and air faction in permafrost core (black line).

## 4.4 Modelled ice contents in Khumbu and Lhotse valleys

In this subsection, we summarise the characteristics of InSAR-derived surface velocities we used as model constraints (Sect. 4.4.1), followed by the modelled ice content of the five rock glaciers in the study area (Sect. 4.4.2).

### 4.4.1 InSAR-derived surface velocities as model constraints

We used InSAR to derive the downslope surface velocities of five rock glaciers situated in the study region. Surface velocities of the nearby debris-covered glaciers were also measured and presented in Fig. S4 and S5 in the supplementary materials. Figure 7 shows the time series of the InSAR-derived surface velocities of the coherently moving sections of the rock glaciers. We observe that the median and mean velocities of each landform have approximately similar values, and both are capable of

characterising the kinematic status of the landforms. By selecting the mean velocity as the representative value, most rock glaciers, except for Tobuche, moved at a nearly stable rate, ranging from 5 cm yr$^{-1}$ to 30 cm yr$^{-1}$ during the observational period, with the largest standard deviation being 3.4 cm yr$^{-1}$ for Lingten (Fig. 7d). The maximum velocity represents the local extreme of downslope rate and was as high as 112.1±12.4 cm yr$^{-1}$ for Lingten during 2019/07/15–2019/08/26 (Fig. 7d). Tobuche

displayed similar stable behaviour before 2010 but had accelerated more than four times from 14.9±0.2 cm yr$^{-1}$ to 81.4±2.4 cm yr$^{-1}$ between 2010 and 2015 (Fig. 7e). The maximum velocity reached was 181.0±57.4 cm yr$^{-1}$ for the period 2015/03/18–2015/03/22 (Fig. 7e). However, the associated uncertainties during this period were high: the relative uncertainties of mean, median, and maximum velocity were 2.9%, 38.2%, and 31.7%, respectively. The acceleration of Tobuche cannot be confidently revealed by our data and 2015 acquisition was therefore discarded from the velocity series used as the modelling

constraint. The extents of coherently moving parts of the five rock glaciers are presented in Fig. 8, with the average velocities derived from the interferograms obtained during the observation period.

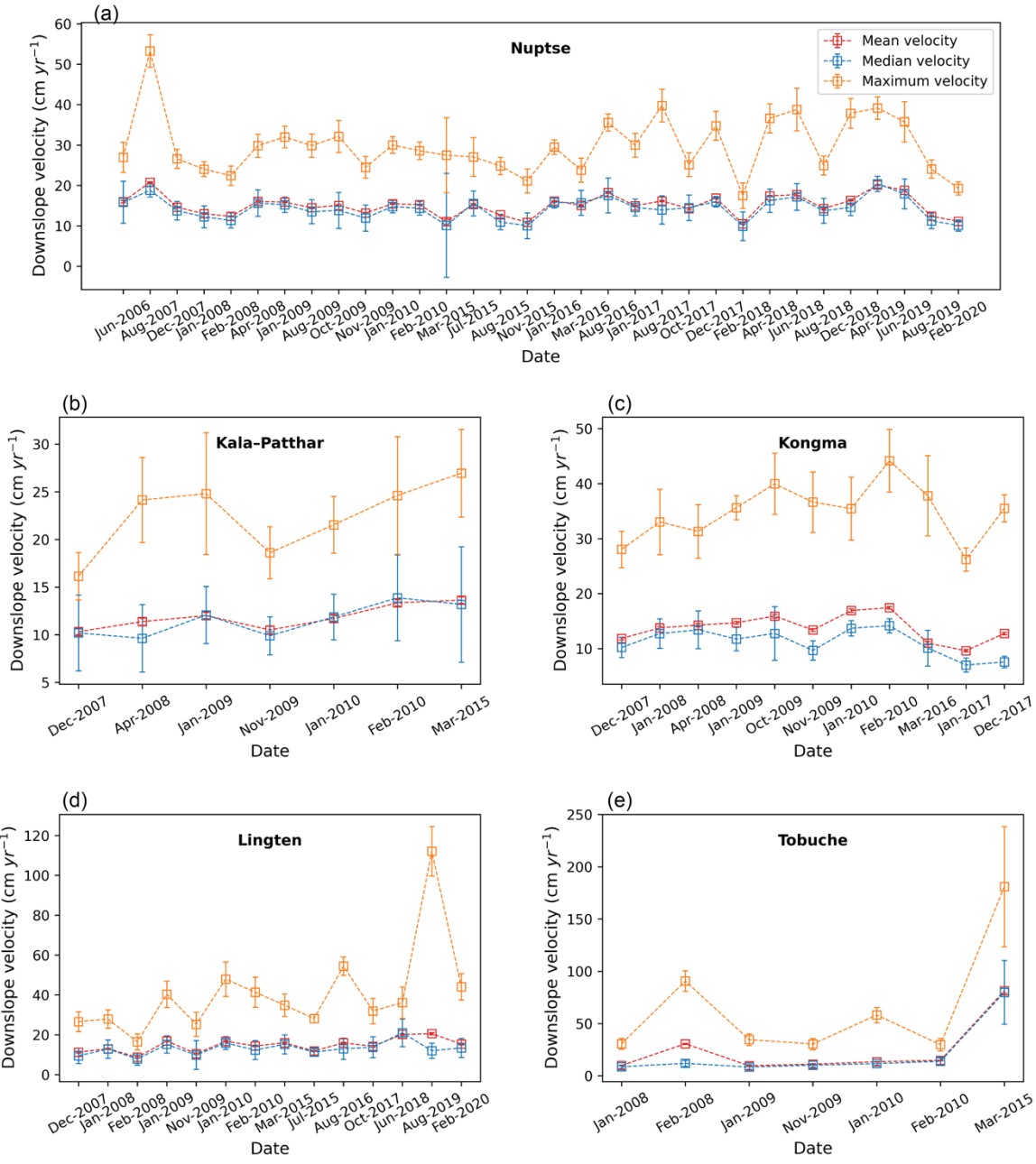

**Figure 7: Time series of the InSAR-derived downslope velocities of the landforms. The spatial mean velocities and uncertainties during each period are shown (red squares and error bars) as well as the median (blue) and maximum (orange) velocities.**

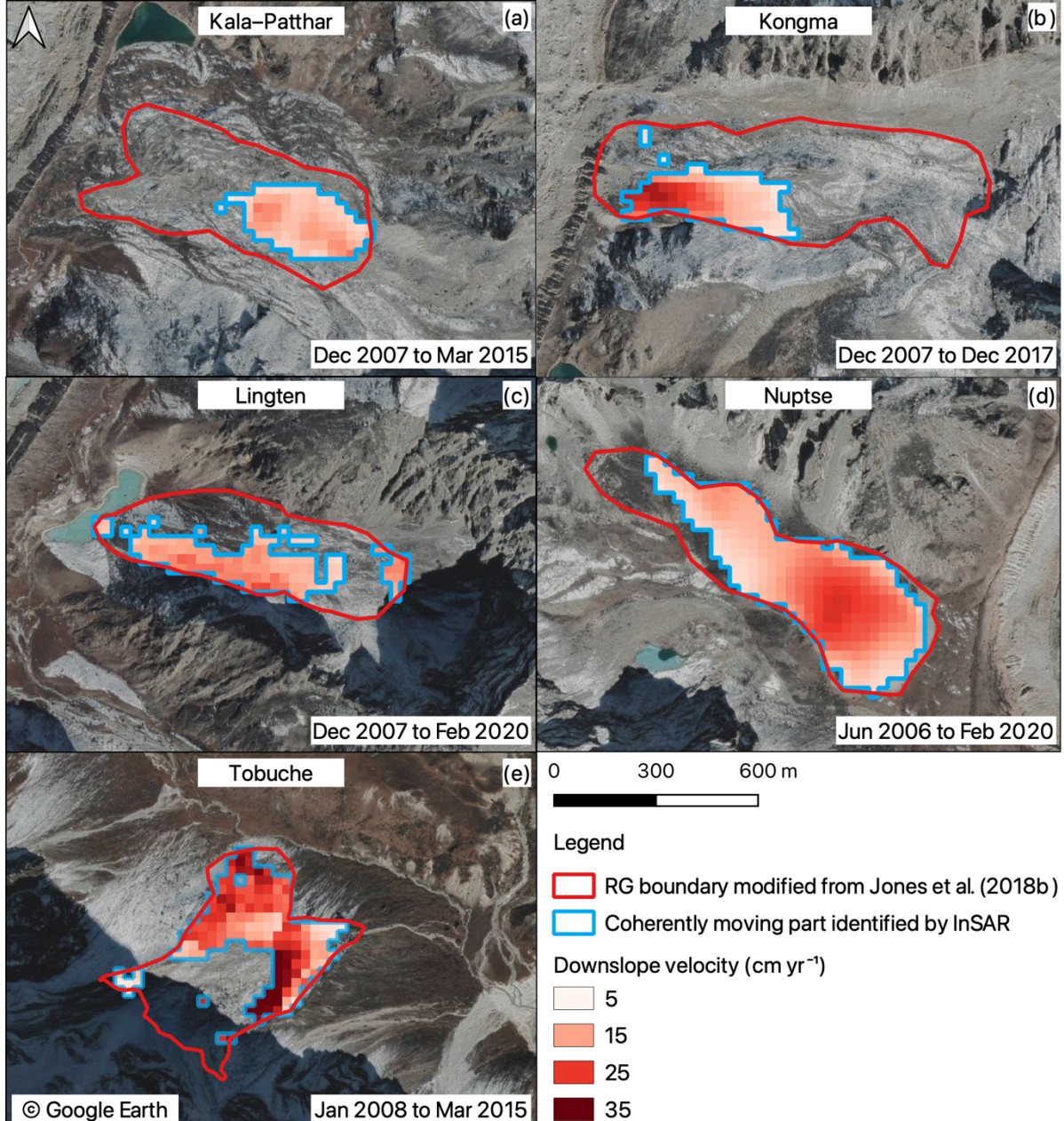


**Figure 8: Velocity field maps show the average velocities of the coherently moving parts of the five rock glaciers (blue outlines) in the study area. The boundaries of the landforms delineated in Jones et al. (2018b) are in red. The transparent areas between the red and blue boundaries are due to low coherence or low velocity during the observational periods.**

### 4.4.2 Modelled ice content

The geometric and structural data used as input parameters are detailed in Table 4. The five rock glaciers are tongue-shaped

features and their areal extents are substantially larger than the three validation rock glaciers (Table 1 and 4). Figure 9 and

Table 5 present the inference ice contents of rock glaciers based on Scheme 2 in the study area. Considering the error of the modelling results (Sect. 4.2, Table 3), the inferred average ice fractions of the coherently moving parts of the landforms range from 70±8% to 74±8%; the water volume equivalent ranges from 1.4±0.2 to 5.9±0.6 million $m^3$ for the coherently moving parts of individual landforms. Nuptse stores the most ice by volume due to its largest dimensions (Table 4). The total amount of water stored in rock glaciers in our study area lies between 12.1 and 15.1 million $m^3$, with an average value of 13.6 million $m^3$.

**Table 4. Summary of the geometric and structural parameters used for inferring ice content of the coherently moving part of rock glaciers in the study area.**

| Rock glacier | Area ($A_{rg}$) (km$^2$) | Width (W) (m) | Active layer thickness ($h_{al}$) (m) | Surface slope ($\alpha$) (°) |
|---|---|---|---|---|
| Kala-Patthar | 0.074 | 240 | 0.68 | 9 |
| Kongma | 0.077 | 300 | 0.83 | 13 |
| Lingten | 0.094 | 240 | 0.65 | 20 |
| Nuptse | 0.234 | 400 | 0.30 | 13 |
| Tobuche | 0.128 | 400 | 1.67 | 16 |

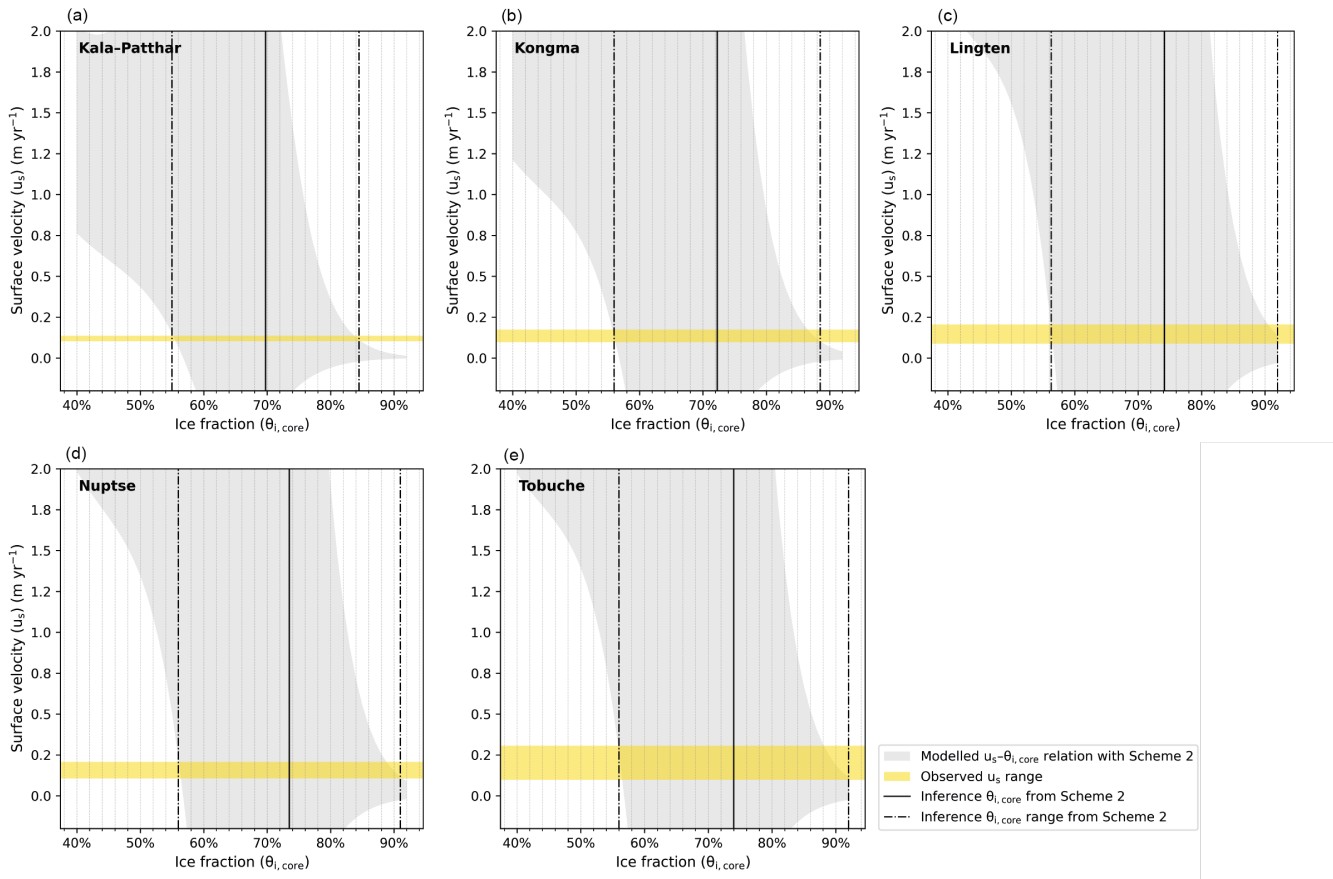

**Figure 9: Modelled relationships between the ice fraction ($\theta_{i,core}$) and the surface velocity ($u_s$) of 95% confidence intervals for the five RGs in Khumbu Valley with model parameterisation Scheme 2 (grey shaded areas). The ranges of the InSAR-derived velocities (yellow bands) are used as the velocity constraints for inferring ice contents from the modelled relationships. The upper and lower boundaries of the estimated ice contents are within the range outlined by the dash-dotted black lines and the solid black lines show the mean values representing the inference ice contents.**

**Table 5. Modelled average ice contents, as well as the minimum and maximum estimates (in brackets) of rock glaciers in Khumbu and Lhotse Valleys and the corresponding water volume equivalents.**

| Rock glacier | Inference ice content (%) | Water volume equivalent (million m³) |
|---|---|---|
| Kala-Patthar | 70±8 | 1.4±0.2 |
| Kongma | 72±8 | 1.5±0.2 |
| Nuptse | 74±8 | 5.9±0.6 |
| Lingten | 74±8 | 2.0±0.2 |
| Tobuche | 74±8 | 2.7±0.3 |

## 4.5 Potential water storage in rock glaciers in the Nepalese Himalaya

Based on the estimated water storage in the five rock glaciers found in this study, the extrapolated amount of water stored in all the intact rock glaciers ranges from 9.0 to 14.0 billion $m^3$ over the entire Nepalese Himalaya, which is the same magnitude as a first-order prediction (16.7 and 25.1 billion $m^3$) made by Jones et al. (2018). In the Nepalese Himalaya, the ratio between the amount of water stored in rock glaciers (11.5 billion $m^3$) and in glaciers (197.6 billion $m^3$) is 1:17. Our modelling-based results are lower than earlier estimates (1:9), yet reveal higher hydrological importance than across the entire Himalayas (1:24) (Jones et al., 2018b; Jones et al., 2021).

## 5 Discussion

We discuss the limitations and prospect of our developed model in this section. Our discussion first focuses on the method limitations in four aspects: (1) incapability of predicting ground ice evolution (Sect. 5.1); (2) limited amount of field data for model calibration (Sect. 5.2); (3) uncertainty in deriving rock glacier thickness (Sect. 5.3); (4) limited application to rock glaciers in quasi-steady-state motion (Sect. 5.4); and (5) uncertainty in estimating regional water storage (Sect. 5.5). Then we present the potential improvements to mitigate the method limitations and the application prospect (Sect. 5.6).

### 5.1 Incapability of predicting ground ice evolution

Our results were presented in the form of a modelled relationship between the ice content and surface velocity (as shown by the grey shading in Fig. 5, S2, S3 and 9), which might mislead the users to interpret the ground ice evolution from rock glacier kinematic variations. For instance, assuming the surface velocity of Kala-Patthar rock glacier reaches 1 m $yr^{-1}$, the corresponding ice fraction would be approximately 60% (detailed in Fig. S6 in the supplement material). However, we cannot draw the conclusion that ground ice stored in Kala-Patthar rock glacier would decrease by 10% if it accelerated to 1 m $yr^{-1}$, because the geometric parameters of the landform would change accordingly, particularly the thickness of the permafrost core and the active layer, making the current modelled relationship no longer valid.

In the proposed approach, we assume that the amount of ice stored in rock glaciers remain constant within the timescale of our study (1–2 decades, constrained by InSAR data), which is consistent with the fact that rock glaciers are currently not a major contribution to surface runoff (Duguay et al., 2015; Jones et al., 2019b). Predicting ground ice changes from kinematic variations is beyond the applicability of our model.

### 5.2 Limited amount of field data for model calibration

The empirical relationship between the effective viscosity and ice content is fundamental to model calibration in this study (Sect. 3.2). Currently, the amount of field data is limited for deriving a statistical relationship with a low degree of uncertainty since detailed knowledge of rock glacier composition is largely lacking.

We relied on the geophysical data obtained from Las Liebres rock glacier in the Andes to calibrate the model (Monnier and Kinnard, 2015b), and hypothesized that the empirical expressions can be generalised to rock glaciers developed in a warm permafrost environment. The validation results achieved from samples in a different region, i.e., the Swiss Alps, proves the

transferability of the model (Sect. 3.3). However, due to the limited amount of calibration data (14 measurements in total), the uncertainty of the derived effective viscosity–ice fraction relationship (dash lines in Fig. 4b) leads to a wide range of propagated uncertainty when modelling the ice content–surface velocity relationship (grey shadings in Fig. 5). More field data are necessary to accurately represent this empirical relationship.

## 5.3 Uncertainty in deriving rock glacier thickness

The uncertainty in deriving rock glacier thickness is discussed here because it influences the surface velocities most significantly. As shown in Eq. 8, the surface velocity is proportional to the thickness to the power of $n + 1$, resulting from the vertical integration of Eq. 7. We use the thickness–area scaling relationship (Eq. 14, Brenning, 2005a) which has also been adopted by previous research on assessing the hydrological importance of rock glaciers (e.g., Azócar and Brenning, 2010; Bodin et al., 2010; Janke et al., 2017; Jones et al., 2018, 2021; Perucca and Esper Angillieri, 2011; Rangercroft et al., 2015; Wagner et al., 2021), yet the reliability of this empirical derivation method has generated discussions (Arenson and Jakob, 2010; Brenning, 2010). Wagner et al. (2021) suggested an adapted relationship by subtracting 10 m from the derived thickness to remove the likely overestimation effect. An alternative empirical method is proposed as a linear relationship between surface slope angle and thickness (Cicoira et al., 2020). We compared the estimated thickness of the validated rock glaciers from the classical thickness–area and the recently established thickness–slope relationships with the field measurements and found that the two sets of results display the same level of error (~2 m, Table S2).

In the validation part, we estimated the thickness-related error by considering the uncertainty involved in delineating the rock glacier area based on Google Earth images, which derives from the occurrence of different image quality and the contrasting interpretations by different operators due to the complex morphology of rock glaciers (Brardinoni et al., 2019; Schmid et al., 2015; Way et al., 2021). We assumed a 40% uncertainty in the area parameter, leading to a ~10% error (or an absolute error of 2–4 m) in thickness. In addition, we conducted analysis assuming a more significant thickness error according to previous studies (Cicoira et al., 2020; Wagner et al., 2021), i.e., 6 m and 10 m, and obtained errors in ice content of 12% and 13%, respectively, which are greater than the 8% uncertainty in our results (Fig. S7 and S8; Table S3).

In general, the uncertainty in deriving rock glacier thickness remains ambiguous, which is primarily attributed to the insufficiency of ground truth data to build a rigorous relationship between the rock glacier thickness and surface parameters (e.g., area, slope). In addition, rock glaciers, especially the talus-derived ones, tend to develop very variable thicknesses across the landform, the distribution of which cannot be inferred using the existing empirical approaches. Thus, the uncertainty introduced by thickness derivation cannot be eliminated when applied to rock glaciers without known information of structure.

## 5.4 Limited application to the coherently moving parts of rock glaciers in quasi-steady-state motion

By using the adapted form of Glen's flow law (Eq. 2), we primarily assumed the rock glacier movement to be steady-state creep driven by viscoelastic deformation of the ice–debris mixture (Moore, 2014). This premise indicates that our method is applicable to rock glaciers currently moving at a relatively stable rate. Recent research has reported abrupt and significant

acceleration of rock glaciers triggered by abnormal surface warming events (Delaloye et al., 2013; Scotti et al., 2017). These
destabilised rock glaciers are beyond the applicability of our method. In this study, we measured surface velocities of rock
glaciers over multiple years to evaluate the stability of the rock glacier motion.

Second, our model is suitable to be applied to the coherently moving part. However, some parts of rock glaciers are in a
transitional kinematic status (practically defined as velocities < 5 cm yr$^{-1}$) or move as an individual portion from the coherently
moving parts. Moreover, the 1-D InSAR method may fail to detect some moving areas of the landforms creeping nearly along
the satellite's flight direction due to the lack of sensitivities of the LOS geometry. These parts may also contain ice but are
excluded from our estimation, causing possible underestimation of ground ice as well.

Third, the motion of rock glaciers undergoing significant subsidence cannot be measured accurately, due to the limitation of
1-D InSAR method: we converted the LOS measurements to surface velocities by assuming the rock glacier moves downslope
without additional subsidence component.

In addition, we also excluded the component of basal sliding processes in our model design (Fig. 3). As observed in the Tien
Shan (Harrison unpubl.), rock glaciers with a melting ice core may undergo basal sliding accompanied by the disruption of
sediment and vegetation at the front of such features. The kinematics of these rock glaciers cannot be appropriately simulated
by our current approach.

## 5.5 Uncertainty in estimating regional water storage

The simple extrapolation method was not designed for an accurate quantification but for an order of magnitude estimation of
the potential water storage in rock glaciers across the Nepalese Himalaya (Sect. 4.5). The inferred average ice content of the
five rock glaciers in the study area lies within a narrow range (70±8% to 74±8%), mainly due to their similar observed
downslope velocities (5–30 cm yr$^{-1}$), used as modelling constraints (Fig. 7; Fig. 8; Sect. 4.4.1). In reality, rock glaciers typically
creep at a rate ranging from decimetre to several metres per year (RGIK – baseline concepts, 2022), thus the average ice
content of the five rock glaciers with similar velocities may not be able to represent that of all rock glaciers with various
velocities in the entire mountain range. Second, the dimensional extent of rock glaciers varies across the Nepalese Himalaya
(Fig. S9). Considering the surface areas of rock glaciers and the thickness–area relationship, the volumes of the landforms lie
between 0.08 million m$^3$ and 228 million m$^3$. The dimensions of Kala-Patthar, Kongma, Nuptse, Tobuche, and Lingten rock
glaciers are at the 26[th], 27[th], 35[th], 50[th], and 72[nd] percentiles of the regional population (Jones et al., 2018), respectively, thus
cannot represent the sizes of all rock glaciers across the mountain range. Therefore, the estimated ratio only serves as a proxy
for assessing the regional hydrological significance of rock glaciers and should be updated as more data become available.

## 5.6 Potential improvements and prospect of the approach

The above discussion on the limitation has demonstrated a critical need for field data from various localities, especially detailed
knowledge of rock glacier composition and internal structure, to reduce the uncertainty in model calibration and to construct
a robust empirical method for deriving rock glacier thickness (Sect. 5.2 and 5.3). In addition, a more accurate 2-D surface

velocity can be obtained by using multi-track InSAR data (e.g., Bertone et al., 2022; Zhang et al., 2021), allowing us to apply the model to rock glaciers with a complex velocity field.

In summary, the lack of ground truth data essentially hinders our approach from achieving high-level accuracy in quantifying ice content of rock glaciers. Nonetheless, the proposed model makes a first attempt to build a framework for inferring ice content with remote sensing-based input by taking advantage of the existing observational data. With the likely emergence of more data to be integrated for model calibration and validation, it is promising to improve the accuracy of the approach. We expect the improved model to be applied to mountain permafrost regions where rock glaciers are widespread for preliminary water storage evaluation.

## 6 Conclusions

We developed an empirical rheological model for inferring ice content of the coherently moving parts of rock glaciers and apply it to estimate the water storage of rock glaciers situated in the Khumbu and Lhotse Valleys using surface velocities derived from InSAR measurements. The main findings are summarised as follows:

(1) An empirical rheological model is presented in this study for estimating ice content of rock glaciers using five input parameters, namely rock glacier area, width, surface slope angle, active layer thickness, and surface velocity, all of which can be obtained from readily available remote sensing products or forthcoming datasets.

(2) Mean downslope velocities of the rock glaciers situated in Khumbu and Lhotse Valleys ranged from 5 cm $yr^{-1}$ to 30 cm $yr^{-1}$ and mostly remained stable during the observational period (2006–2020).

(3) The inferred average ice contents of rock glaciers in Khumbu and Lhotse Valleys ranges from 70±8% to 74±8%; the water volume equivalent ranges from 1.4 to 5.9 million $m^3$ for individual landforms. Nuptse rock glacier stores the most ice due to its largest dimensions among the five studied rock glaciers. Total amount of water stored in the five rock glaciers in Khumbu and Lhotse Valleys ranges from 12.1±0.2 to 15.1±0.6 million $m^3$, with an average value of 13.6 million $m^3$.

(4) Considering previous estimates and extrapolating from our inferred results in Khumbu and Lhotse Valleys, the total amount of water stored in rock glaciers over the Nepalese Himalaya is in the magnitude of 10 billion $m^3$, and the ratio between water storage in rock glaciers and glaciers is 1:17.

This study develops an approach to inferring ice content of rock glaciers by using surface-velocity-constrained model. The estimated ice content and water storage in the study area highlights the hydrological significance of rock glaciers in the Nepalese Himalaya. We argue that the model shows great promise in being able to assess ice storage in rock glaciers although more field data are needed to improve the reliability of this initial modelling framework.

**Code and data availability**

The source code of ISCE is available at https://github.com/isce-framework/isce2. The ALOS PALSAR and ALOS-2 PALSAR-2 data are copyrighted and provided by the Japan Aerospace Exploration Agency through the EO-RA2 project ER2A2N081. Data for the rock glacier kinematics in the Swiss Alps are available at http://www.permos.ch/data.html. The ESA CCI permafrost data are available at http://catalogue.ceda.ac.uk/uuid/1f88068e86304b0fbd34456115b6606f. The code of the modelling approach for estimating ice content will be provided by Yan Hu upon request.

**Author contribution**

YH developed the code, performed the data analysis and interpretation, visualised the results, and wrote the majority of the manuscript. SH conceptualised the research goal, supervised the study, and wrote Sect. 1 of the draft. LL advised YH and actively helped the investigation process. JLW helped formulate the initial framework of the method and collect research data. All the authors contributed to the reviewing and editing of the manuscript.

**Competing interests**

The authors declare that they have no conflict of interest.

**Acknowledgements**

We thank Juliet Ermer for helping to digitalize landform boundaries used in this work.

**Financial support**

This work is supported by CUHK Global Scholarship Programme, CUHK-Exeter Joint Centre for Environmental Sustainability and Resilience (ENSURE, 4930821), the Hong Kong Research Grants Council (CUHK14303417 and HKPFS PF16-03859), and CUHK Direct Grant for Research (4053481).

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
