# Peer review of "Modelling rock glacier ice content based on InSAR-derived velocity, Khumbu and Lhotse Valleys, Nepal"

_The Cryosphere, 2021_

## Author Comment (AC1)

**Responses to RC1**

The study from Hu et al. entitled 'Modelling rock glacier velocity and ice content, Khumbu and Lhotse Valleys, Nepal' proposes a model to infer rock glacier ice content based on InSAR velocity measurements. The model is calibrated based on the observational data of the Chilean Las Libres rock glaciers and validated using data from four rock glaciers in the Alps, before to be applied in NE Nepal. The objective is to estimate the water storage of the rock glaciers at the regional scale.

The research is very comprehensive, the approach is novel and valuable for future studies in similar mountain permafrost environments. The study's scope is well suitable for publication in The Cryosphere. I have no major concern regarding the main methodology and results, but the paper could definitely be improved by modifying the structure, clarifying some steps of the procedure, and extending the discussion. These main points are further explained thereafter. Detailed comments are listed at the end of the review.

Re: We thank the reviewer for his/her insightful, constructive, and detailed comments. We take the suggestions carefully and address all the comments with our point-by-point replies given below. The line numbers refer to the previously submitted discussion paper, aiming to point out where the revisions are made to the discussion paper accordingly.

1. Workflow and structure:

Due to the extensive work of the authors, the complex articulation of the research steps, the multiples datasets and areas used for the model calibration, validation, and application, it is sometimes hard to follow the workflow. I believe that some adjustments of structure may easily help the reader to go through the paper and understand the main elements.

In the abstract (l.15-18), at the end of the introduction (l.58-65) and in Fig.2, the workflow follows a logical order, starting with the model design and finishing with the model application. However, the methods and results sections are upside-down, starting with InSAR data and continuing with the model. Consequently, we go back and forth between the rock glacier sites used at the different steps and the reader gets a bit lost.

For example: 3.2.5. is far after 3.2.1, although the application is based on InSAR. And 4.2 is coming just after the InSAR results in Nepal but the rock glacier velocity mentioned at l.292 is in that case simulated on Swiss rock glaciers.

In addition, I think that Fig.4 is a result and should be added in part 4. The extrapolation to whole Nepal may also be considered as a result (as you also somewhat acknowledge by listing it as a main conclusion at l.478-480).

One suggestion of structure (both for methods and results): model calibration, model validation, sensitivity analysis, model application based on InSAR, regional extrapolation. And then really focus the discussion on the limitations and prospects.

Re: We agree with this more consistent and easy-to-follow structure proposed by the reviewer. We have adopted the suggested paper structure and reconstructed the sub-sections of methods and results into the following sequence: 3.1 model design and assumptions, 3.2 model calibration, 3.3 model validation, 3.4 sensitivity test, 3.5 model application based on InSAR, 3.6 regional extrapolation; 4.1 Calibrated

parameterization schemes, 4.2 model validation, 4.3 model sensitivity, 4.4 modelled ice content in Khumbu and Lhotse valleys, 4.5 potential water storage in rock glaciers in the Nepalese Himalaya.

(Kindly remind that the section/sub-section numbers in this response letter still refer to that in the previously submitted discussion paper)

2. InSAR coherently moving parts:

Something is missing to fully understand your definition of coherently moving parts and why you decided to do so.

At l.109, I don't understand the point (2). It seems to me that it may tend to exaggerate the rate if artificially discarding low velocity. At l.111-112: partly same question: why only higher than 5 cm/yr in more than half of the periods? I don't think it falls into the definition of what is coherent or not, at least not from an InSAR point of view. And from a process point-of-view, what about areas that are coherently not moving (or slowly)?

Do you assume that under < 5 cm/yr there is no more activity/ice, and consider the previous inventory outdated? If yes, it makes somewhat sense but it is important to clearly explain it in the methods and better discuss it in Section 5. If not, one consequence on the results is that the covered areas are much smaller than the initial inventoried landforms (Fig.6, especially for a and b). Did you then extrapolated the ice/water volume to the whole rock glacier, and if not, which potential underestimation may it cause, also for the regional extrapolation presented in Section 5.1?

Re: We set 5 cm yr$^{-1}$ as a threshold for selecting valid InSAR observations (l. 109) in consideration of the conservative estimate of uncertainty in ALOS-1 PALSAR interferometry (Wang et al., 2017).

Then at l. 111–112, we define the coherently moving part mainly for simplifying the non-uniform spatial distribution of surface velocities of rock glaciers in nature, which deviates from the assumed homogeneous model (as illustrated in Fig. 3), where the surface velocities should be constant all over the landform given a homogeneous composition and geometry (as mathematically expressed in Equation 9). To deal with this deviation, we intentionally reduce the spatial and temporal resolution of the InSAR-derived kinematic data by taking the range of spatially averaged velocities of the rock glacier during the observational periods to represent its overall movement. By defining the coherently moving parts, we aim to identify the portion of the landform that approximately corresponds with our designed model (Sect. 3.2.1, Fig. 3) and thus to ensure it is suitable for applying the homogeneous model and inferring an average ice fraction accordingly. We set 5 cm yr$^{-1}$ as a threshold considering that a pixel with a velocity above it is an area actively in motion with the landform as a whole.

For the areas that do not meet the criteria, they may be active (> 5 cm yr$^{-1}$) during certain periods and contain ice, but we doubt whether they should be regarded as part of the assumed homogeneous landform — or from the process point-of-view — whether permafrost in these active areas move along the same plane at depth, as the internal structure of rock glaciers in reality are not homogeneous either.

As regards to the covered areas for estimating ice content in our model application (Sect. 4.4), they are indeed smaller than the inventoried landforms. In this regard, the inference we made is a conservative one. However, the mountain range scale extrapolation (Sect. 5.1) is not drawn from the areal extent of the previously inventoried rock glaciers. We made a simple extrapolation based on the average ice

content of the rock glacier estimated from our study area and the number of the landforms across the Nepalese Himalaya.

3. Method justification vs discussion:

Section 5.2 proposes a relevant list of elements (l.372-375) that can be seen as limitations and supposed to be used to discuss the validity of the approach. However, the way most points are discussed is a bit frustrating: it sounds more like justifying the choices (which should be part of the methods) than acknowledging the limitations and putting the results into a larger context.

For example: at l.403-407: 'we infer that these rock glaciers develop in a warm permafrost environment for the following reasons: …'. This is not really a discussion, rather an explanation for a chosen assumption. In general for 5.2.2: I don't think the question of the warm permafrost assumption has not been really introduced before.

At l.437: 'We introduce this concept because it corresponds with the general model setup.': Saying that it follows the design you chose is not really an explanation, neither a discussion. Overall in 5.2.5: Before justifying it, explain what could be the problems.

In 5.2.6: Ways to tackle the issue are presented (l.451), but the issue itself is not really introduced (saying that the rheology of rock glaciers in Nepal are not necessarily similar than Las Libres).

Re: We appreciate the sound judgement made by the reviewer. Sect. 5.2 lacks the clear acknowledgement of the limitations and discussion in a larger context. Much of the content mentioned here, such as the Sect. 5.2.2, should be re-structured to the methodology section as necessary justifications.

For Sect. 5.2.5, a more detailed explanation has been provided in the previous response. One problem of our definition is the likely underestimation of the ice content in the rock glacier.

In Sect. 5.2.6, we have now clearly stated that the rheology of rock glaciers in Nepal are not necessarily similar to Las Libres.

Additional thinks that could be further discussed in Section 5:

Elements previously mentioned regarding the coherently moving area definition and the update of the inventory using InSAR-kinematics.

How to be sure that the velocity you are measuring is really related to rock glacier creep? As single SAR geometries are used, the values are initially along LOS and could f.ex correspond to subsidence due to melting.

In the model, there is no water at all in the active layer. Is it realistic? Would it change the results if adding a water content as well?

Re: The issues regarding the coherently moving part have been elaborated in the previous response and constructed a discussion section in the revised manuscript.

We acknowledge the limitations of using 1-D InSAR detection for measuring downslope velocities of rock glaciers, though it is a commonly adopted method in recent studies (Brencher et al., 2021; Hu et al., 2021; Liu et al., 2013; Wang et al., 2017). The assumption that the rock glacier moves towards the downslope direction is no longer valid provided that significant subsidence is ongoing. However, the

landforms in our study area are not undergoing melting-induced subsidence, as we do not observe any surface depressions or cracks from optical images.

It is unrealistic that the active layer does not contain any water at all. However, we ignore this variable for two reasons: (1) it is difficult to quantitatively determine or assume the water content stored in the active layer of a rock glacier. (2) in our model setup, the active layer only affects the landform motion by altering the driving force. Therefore, if the new variable, i.e., water fraction in the active layer, is integrated, it would play a similar role as the existing variables including the active layer thickness, the debris density, and the debris fraction in the active layer, all of which are insensitive factors of our model (Sect. 4.3, Fig. 10).

4. Detailed comments:

Title: As you actually used velocity measurements as input to the model in your study area, a title such as 'Modelling rock glacier ice content based on InSAR velocity, Khumbu and Lhotse Valleys, Nepal' would sound more correct to me.

Re: We agree with the suggested title.

l.14 and 16: Repetition 'model to infer ice content of rock glaciers' could be avoided.

Re: Revised. At l. 16, "We apply the model to five rock glaciers in Khumbu and Lhotse Valleys, northern Nepal."

l.21-22: This sentence could be simplified. For ex: 'Due to the accessibility of the model inputs, the approach is easily applicable to permafrost regions where…, and thus valuable to estimate the water storage…'

Re: We have simplified the sentence: "Due to the accessibility of the model inputs, the approach is easily applicable to permafrost regions where previous investigation is lacking, and thus valuable to estimate the water storage potential of the remotely located rock glaciers."

l.29: 'The potential hydrological value of rock glaciers, and thus their importance in terms of hydrological research… Corte (1976); despite this, research...': long sentence, with strange structure and quite some repetitions. Possible to simplify?

Re: We have simplified the sentence: "Corte (1976) first proposed the potential hydrological value of rock glaciers. However, research on the role of rock glaciers in maintaining hydrological stores in mountain catchments remains limited."

l.35: 'triggers' instead of 'produces'? / 'rock slope failure and mountainside collapses': what the difference?

Re: We have re-written the sentence: "The paraglacial response of high mountain slopes would contribute to this process, as glaciers undergo downwasting, which triggers rock slope failures and mountainside collapses and increases the flux of rock debris to glacier surfaces."

l.38: Could start the sentence directly with 'Jones et al. (2021)…'

Re: We have changed the sentence: "Jones et al. (2021) was the first to show that…"

l.39-40: 'The relative importance of rock glacier ice content compared to glaciers in the region is 1:25, …'

Re: We have changed the wording according to the reviewer's suggestion.

l.42-43: Maybe a personal preference and definitively a detail: Easier to write without ; and making two sentences.

Re: We have edited the sentence: "We also expect rock glaciers to provide water supplies long after glaciers have melted. In other high arid mountains, such as the Andes, ice-cored rock glaciers have persisted in valleys long after glacier recession (Azócar and Brenning, 2010; Monnier and Kinnard, 2015a)."

l.45: I don't understand 'the likelihood of glacier-rock glacier transition' part and I believe you are anyway not discussing it in this paper. I would suggest: 'However, there is a lack of modelling studies to test these postulations and assess the hydrological impactions of the glacier-rock glacier transition'. But, if the point of it is to potentially use the results of this study as a baseline, with future updates to see the change of ratio (ice content of RG compared to G), you can also add something about it in the discussion (prospect).

Re: We would like to take the suggestion and change the sentence: "However, there is a lack of modelling studies to test these postulations and assess the hydrological impactions of the glacier-rock glacier transition."

l.47-48: Contradicts with the previous paragraph where you refer to Jones et al. (2021), who have provided quantitative information concerning ice content. You may consider inverse the paragraph order, and replace "absence of quantitive information" by something like "we have little quantitative information".

Re: We have changed the sentence: "We have little quantitative information concerning the ice content of rock glaciers, which hinders our understanding of the likely future hydrological role of rock glaciers."

l.63-65: You are not modelling the kinematic response, you are measuring it and modelling the ice content. Rephrase to for ex: 'We apply the calibrated model for five rok glaciers… and model their ice contents based on remote sensing…'

Re: We have revised the sentence: "Finally, we apply the calibrated model for five rock glaciers in the study area of north-eastern Nepal and model their ice contents based on remote sensing-derived downslope velocities as constraints."

l.67-68: The Khumbu and Lhotse glaciers draining… to remove the unecessary parantheses.

Re: We have changed the sentence: "Among the highest in the world, the Khumbu and Lhotse glaciers draining Everest and Lhotse and have well defined debris-covered snouts."

l.73: Altitudinal limit of permafrost: missing a reference here.

Re: The references are the following two papers: Jakob, 1992; Fujii and Higuchi, 1976. We have changed the sentence: "The five rock glaciers examined in this study are situated at 4900–5090 m a.s.l., near the altitudinal boundary of discontinuous permafrost in the region: previous seismic refraction surveys conducted on active rock glaciers indicate that the lower limit of permafrost occurrence in this

region to be ~5000–5300 m a.s.l. (Jakob, 1992), which is consistent with an earlier estimate of 4900 m a.s.l. based on ground temperature measurements (Fujii and Higuchi, 1976)."

l.78: 'For the period of 1994–2013, recorded accumulated annual precipitation was 449 mm yr-1, …

Re: Modified.

l.83: You give a reference for the delineated RGs, but not for the DCG.

Re: We have added an introduction in the caption (l. 83): "The RGs are delineated by Jones et al. (2018) and the DCGs by the authors based on Google Earth images."

l.85-86: See main comment: here the structure is counter-intuitive (opposite of the introduction).

Re: We have changed the structure according to the reviewer's suggestion.

l.93: I guess here you mean 'We selected the interferograms…'

Re: Yes, we have modified the wording.

l.97: Missing an information about the final resolution you achieve.

Re: We have added a sentence at l. 97: "The final resolution we achieved is ~ 30m."

l.100: How do you know it is stable? Based on visual interpretation? Good to say it. And rather say: 'supposed to be stable'.

Re: Yes, we made the assumption based on visual interpretation of the Google Earth images, for instance, a reference pixel tends to occur at the surface of flat bedrock. We have changed the sentence: "We randomly selected three pixels at places supposed to be stable near each ice–debris landform and …"

l.101: The water vapour is not delayed, the phase is. The end of sentence is also a bit clumsy I think. Maybe 'atmospheric and ionospheric effects including phase delay due to water vapour can be effectively removed because they lead to long-wavelength spatial artefacts and...'

Re: We have revised the sentence to the suggested more accurate one: "By doing so, atmospheric delays can be effectively removed because these lead to long-wavelength artefacts and can be assumed as constant within the range of our study objects."

l.102: 'because these lead to long-wavelength artefacts across the region'.

Re: Modified. See the previous response.

l.105: 'projected ... onto the downslope direction'.

Re: Revised.

l.107-108: The start of the sentence is about the criteria to select valid pixels, while point (1) describes which pixels were discarded. Phrasing in (1) could be inversed (> 0.3 are kept).

Re: We have changed the sentence: "(1) the pixels showing acceptable coherence (> 0.3) are kept…"

l.109: I don't understand point (2). It seems to me that it may tend to exaggerate the rate if artifically discarding low velocity. See main comment about InSAR.

Re: This is due to the precision estimate of InSAR measurement using ALOS-PALSAR data. See our previous response to the main comment.

l.111-112: Partly same question as my point 24: Why that? See main comment.

Re: Please refer to the response to the main comment.

Table 1, caption: List of … interferograms used in the study.

Re: We have revised the caption: "List of ALOS PALSAR and ALOS-2 PALSAR-2 interferograms used in the study."

Figure 2: As I understood, you just used the coherent part as input to the model, so it may be enough to write 'InSAR-derived kinematics on coherently moving parts'.

Re: We have changed the notes of this figure.

[Figure]

l.123: Since you are not assuming shear horizon at depth in the model, it sounds weird to have it mentioned at the second line of the section, without then acknowledging in a way or another the limitation before the discussion.

Re: We have added an acknowledgement immediately after this sentence: "In this study, we neglected the existence of shear horizon in the model design."

l.134-137: I am struggling to understand the point of this part. Too detailed or not enough. What is happening when the critical volumetric debris content is reached? What is the implication for this study? If it is important, one would like to know the actual relation between the ice-debris mixture strength parameter, and the debris content.

Re: When the critical volumetric debris content (42%, according to Moore (2014)) is reached, the presence of rich debris would introduce competing effects to the deformation of ice-debris mixture: on the one hand, increased debris fraction causes strengthening of the mixture by introducing interparticle friction (Ting et al., 1983); on the other hand, the addition of debris decreases the shear strength due to

the lubricating and stress-modulating effects exerted by the unfrozen water concentrating at the debris-ice interfaces (Arenson et al., 2007; Ikeda et al., 2008). In this study, we ignored this complicated mechanism produced by increased debris and assumed that the rock glacier has an ice-rich core, i.e., debris fraction is less than 42%.

We have added a statement at l. 134: "We assumed the rock glacier has an ice-rich core."

l.186 and 190: Little detail: not sure it is necessary to have 'collected by'/'detailed in' before references.

Re: We have removed the phrases.

l.195: '… by Arenson and Springman (2005a) who evidenced a parabolic relationship…'

Re: We have changed the sentence: "This trend was also depicted by Arenson and Springman (2005a) who evidenced a parabolic relationship between the minimum axial creep strain rate and the volumetric ice content."

l.201-204: Instead of using 4 lines, you could just entitle the equation lines: Scheme 1: us = / Scheme 2: us = / Scheme 3: us =

Re: We have made the suggested changes and removed the four lines.

Figure 4: It could be moved to Results. Also, since you numbered the Schemes 1-2-3, it would be good to label the subplots a)-c) accordingly, by adding subtitles to make it easy to understand.

Re: We have labeled the subplots according to this comment.

[Figure]

 Long sentence, hard to understand since it is a double-validation of both the velocity and the ice content. May find a way to rephrase / divide the sentence.

Re: We have re-written the sentence: "We simulated the surface velocity ($u_s$) of each rock glacier by varying volumetric ice content ($\theta_{i,core}$) of the permafrost core. Then we compared the modelled velocity with the measured velocity from Terrestrial Geodetic Surveys (PERMOS, 2019).

l.234: Air density: provide the actual values.

Re: In the validation part, $\rho_a$=1.007 kg/m³, as the range of elevation for the four rock glaciers is between 2600 m and 2900 m.

l.245: Currently not really understandable: what is the usual value range in reality?

Re: We take 1450–3450 kg/m³ as a realistic range, as the lower value…

Table 3: Necessary information? Could be moved to Supplementary, to shorten a bit the really heavy method section.

Re: We have moved the table to the Supplementary File.

l.255: 'Active layer thickness was determined as the mean value over the extent of each rock glacier, based on the 2006–2017 estimate from the…'

Re: We have changed the sentence to: "Active layer thickness was determined as the mean value over the extent of each rock glacier, based on the 2006–2017 estimate from the European Space Agency Permafrost Climate Change Initiative Product (ESA CCI) (Obu et al., 2020)"

l.259: Is the estimate of water based on the whole inventoried rock glacier or the coherently moving part? See main comments.

Re: It is based on the coherently moving part.

l.265: In a way, this table is already a results, as it is based on the coherently moving parts of the rock glaciers, presented later in the paper.

Re: Concur. We have moved the table to the result section.

l.267-268: It cut the workflow to separate InSAR to the model application. See main comment about structure.

Re: We agree with this comment and have changed the manuscript structure accordingly.

l.274: '...approximately similar values'

Re: Modified.

l.276: '...during the observational periods'. You may also emphasize somewhere what the timeseries vary from site to site.

Re: Detailed information is presented in Fig. 5. We did not further analyze the variations among the landforms because here we want to emphasize the common feature that all the rock glaciers move at a nearly stable rate.

l.279: 'since 2010': The evidenced acceleration is based on one value also, i.e. the difference between the two last acquisition dates, right? Maybe writing 'between 2010 and 2015 acquisitions' would be more correct.

Re: We have changed the wording: "Tobuche displayed similar stable behaviour before 2010 but had accelerated by more than four times from $14.9\pm0.2$ cm yr$^{-1}$ to $81.4\pm2.4$ cm yr$^{-1}$ between 2010 and 2015 acquisitions."

In 4.1: More references to the Fig.5 subplots would help the reader to make the link.

Re: We have added more references: "…with the largest standard deviation being 3.4 cm yr$^{-1}$ for Lingten (Fig. 5d). The maximum velocity represents the local extreme of downslope motion and was as high as $112.1\pm12.4$ cm yr$^{-1}$ for Lingten during 2019/07/15–2019/08/26 (Fig. 5d). Tobuche displayed similar stable behaviour before 2010 but had accelerated by more than four times from $14.9\pm0.2$ cm yr$^{-1}$ to $81.4\pm2.4$ cm yr$^{-1}$ between 2010 and 2015 acquisitions (Fig. 5e). The maximum velocity reached $181.0\pm57.4$ cm yr$^{-1}$ during 2015/03/18–2015/03/22 (Fig. 5e)."

Figure 6: Missing scales.

Re: We have added a scale to Fig. 6.

[Figure]

l.293-296: Partly repetition with Methods.

Re: We have removed these three lines.

l.301: Reference and inference ice content: why not simply saying 'observed' vs 'modelled'?

Re: We have changed the wording to 'observed' and 'modelled'.

l.301-303: Missing references to Scheme 1/3 graphs (Fig. 7, 9). If you think there are unnecessary, you may consider moving them to Supplementary.

Re: We have moved Fig. 7 and Fig. 9 to Supplementary.

l.303-304: 'However, the above bias is not statistically useful for correcting the modelling results due to the limited amount of validation data.' Not clear, could be more discussed, here or in Section 5?

Re: Here we want to point out that we did not use the average bias 8.4% to correct the modelled ice content for the rock glaciers in Khumbu Valley. We have changed the sentence: "The above bias (8.4%) is not used for correcting the modelling results due to the limited amount of validation data."

Figure 7: Is it correct to say that the intersect of the yellow & blue lines correspond here to the 'truth' (observed values / references)? If yes, it would be useful to highlight it better (encircle it for ex).

Re: The yellow lines are the observed velocities, and the blue lines are the reference ice content. They somehow correspond to the 'truth'. What we want to highlight in Fig. 7 is the intersection between the grey shade and the yellow band, as marked by dash-dotted black lines. Such intersection indicates the range of estimated ice content.

Figure 8/9, captions: Add full captions instead of referring to 7.

Re: We have added full captions in the Supplementary File.

l.325: 'The model has higher sensitivity to the surface slope angle...

Re: Modified.

l.327: '...the model is mostly sensitive to...'

Re: Modified.

l.334: Interred ice content based on Scheme 2, right?

Re: Yes, we have changed the sentence.

l.336-339: Separate the information related to % and total volume, and add a reference to geometrical information from Table 4 would help making sense of it.

Re: We keep the original structure of this paragraph because the two water volume equivalents were calculated corresponding to the two ice content estimates, i.e., the average value and the inferred range. We have added a reference to Table 4: "Nuptse stores the most ice by volume due to its largest dimensions (Table 4)."

Section 5.1: I would say that it is a result. See main comment.

Re: We have re-organized the paper structure according to the reviewer's comments.

l.364-365: Based on which study? Jones or yours? As you refer to previous research just before, it is not fully clear.

Re: It is Jones's study we are referring to. We have changed the sentence: "...which is in the same magnitude predicted by Jones et al. (2018)."

l.367: '... across the entire Himalayas'

Re: Modified.

l.373: '(3) absence of shear horizon' (also at l.408).

Re: Modified at the two places (l. 373 and l.408).

l.382: '… to evaluate the stability…'

Re: Modified.

l.385: You could probably cut 'This is not surprising'.

Re: We have removed the phrase.

l.391: 'creep parameter' is only mentionned once before and referring to n, not A (l.192). A is described in more general terms at l.128.

Re: A and n both can be referred to as creep parameters. We have made edits to l. 128: "…where A and n are creep parameters reflecting variations in environmental conditions…"

l.413: 'This short-term feature of rock glacier kinematics is assumed to be insignificant…'. And it would be more logical to move this statement at the end of 5.2.3.

Re: We have changed the sentence and moved to the end of this subsection: "This short-term feature of rock glacier kinematics is assumed to be insignificant to modelling the relationship between ice content and multi-annual average movement velocity in our study."

l.421: Add reference to Fig.10.

Re: Added.

l.428-429: 'Thus, the uncertainty introduced… is unavoidable.' I don't see the causal link with the previous sentence here. It is not because Cicoira et al. (2020) also had accuracies at the same level that it is unavoidable.

Re: We have revised the sentence: "The uncertainty introduced by thickness derivation when applied to rock glaciers without known information of structure cannot be eliminated with the existing empirical methods."

l.437-438: 'We introduce this concept because it corresponds with the general model setup.' That is no explanation… Just saying 'we did it because we designed it that way'…

Re: We have provided more detailed explanations (also see our response to the main comment): "We introduce this concept mainly for simplifying the non-uniform spatial distribution of surface velocities of rock glaciers in nature, which deviates from the assumed homogeneous model (as illustrated in Fig. 3), where the surface velocities should be constant all over the landform given a homogeneous composition and geometry (as mathematically expressed in Equation 9). To deal with this deviation, we intentionally reduce the spatial and temporal resolution of the InSAR-derived kinematic data by taking the range of spatial mean velocities of the rock glacier during the observational periods to represent its overall movement. By defining the coherently moving parts, we aim to identify the portion of the landform that approximately corresponds with our designed model (Sect. 3.2.1, Fig. 3) and thus suitable for applying the homogeneous model and inferring an average ice fraction accordingly."

l.443-445: Without more explanations, this is not understandable.

Re: We have removed this unnecessary sentence.

l.451: Which issue? You have not mentioned an issue yet.

Re: We have re-written the sentence: "In addition, we tested this hypothesis in two ways: first…"

l.465 and l.481: 'surface-velocity-constraints': surface velocity wouldn't be enough? To avoid a long word in 3 parts.

Re: We have replaced the long word with 'surface velocity'.

l.469: 'emerging': What does it mean in that case?

Re: We have replaced it with "forthcoming datasets".

Arenson, L. U., Springman, S. M., and Sego, D. C. (2007). The rheology of frozen soils. *Applied Rheology, 17*(1), 12147-12141-12147-12114.

Brencher, G., Handwerger, A. L., and Munroe, J. S. (2021). InSAR-based characterization of rock glacier movement in the Uinta Mountains, Utah, USA. *The Cryosphere, 15*(10), 4823-4844. https://doi.org/10.5194/tc-15-4823-2021

Hu, Y., Liu, L., Wang, X., Zhao, L., Wu, T., Cai, J., Zhu, X., and Hao, J. (2021). Quantification of permafrost creep provides kinematic evidence for classifying a puzzling periglacial landform. *Earth Surface Processes and Landforms, 46*(2), 465-477. https://doi.org/10.1002/esp.5039

Ikeda, A., Matsuoka, N., and Kaab, A. (2008). Fast deformation of perennially frozen debris in a warm rock glacier in the Swiss Alps: An effect of liquid water. *Journal of Geophysical Research-Earth Surface, 113*(F1). https://doi.org/10.1029/2007JF000859

Jones, D. B., Harrison, S., Anderson, K., Selley, H. L., Wood, J. L., and Betts, R. A. (2018). The distribution and hydrological significance of rock glaciers in the Nepalese Himalaya. *Global and Planetary Change, 160*, 123-142. https://doi.org/10.1016/j.gloplacha.2017.11.005

Liu, L., Millar, C. I., Westfall, R. D., and Zebker, H. A. (2013). Surface motion of active rock glaciers in the Sierra Nevada, California, USA: inventory and a case study using InSAR. *The Cryosphere, 7*(4), 1109-1119. https://doi.org/10.5194/tc-7-1109-2013

Moore, P. L. (2014). Deformation of debris-ice mixtures. *Reviews of Geophysics, 52*(3), 435-467. https://doi.org/https://doi.org/10.1002/2014rg000453

Ting, J. M., Torrence Martin, R., and Ladd, C. C. (1983). Mechanisms of strength for frozen sand. *Journal of Geotechnical Engineering, 109*(10), 1286-1302.

Wang, X. W., Liu, L., Zhao, L., Wu, T. H., Li, Z. Q., and Liu, G. X. (2017). Mapping and inventorying active rock glaciers in the northern Tien Shan of China using satellite SAR interferometry. *Cryosphere, 11*(2), 997-1014. https://doi.org/10.5194/tc-11-997-2017

---

## Author Response (AR1)

**Responses to RC1**

The study from Hu et al. entitled 'Modelling rock glacier velocity and ice content, Khumbu and Lhotse Valleys, Nepal' proposes a model to infer rock glacier ice content based on InSAR velocity measurements. The model is calibrated based on the observational data of the Chilean Las Libres rock glaciers and validated using data from four rock glaciers in the Alps, before to be applied in NE Nepal. The objective is to estimate the water storage of the rock glaciers at the regional scale.

The research is very comprehensive, the approach is novel and valuable for future studies in similar mountain permafrost environments. The study's scope is well suitable for publication in The Cryosphere. I have no major concern regarding the main methodology and results, but the paper could definitely be improved by modifying the structure, clarifying some steps of the procedure, and extending the discussion. These main points are further explained thereafter. Detailed comments are listed at the end of the review.

Re: We thank the reviewer for his/her insightful, constructive, and detailed comments. We take the suggestions carefully and address all the comments with our point-by-point replies given below. The line numbers refer to the previously submitted discussion paper, aiming to point out where the revisions are made to the discussion paper accordingly.

1. Workflow and structure:

Due to the extensive work of the authors, the complex articulation of the research steps, the multiples datasets and areas used for the model calibration, validation, and application, it is sometimes hard to follow the workflow. I believe that some adjustments of structure may easily help the reader to go through the paper and understand the main elements.

In the abstract (l.15-18), at the end of the introduction (l.58-65) and in Fig.2, the workflow follows a logical order, starting with the model design and finishing with the model application. However, the methods and results sections are upside-down, starting with InSAR data and continuing with the model. Consequently, we go back and forth between the rock glacier sites used at the different steps and the reader gets a bit lost.

For example: 3.2.5. is far after 3.2.1, although the application is based on InSAR. And 4.2 is coming just after the InSAR results in Nepal but the rock glacier velocity mentioned at l.292 is in that case simulated on Swiss rock glaciers.

In addition, I think that Fig.4 is a result and should be added in part 4. The extrapolation to whole Nepal may also be considered as a result (as you also somewhat acknowledge by listing it as a main conclusion at l.478-480).

One suggestion of structure (both for methods and results): model calibration, model validation, sensitivity analysis, model application based on InSAR, regional extrapolation. And then really focus the discussion on the limitations and prospects.

Re: We agree with this more consistent and easy-to-follow structure proposed by the reviewer. We have adopted the suggested paper structure and reconstructed the sub-sections of methods and results into the following sequence: 3.1 model design and assumptions; 3.2 model calibration; 3.3 model validation; 3.4 sensitivity test; 3.5 model application (3.5.1 deriving surface velocity constraints with Differential InSAR, 3.5.2 deriving geometric and structural parameters from remote sensing products); 3.6 regional

extrapolation; 4.1 Calibrated parameterization schemes; 4.2 model validation; 4.3 model sensitivity; 4.4 modelled ice content in Khumbu and Lhotse valleys (4.4.1 InSAR-derived surface kinematics as model constraints, 4.4.2 modelled ice content); 4.5 potential water storage in rock glaciers in the Nepalese Himalaya.

We have rewritten the discussion section focusing on the limitation and prospects of our work: 5.1 Incapability of predicting ground ice evolution; 5.2 Limited amount of field data for model calibration; 5.3 Uncertainty in deriving rock glacier thickness; 5.4 Limited application to rock glaciers in quasi-steady-state motion; 5.5 Uncertainty in estimating regional water storage; and 5.6 Potential improvements and prospect of the approach.

(Kindly remind that unless clearly stated otherwise, the section/sub-section numbers in this response letter still refer to that in the previously submitted discussion paper)

2. InSAR coherently moving parts:

Something is missing to fully understand your definition of coherently moving parts and why you decided to do so.

At l.109, I don't understand the point (2). It seems to me that it may tend to exaggerate the rate if artificially discarding low velocity. At l.111-112: partly same question: why only higher than 5 cm/yr in more than half of the periods? I don't think it falls into the definition of what is coherent or not, at least not from an InSAR point of view. And from a process point-of-view, what about areas that are coherently not moving (or slowly)?

Do you assume that under < 5 cm/yr there is no more activity/ice, and consider the previous inventory outdated? If yes, it makes somewhat sense but it is important to clearly explain it in the methods and better discuss it in Section 5. If not, one consequence on the results is that the covered areas are much smaller than the initial inventoried landforms (Fig.6, especially for a and b). Did you then extrapolated the ice/water volume to the whole rock glacier, and if not, which potential underestimation may it cause, also for the regional extrapolation presented in Section 5.1?

Re: We set 5 cm yr$^{-1}$ as a threshold for selecting valid InSAR observations (l. 109) in consideration of the conservative estimate of uncertainty in ALOS-1 PALSAR interferometry (Wang et al., 2017).

Then at l. 111–112, we define the coherently moving part mainly for simplifying the non-uniform spatial distribution of surface velocities of rock glaciers in nature, which deviates from the assumed homogeneous model (as illustrated in Fig. 3), where the surface velocities should be constant all over the landform given a homogeneous composition and geometry (as mathematically expressed in Equation 9). To deal with this deviation, we intentionally reduce the spatial and temporal resolution of the InSAR-derived kinematic data by taking the range of spatially averaged velocities of the rock glacier during the observational periods to represent its overall movement. By defining the coherently moving parts, we aim to identify the portion of the landform that approximately corresponds with our designed model (Sect. 3.2.1, Fig. 3) and thus to ensure it is suitable for applying the homogeneous model and inferring an average ice fraction accordingly. We set 5 cm yr$^{-1}$ as a threshold considering that a pixel with a velocity above it is an area actively in motion with the landform as a whole.

For the areas that do not meet the criteria, they may be active (> 5 cm yr$^{-1}$) during certain periods and contain ice, but we doubt whether they should be regarded as part of the assumed homogeneous

landform — or from the process point-of-view — whether permafrost in these active areas move along the same plane at depth, as the internal structure of rock glaciers in reality are not homogeneous either.

As regards to the covered areas for estimating ice content in our model application (Sect. 4.4), they are indeed smaller than the inventoried landforms. In this regard, the inference we made is a conservative one. However, the mountain range scale extrapolation (Sect. 5.1) is not drawn from the areal extent of the previously inventoried rock glaciers. We made a simple extrapolation based on the average ice content of the rock glacier estimated from our study area and the number of the landforms across the Nepalese Himalaya.

3. Method justification vs discussion:

Section 5.2 proposes a relevant list of elements (l.372-375) that can be seen as limitations and supposed to be used to discuss the validity of the approach. However, the way most points are discussed is a bit frustrating: it sounds more like justifying the choices (which should be part of the methods) than acknowledging the limitations and putting the results into a larger context.

For example: at l.403-407: 'we infer that these rock glaciers develop in a warm permafrost environment for the following reasons: …'. This is not really a discussion, rather an explanation for a chosen assumption. In general for 5.2.2: I don't think the question of the warm permafrost assumption has not been really introduced before.

At l.437: 'We introduce this concept because it corresponds with the general model setup.': Saying that it follows the design you chose is not really an explanation, neither a discussion. Overall in 5.2.5: Before justifying it, explain what could be the problems.

In 5.2.6: Ways to tackle the issue are presented (l.451), but the issue itself is not really introduced (saying that the rheology of rock glaciers in Nepal are not necessarily similar than Las Libres).

Re: We appreciate the sound judgement made by the reviewer. Sect. 5.2 lacks the clear acknowledgement of the limitations and discussion in a larger context. Much of the content mentioned here, such as the Sect. 5.2.2, should be re-structured to the methodology section as necessary justifications.

For Sect. 5.2.5, a more detailed explanation has been provided in the previous response.

In Sect. 5.2.6, we have now rewritten the discussion part.

Additional thinks that could be further discussed in Section 5:

Elements previously mentioned regarding the coherently moving area definition and the update of the inventory using InSAR-kinematics.

How to be sure that the velocity you are measuring is really related to rock glacier creep? As single SAR geometries are used, the values are initially along LOS and could f.ex correspond to subsidence due to melting.

In the model, there is no water at all in the active layer. Is it realistic? Would it change the results if adding a water content as well?

Re: Detailed explanations regarding the coherently moving part have been elaborated in the previous response and presented in the methodology part in the revised manuscript (Sect. 3.5.1): "By defining

the coherently moving parts, we aim to identify the portion of the landform that approximately corresponds with our designed model (Sect. 3.1, Fig. 3) and thus to ensure it is suitable for applying the homogeneous model and inferring an average ice fraction accordingly. We set 5 cm yr-1 as a threshold considering that a pixel with a velocity above it is an area actively in motion with the landform as a whole."

We acknowledge the limitations of using 1-D InSAR detection for measuring downslope velocities of rock glaciers, though it is a commonly adopted method in recent studies (Brencher et al., 2021; Hu et al., 2021; Liu et al., 2013; Wang et al., 2017). The assumption that the rock glacier moves towards the downslope direction is no longer valid provided that significant subsidence is ongoing. However, the landforms in our study area are not undergoing melting-induced subsidence, as we do not observe any surface depressions or cracks from optical images. We discussed this aspect in the new discussion section (Sect. 5.4 and 5.5): "…the motion of rock glaciers undergoing significant subsidence cannot be measured accurately, due to the limitation of 1-D InSAR method: we convert the LOS measurements to surface velocities by assuming the rock glacier moves towards downslope direction without additional subsidence component. Rock glaciers showing strong subsidence indicators from optical images, such as surface depressions or cracks, are not suitable for the current method… an accurate 3-D surface velocity can be obtained by using multi-track InSAR method, allowing us to apply the model to rock glaciers with a complex velocity field."

It is unrealistic that the active layer does not contain any water at all. However, we ignore this variable for two reasons: (1) it is difficult to quantitatively determine or assume the water content stored in the active layer of a rock glacier. (2) in our model setup, the active layer only affects the landform motion by altering the driving force. Therefore, if the new variable, i.e., water fraction in the active layer, is integrated, it would play a similar role as the existing variables including the active layer thickness, the debris density, and the debris fraction in the active layer, all of which are insensitive factors of our model (Sect. 4.3, Fig. 10).

4. Detailed comments:

Title: As you actually used velocity measurements as input to the model in your study area, a title such as 'Modelling rock glacier ice content based on InSAR velocity, Khumbu and Lhotse Valleys, Nepal' would sound more correct to me.

Re: We agree with the suggested title.

l.14 and 16: Repetition 'model to infer ice content of rock glaciers' could be avoided.

Re: Revised. At l. 16, "We apply the model to five rock glaciers in Khumbu and Lhotse Valleys, north-eastern Nepal."

l.21-22: This sentence could be simplified. For ex: 'Due to the accessibility of the model inputs, the approach is easily applicable to permafrost regions where…, and thus valuable to estimate the water storage…'

Re: We have simplified the sentence: "Due to the accessibility of the model inputs, the approach is easily applicable to permafrost regions where previous investigation is lacking, and thus valuable to estimate the water storage potential of the remotely located rock glaciers."

 'The potential hydrological value of rock glaciers, and thus their importance in terms of hydrological research… Corte (1976); despite this, research...': long sentence, with strange structure and quite some repetitions. Possible to simplify?

Re: We have simplified the sentence: "Corte (1976) first proposed the potential hydrological value of rock glaciers, yet research on the role of rock glaciers in maintaining hydrological stores in mountain catchments remains limited."

l.35: 'triggers' instead of 'produces'? / 'rock slope failure and mountainside collapses': what the difference?

Re: We have re-written the sentence: "The paraglacial response of high mountain slopes would contribute to this process, as glaciers undergo downwasting, which triggers rock slope failures and mountainside collapses and increases the flux of rock debris to glacier surfaces."

l.38: Could start the sentence directly with 'Jones et al. (2021)…'

Re: We have changed the sentence: "Jones et al. (2021) was the first to show that…"

l.39-40: 'The relative importance of rock glacier ice content compared to glaciers in the region is 1:25, …'

Re: To make it clear, we have also taken the suggestion from another reviewer and changed the sentence: "The ratio between rock glacier ice content and that in glaciers in the region was 1:25…"

l.42-43: Maybe a personal preference and definitively a detail: Easier to write without ; and making two sentences.

Re: We have edited the sentence: "In the future, we expect rock glaciers with high ice content to provide water supplies long after glaciers have melted. In other high arid mountains, such as the Andes, ice-cored rock glaciers have persisted in valleys long after glacier recession (Azócar and Brenning, 2010; Monnier and Kinnard, 2015a)."

l.45: I don't understand 'the likelihood of glacier-rock glacier transition' part and I believe you are anyway not discussing it in this paper. I would suggest: 'However, there is a lack of modelling studies to test these postulations and assess the hydrological impactions of the glacier-rock glacier transition'. But, if the point of it is to potentially use the results of this study as a baseline, with future updates to see the change of ratio (ice content of RG compared to G), you can also add something about it in the discussion (prospect).

Re: The reviewer is right. We are not going to discuss the transition in this study. In the revised manuscript, we have rewritten this part and removed this sentence.

l.47-48: Contradicts with the previous paragraph where you refer to Jones et al. (2021), who have provided quantitative information concerning ice content. You may consider inverse the paragraph order, and replace "absence of quantitative information" by something like "we have little quantitative information".

Re: We have changed the sentence: "To date, we have little quantitative information concerning the ice content of rock glaciers, which hinders our understanding of the potential future hydrological role of rock glaciers."

l.63-65: You are not modelling the kinematic response, you are measuring it and modelling the ice content. Rephrase to for ex: 'We apply the calibrated model for five rock glaciers… and model their ice contents based on remote sensing…'

Re: We have revised the sentence: "Finally, we apply the calibrated model for five rock glaciers in the study area of north-eastern Nepal and model their ice contents based on remote sensing-derived downslope velocities as constraints."

l.67-68: The Khumbu and Lhotse glaciers draining… to remove the unnecessary parentheses.

Re: We have changed the sentence: "Among the highest in the world, the Khumbu and Lhotse glaciers draining Everest and have well defined debris-covered snouts."

l.73: Altitudinal limit of permafrost: missing a reference here.

Re: The references are the following two papers: Jakob, 1992; Fujii and Higuchi, 1976. We have changed the sentence: "The five rock glaciers examined in this study are situated at 4900–5090 m a.s.l., near the altitudinal boundary of discontinuous permafrost in the region: previous seismic refraction surveys conducted on active rock glaciers indicate that the lower limit of permafrost occurrence in this region to be ~5000–5300 m a.s.l. (Jakob, 1992), which is consistent with an earlier estimate of 4900 m a.s.l. based on ground temperature measurements (Fujii and Higuchi, 1976)."

l.78: 'For the period of 1994–2013, recorded accumulated annual precipitation was 449 mm yr-1, …

Re: Modified.

l.83: You give a reference for the delineated RGs, but not for the DCG.

Re: We have added an introduction in the caption (l. 83): "The RGs are delineated by Jones et al. (2018) and the DCGs by the authors based on Google Earth images."

l.85-86: See main comment: here the structure is counter-intuitive (opposite of the introduction).

Re: We have changed the structure according to the reviewer's suggestion.

l.93: I guess here you mean 'We selected the interferograms…'

Re: Yes, we have modified the wording.

l.97: Missing an information about the final resolution you achieve.

Re: We have added a sentence at l. 97: "The final resolution we achieved is ~ 30m."

l.100: How do you know it is stable? Based on visual interpretation? Good to say it. And rather say: 'supposed to be stable'.

Re: Yes, we made the assumption based on visual interpretation of the Google Earth images, for instance, a reference pixel tends to occur at the surface of flat bedrock. We have changed the sentence: "We randomly selected three pixels at places supposed to be stable near each ice–debris landform and …"

l.101: The water vapour is not delayed, the phase is. The end of sentence is also a bit clumsy I think. Maybe 'atmospheric and ionospheric effects including phase delay due to water vapour can be effectively removed because they lead to long-wavelength spatial artefacts and...'

Re: We have revised the sentence to the suggested more accurate one: "By doing so, atmospheric delays can be effectively removed because these lead to long-wavelength artefacts and can be assumed as constant within the range of our study objects."

l.102: 'because these lead to long-wavelength artefacts across the region'.

Re: Modified. See the previous response.

l.105: 'projected ... onto the downslope direction'.

Re: Revised.

l.107-108: The start of the sentence is about the criteria to select valid pixels, while point (1) describes which pixels were discarded. Phrasing in (1) could be inversed (> 0.3 are kept).

Re: We have changed the sentence: "(1) the pixels showing acceptable coherence (> 0.3) are kept…"

l.109: I don't understand point (2). It seems to me that it may tend to exaggerate the rate if artifically discarding low velocity. See main comment about InSAR.

Re: This is due to the precision estimate of InSAR measurement using ALOS-PALSAR data. See our previous response to the main comment.

l.111-112: Partly same question as my point 24: Why that? See main comment.

Re: Please refer to the response to the main comment.

Table 1, caption: List of … interferograms used in the study.

Re: We have revised the caption: "List of ALOS PALSAR and ALOS-2 PALSAR-2 interferograms used in the study."

Figure 2: As I understood, you just used the coherent part as input to the model, so it may be enough to write 'InSAR-derived kinematics on coherently moving parts'.

Re: We have changed the notes of this figure.

[Figure]

l.123: Since you are not assuming shear horizon at depth in the model, it sounds weird to have it mentioned at the second line of the section, without then acknowledging in a way or another the limitation before the discussion.

Re: We have replaced the sentence with: "Active rock glaciers are viscous flow features distributed in ice-rich alpine permafrost (Ballantyne, 2018; Berthling, 2011; Haeberli, 2000)."

l.134-137: I am struggling to understand the point of this part. Too detailed or not enough. What is happening when the critical volumetric debris content is reached? What is the implication for this study? If it is important, one would like to know the actual relation between the ice-debris mixture strength parameter, and the debris content.

Re: When the critical volumetric debris content (42%, according to Moore (2014)) is reached, the presence of rich debris would introduce competing effects to the deformation of ice-debris mixture: on the one hand, increased debris fraction causes strengthening of the mixture by introducing interparticle friction (Ting et al., 1983); on the other hand, the addition of debris decreases the shear strength due to the lubricating and stress-modulating effects exerted by the unfrozen water concentrating at the debris-ice interfaces (Arenson et al., 2007; Ikeda et al., 2008). In this study, we ignored this complicated mechanism produced by increased debris and assumed that the rock glacier has an ice-rich core, i.e., debris fraction is less than 42%.

We have added a statement at l. 134: "We assumed the rock glacier has an ice-rich core."

l.186 and 190: Little detail: not sure it is necessary to have 'collected by'/'detailed in' before references.

Re: We have removed the phrases.

l.195: '… by Arenson and Springman (2005a) who evidenced a parabolic relationship…'

Re: We have changed the sentence: "This trend was also depicted by Arenson and Springman (2005a) who evidenced a parabolic relationship between the minimum axial creep strain rate and the volumetric ice content."

l.201-204: Instead of using 4 lines, you could just entitle the equation lines: Scheme 1: us = / Scheme 2: us = / Scheme 3: us =

Re: We have made the suggested changes and removed the four lines.

Figure 4: It could be moved to Results. Also, since you numbered the Schemes 1-2-3, it would be good to label the subplots a)-c) accordingly, by adding subtitles to make it easy to understand.

Re: We have labeled the subplots according to this comment.

[Figure]

l.213-216: Long sentence, hard to understand since it is a double-validation of both the velocity and the ice content. May find a way to rephrase / divide the sentence.

Re: We have re-written the sentence: "We simulated the surface velocity ($u_s$) of each rock glacier by varying volumetric ice content ($\theta_{i,core}$) of the permafrost core. Then we compared the modelled velocity with the measured velocity from Terrestrial Geodetic Surveys (PERMOS, 2019).

l.234: Air density: provide the actual values.

Re: In the validation part, $\rho_a$=1.007 kg/m$^3$, as the range of elevation for the four rock glaciers is between 2600 m and 2900 m.

l.245: Currently not really understandable: what is the usual value range in reality?

Re: We have added the realistic range in the sentence: "…we changed the value range to be consistent with the usual value range in reality ($\rho_d$: 1450–3450 kg/m$^3$; $\theta_{d,al}$: 13–93%)"

Table 3: Necessary information? Could be moved to Supplementary, to shorten a bit the really heavy method section.

Re: We have moved the table to the Supplementary File.

l.255: 'Active layer thickness was determined as the mean value over the extent of each rock glacier, based on the 2006–2017 estimate from the…'

Re: We have changed the sentence to: "Active layer thickness was determined as the mean value over the extent of each rock glacier, based on the 2006–2017 estimate from the European Space Agency Permafrost Climate Change Initiative Product (ESA CCI) (Obu et al., 2020)"

l.259: Is the estimate of water based on the whole inventoried rock glacier or the coherently moving part? See main comments.

Re: It is based on the coherently moving part.

l.265: In a way, this table is already a results, as it is based on the coherently moving parts of the rock glaciers, presented later in the paper.

Re: Concur. We have moved the table to the result section.

l.267-268: It cut the workflow to separate InSAR to the model application. See main comment about structure.

Re: We agree with this comment and have changed the manuscript structure accordingly.

l.274: '...approximately similar values'

Re: Modified.

l.276: '...during the observational periods'. You may also emphasize somewhere what the timeseries vary from site to site.

Re: Detailed information is presented in Fig. 5. We did not further analyze the variations among the landforms because here we want to emphasize the common feature that all the rock glaciers move at a nearly stable rate.

l.279: 'since 2010': The evidenced acceleration is based on one value also, i.e. the difference between the two last acquisition dates, right? Maybe writing 'between 2010 and 2015 acquisitions' would be more correct.

Re: We have changed the wording: "Tobuche displayed similar stable behaviour before 2010 but had accelerated by more than four times from $14.9\pm0.2$ cm yr$^{-1}$ to $81.4\pm2.4$ cm yr$^{-1}$ between 2010 and 2015 acquisitions."

In 4.1: More references to the Fig.5 subplots would help the reader to make the link.

Re: We have added more references: "…with the largest standard deviation being 3.4 cm yr$^{-1}$ for Lingten (Fig. 5d). The maximum velocity represents the local extreme of downslope motion and was as high as $112.1\pm12.4$ cm yr$^{-1}$ for Lingten during 2019/07/15–2019/08/26 (Fig. 5d). Tobuche displayed similar stable behaviour before 2010 but had accelerated by more than four times from $14.9\pm0.2$ cm yr$^{-1}$ to $81.4\pm2.4$ cm yr$^{-1}$ between 2010 and 2015 acquisitions (Fig. 5e). The maximum velocity reached $181.0\pm57.4$ cm yr$^{-1}$ during 2015/03/18–2015/03/22 (Fig. 5e)."

Figure 6: Missing scales.

Re: We have added a scale to Fig. 6.

[Figure]

l.293-296: Partly repetition with Methods.

Re: We have removed the first part of the sentence.

l.301: Reference and inference ice content: why not simply saying 'observed' vs 'modelled'?

Re: We have changed the wording to 'observed' and 'modelled'.

l.301-303: Missing references to Scheme 1/3 graphs (Fig. 7, 9). If you think there are unnecessary, you may consider moving them to Supplementary.

Re: We have added references to Fig. 7 and Fig. 9.

l.303-304: 'However, the above bias is not statistically useful for correcting the modelling results due to the limited amount of validation data.' Not clear, could be more discussed, here or in Section 5?

Re: In the revised manuscript, we have changed our interpretation of the bias and modified the sentence accordingly: "We used the average bias derived from Scheme 2 to represent the uncertainty of our approach**.**"

Figure 7: Is it correct to say that the intersect of the yellow & blue lines correspond here to the 'truth' (observed values / references)? If yes, it would be useful to highlight it better (encircle it for ex).

Re: The yellow lines are the observed velocities, and the blue lines are the reference (or observed) ice content. They somehow correspond to the 'truth'. What we want to highlight in Fig. 7 is the intersection between the grey shade and the yellow band, as marked by dash-dotted black lines. Such intersection indicates the range of estimated ice content.

We also added a description in the caption of Fig. 8: "For each rock glacier, the intersection between the simulated $\theta_{i,core}$ - $u_s$ relationship (grey shaded area) and the observed velocity (yellow band) gives the estimated range of ice content, as marked by the dash-dotted black lines. The inference ice content is taken as the average value of the estimated range and indicated by the solid black line."

Figure 8/9, captions: Add full captions instead of referring to 7.

Re: We have added full captions to Fig. 8 and 9.

l.325: 'The model has higher sensitivity to the surface slope angle...

Re: Modified.

l.327: '...the model is mostly sensitive to...'

Re: Modified.

l.334: Interred ice content based on Scheme 2, right?

Re: Yes, we have changed the sentence.

l.336-339: Separate the information related to % and total volume, and add a reference to geometrical information from Table 4 would help making sense of it.

Re: We keep the original structure of this paragraph because the two water volume equivalents were calculated corresponding to the two ice content estimates, i.e., the average value and the inferred range. We have added a reference to Table 4: "Nuptse stores the most ice by volume due to its largest dimensions (Table 4)."

Section 5.1: I would say that it is a result. See main comment.

Re: We have re-organized the paper structure according to the reviewer's comments.

l.364-365: Based on which study? Jones or yours? As you refer to previous research just before, it is not fully clear.

Re: It is Jones's study we are referring to. We have changed the sentence: "…which is in the same magnitude of a first-order prediction made by Jones et al. (2018)."

l.367: '… across the entire Himalayas'

Re: Modified.

l.373: '(3) absence of shear horizon' (also at l.408).

Re: We have restructured the manuscript and paraphrased the relevant sentences: "Here we neglected the presence of shear horizon…"

l.382: '… to evaluate the stability…'

Re: Modified.

l.385: You could probably cut 'This is not surprising'.

Re: We have removed the phrase.

l.391: 'creep parameter' is only mentioned once before and referring to n, not A (l.192). A is described in more general terms at l.128.

Re: A and n both can be referred to as creep parameters. We have made edits to l. 128: "…where A and n are creep parameters reflecting variations in environmental conditions…"

l.413: 'This short-term feature of rock glacier kinematics is assumed to be insignificant…'. And it would be more logical to move this statement at the end of 5.2.3.

Re: As the manuscript has been restructured, we have changed the sentence and moved to the Sect. 3.5.1 where we describe the method to derive surface velocity constraints with InSAR : "…the short-term feature of rock glacier kinematics is assumed to be insignificant to modelling the relationship between ice content and multi-annual average movement velocity in our study."

l.421: Add reference to Fig.10.

Re: We have rewritten the discussion on rock glacier thickness derivation.

l.428-429: 'Thus, the uncertainty introduced… is unavoidable.' I don't see the causal link with the previous sentence here. It is not because Cicoira et al. (2020) also had accuracies at the same level that it is unavoidable.

Re: We have rewritten the discussion on thickness derivation and revised the sentence: "The uncertainty introduced by thickness derivation when applied to rock glaciers without known information of structure cannot be eliminated at present."

l.437-438: 'We introduce this concept because it corresponds with the general model setup.' That is no explanation… Just saying 'we did it because we designed it that way'…

Re: In the restructured manuscript, we have added the method justification in Sect. 3.5.1: "By defining the coherently moving parts, we aim to identify the portion of the landform that approximately corresponds with our designed model (Sect. 3.1, Fig. 3) and thus suitable for applying the homogeneous model and inferring an average ice fraction accordingly."

l.443-445: Without more explanations, this is not understandable.

Re: We have removed this unnecessary sentence.

l.451: Which issue? You have not mentioned an issue yet.

Re: We have rewritten this discussion section.

 'surface-velocity-constraints': surface velocity wouldn't be enough? To avoid a long word in 3 parts.

Re: We have replaced the long word with 'surface velocities' in the first place, and kept the second one to form a short sentence.

 'emerging': What does it mean in that case?

Re: We have replaced it with "forthcoming datasets".

**Responses to RC2**

1. I read the manuscript with great interest and anticipation, and I do like to congratulate the authors on their effort of presenting a manuscript that is, editorially, well written and properly developed. However, I do have some fundamental concerns with the approach and assumptions used and I therefore do not recommend that the text is published in The Cryosphere as presented. Unfortunately, this manuscript is following a trend that I have observed in recent years, in particular where authors publish their work without having sufficient field data to support it. I do understand that there are only a few data available on rock glaciers, but that is a fact we must accept, which also means that theoretical approaches that heavily depend on reliable field data for calibration and validation simply should not be published. Therefore, I strongly believe that the approach shall be revisited before the authors submit a revised manuscript.

Re: We thank Dr. Arenson for his insightful and constructive comments. We take the suggestions carefully and address all the comments with our point-by-point replies given below. The line numbers refer to the previously submitted discussion paper, aiming to point out where the revisions are made to the discussion paper accordingly.

(Kindly remind that unless clearly stated otherwise, the section/sub-section numbers in this response letter still refer to that in the previously submitted discussion paper)

2. I will address individual aspects below, but the fundamental problem I have with the proposed approach is that following the author's approach, we would infer that the ground ice contents of the rock glaciers in the Alps, for example, is increasing in response to climate change. Several studies show that the creep velocities of rock glaciers are increasing in the Alps (Note: I specifically do not add many references in this paragraph as I'm sure the authors are well aware of this literature, and I would not be able to pay justice to all the authors that have contributed to some of the elementary statements I use). So, if we were to calculate the ice content using today's surface velocities, and then repeat the calculation again in 10 years it is very likely that an increase in ground ice content would result. This is a fundamental mistake, illustrating that it is impossible to link the parameters used with rock glacier surface velocities in order to estimate ice contents without making huge mistakes. The change in velocity that we are currently monitoring, mainly in the Alps, is related to permafrost degradation in the rock glaciers, specifically the warming of the ice and potential increase in unfrozen water content. Both impacting the creep parameters. While the actual ground ice content may not even change, creep velocities increase in response to warmer conditions, another aspect that was not included in the proposed model where ground temperatures are assumed to be constant. Ground ice melt in rock glaciers in response to climate change is extremely slow because of the latent heat. The higher the ground ice content, which would in turn benefit higher creep velocities, the more latent heat is stored in the ground, requiring more energy (time) in order to melt the ground ice. In other words, there are multiple processes at play that influence the ground ice content, the degradation and the velocity. The simplified approach presented does not consider this complexity, which, as illustrated above, could result in erroneous conclusions.

I do understand that specifically section 5.2 addresses the uncertainties, but if the authors would read those lines carefully, they would probably agree that they are telling the reader that there are so many uncertainties that even the authors are no longer sure if the approach is realistic or not. This is a dangerous approach because on the one hand the manuscript provides a very clear approach on how to calculate ice contents, but at the same time, the paper also says that it may actually all not be correct because of all the simplified assumptions used. For example, I appreciate that the authors indicate the

acceleration in one sentence on line 378. However, this does not resolve the major flaw of the paper indicated above. Like many researchers, the authors assume some sort of steady-state behaviour, which is typically an accurate assumption when modelling glacier dynamics. However, rock glacier kinematics responds on different time scales and therefore it is inaccurate to use assumption tailored for quasi steady state conditions on a process (landform) that is constant transition, always lagging behind modern climate conditions.

Re: We would like to clarify that our proposed approach cannot predict an increasing trend of ground ice content. Firstly, the ice content and surface velocity are not positively correlated in the modelled relationship. Take the Schafberg rock glacier presented in Fig. 8 as an example: if the surface velocity increased slightly (orange band in Fig. R1), the inferred ice content would actually decrease (dotted line in Fig. R1); and if the velocity increased significantly (red band in Fig. R1), the inferred ice content would again increase (dash-dotted line in Fig. R1).

[Figure]

**Figure R1: Revised based on Fig. 8 in the discussion paper. The grey shaded area is the simulated relationship between ice fraction and surface velocity. The yellow band shows surface velocity derived from in-situ measurement, and the solid vertical line indicates the inferred ice content (69%). The orange band presents a hypothetical velocity range, which is higher than the real data (yellow band), and the corresponding estimated ice content is shown by the dotted line (58.5%). Similarly, the red band shows a scenario where the surface velocity is very high, and the ice content is indicated by the dash-dotted line (63.5%).**

However, we do not indicate that while a rock glacier is accelerating, its ice content would first decrease then increase, which is obviously unrealistic. We should have clearly stated in the manuscript that our proposed approach is unable to model the evolution of ice content from the velocity variations, because if one considers the melting/formation of ground ice, the geometric parameters of the rock glacier would change accordingly, particularly the thickness of the permafrost core and the active layer, respectively. We had tried to deal with the ground ice variations but found it difficult to model the geometric changes due to the complexity of degradation mechanisms. In addition, it would be fundamentally flawed to

model ground ice changes without introducing a temperature evolution scheme. In the proposed model, we use the fixed thickness parameter derived from the current rock glacier geometry, and a constant ground temperature distribution. In fact, we quite agree with the reviewer's thought that "ground ice melt in rock glaciers in response to climate change is extremely slow because of the latent heat" and "while the actual ground ice content may not even change, creep velocities increase in response to warmer conditions…", and actually assume that within the time frame concerned in our study (1–2 decades, constrained by available InSAR data), the ice content of rock glaciers remains constant, although their velocities may change. In other words, the proposed approach aims to estimate the current amount of ground ice stored in rock glaciers from their surface velocities in recent years. Predicting ground ice changes from kinematic variations is beyond the scope of our work.

To clarify this point, we firstly added a statement at the end of the introduction section in the revised manuscript: "The proposed approach aims to estimate the current amount of ground ice stored in rock glaciers and to assess the hydrological importance of rock glaciers as freshwater reservoir in the long term." Then we elaborated this point in the introduction section: "Few studies have investigated the hydrological contribution of rock glaciers to surface runoffs at annual or seasonal timescale (e.g., Geiger et al., 2014; Harrington et al., 2018; Krainer and Mostler, 2002; Winkler et al., 2016), and little evidence has shown that rock glacier discharge is a prominent water source at present due to the insulation effect produced by their blocky surfaces (Duguay et al., 2015; Jones et al., 2019; Pruessner et al., 2021). Yet, on the multi-annual to centennial and millennial timescales, we expect rock glaciers of high ice content to serve as water reservoirs long after glaciers have melted." In the new discussion section, we introduced the incapability of predicting ground ice evolution to avoid possible misapplication.

Then we would like to explain how we understand the steady-state creep assumption of rock glaciers adopted by this work. We take advantage of the multi-temporal measurements conducted by InSAR, as detailed in Sect. 4.1, to make sure that the motion of the studied objects is in a quasi-steady state condition. For instance, the sudden acceleration Tobuche rock glacier exhibited (Fig. 5e) has been excluded from the velocity range for inferring ice content. Slight kinematic fluctuations exist (5–30 cm yr$^{-1}$, line 276, Fig. 5) partly due to seasonal variations. In fact, we take the velocity range as a constraint to estimate ice content (line 256–258), which is consistent with our underlying assumption that within the timescale under consideration, the rock glacier is in quasi-steady-state creep and contains constant amount of ice, as elaborated in the previous paragraph.

In the revised manuscript, we have added justifications in the subsection to describe the method of deriving velocity constraints: "…we take the range of the mean velocities over the observational period as the velocity constraint for modelling ice content. By doing so, the short-term feature of rock glacier kinematics is assumed to be insignificant to modelling the relationship between ice content and multi-annual average movement velocity in our study."

As regards with the other assumptions discussed in Sect. 5.2, we are aware of the fact that discussing the deviation of assumptions from reality may leave the readers with an impression that the model is unreliable. However, certain simplifications are inevitable in modelling work and deserve sufficient justifications, which motivates us to develop the arguments presented in Sect. 5.2. One way to provide these necessary justifications without overstressing the uncertainties is to move them to corresponding places in the methodology section, as also suggested by another reviewer. The new discussion section focuses on the limitations and prospects of our approach.

3. On line 481 the authors conclude that "This study demonstrates the effectiveness of inferring ice content of rock glaciers by using a surface-velocity-constrained." However, that is not really what this

paper is doing. The proposed approach uses such a correlation, assuming it is accurate, not demonstrating. There is a lack of data that can actually be used to demonstrate that the proposed approach is valid. The authors are therefore turning the initial hypothesis into a conclusion without proofing it.

Re: We agree that line 481 is an inappropriate statement, though we tried to validate the approach using field data from four Swiss rock glaciers (Sect. 3.2.3 and 4.2), in-situ measured ice content of rock glaciers in Khumbu Valley is lacking.

We have changed the sentence: "This study develops an approach to inferring ice content of rock glaciers by using a surface-velocity-constrained model."

4. In the following I will provide some more specific comments I have on the manuscript:

The authors must be much more careful with the wording and make sure to avoid blank statements, such as "... is important" without specifying important in what respect, and providing a reference or demonstrate the importance as par of the contribution.

Line 10: Unfortunately, the authors copy misleading statements others have made regarding using rock glaciers as freshwater resources. It is important to understand that a rock glacier is not a special type of a glacier. There is no exchange in ice, and there is no annual runoff from a glacier as we know it exists from a glacier. The hydrological behaviour of a rock glacier is completely different, and therefore it cannot be compared with a glacier when it comes to how runoff from a rock glacier should be seen as a source of freshwater. In fact, when one does calculate how much ground ice from a rock glacier is melting during a summer, even under an extremely hot summer, the authors would realise that the amount is extremely low, and in fact, often much lower than the potential evaporation. Specifically, in arid areas. In other words, water that is released from a ground ice melt is most likely not available as freshwater. The current wording is therefore creating potential anticipation that simply does not exist.

Re: We agree that rock glaciers, at the present time, do not supply freshwater through surface runoff as glaciers do. This is consistent with our assumption that ground ice stored in rock glaciers remain constant at decadal timescale (See the response to the main comment).

To avoid any misunderstanding, we have changed the sentence (line 10): "Rock glaciers contain significant amounts of ground ice and serve as potential freshwater reservoirs as mountain glaciers melt in response to climate warming in the long term."

5. Line 19: The thickness of a rock glacier is a fundamental parameter. Can the authors please clearly define what they mean by the thickness of a rock glacier? As a first step, it would be helpful to define the bottom of a rock glacier, is it defined by the base of the permafrost, the depth to bedrock, or the interface between the original terrain and the material of the rock glacier that had been transported there?

Re: In this study, we define the bottom of a rock glacier as the depth where no deformation occurs beneath. In practice, we calculate the thickness based on the empirical relationship proposed by Brenning (2005b), who defines the thickness as the depth between the surface and the base of ice-rich permafrost.

6. Line 21: Please provide clear definitions for terms such as reservoir and resource, and explain the differences in how they are used in the manuscript.

Re: We use the two words in an interchangeable way: both terms mean a place where ice is stored and potentially available for use in the future. To clear up confusion, we have replaced 'resource' with 'reservoir' throughout the text.

7. Line 21 ff. When presenting results, it is a) critical that the error range is provided, and b) that the number of significant digits reflects the accuracy. It is not appropriate to present a result to the 10th of a percent, when the error is in the 10th of percent.

Re: Concur. We have changed the flow and structure of the result presentation by firstly reporting the average bias (8%) as a reference level of uncertainty for the inferred volumetric ice fraction. For point b), we have checked the values throughout the text and made changes according to the error range. For instance, the ice fraction results are reported as 71% and 75%, with an error range of 8%.

8. Line 26: Please provide references for that statement, also, it is worth noting that this is only true for intact rock glaciers. Rock glaciers are geomorphic landforms and you can't simply ignore relist rock glaciers, for example. As mentioned above, a rock glacier is not a special type of a glacier and as such this periglacial landform must be considered differently when writing about them.

Re: We have re-written the sentence with the reference provided: "Rock glaciers are valley-floor and valley-side landforms occurred in the periglacial realm. Intact rock glaciers contain ground ice and are common in the cold mountain regions (Ballantyne, 2018)."

9. Line 28: With regard to Azócar and Brenning, 2010, I encourage the authors to carefully read the comment by Arenson and Jakob on that paper.

Re: We have carefully read the paper and the response paper followed (Arenson and Jakob, 2010; Brenning, 2010). We found it inspiring to read the in-depth discussion on the "hydrological significance of rock glaciers".

We have clarified the relevant concepts in the introduction section to avoid confusion: "Few studies have investigated the hydrological contribution of rock glaciers to surface runoffs at annual or seasonal timescale (e.g., Geiger et al., 2014; Harrington et al., 2018; Krainer and Mostler, 2002; Winkler et al., 2016), and little evidence has shown that rock glacier discharge is a prominent water source at present due to the insulation effect produced by their blocky surfaces (Duguay et al., 2015; Jones et al., 2019b; Pruessner et al., 2021). Yet, on the multi-annual to centennial and millennial timescales, we expect rock glaciers of high ice content to serve as water reservoirs long after glaciers have melted."

10. Line 29: What exactly is a "hydrological value"?

Re: Here the "hydrological value" proposed by Corte (1976) refers to rock glaciers serving as both the water storage and annual runoff sources. In this study, we do not consider the hydrological value of the latter type and assume the ground ice remains constant in the study time scale (as detailed in the response to the main comment).

11. Line 39: I'm not clear what the "ratio of importance" is. Do you simply mean the ratio? If so, then the word "important" doesn't have a meaning.

Re: We have modified the sentence: "The ratio between rock glacier ice content and that in glaciers in the region was 1:25…"

12. Line 42: See my earlier comment regarding water supply. In order to demonstrate that this statement is accurate, please provide a thermal analysis that shows how much melt you will get and then compare it with potential evaporation and infiltration.

Re: We have rewritten the paragraph and removed this inaccurate statement regarding water supplies, as this manuscript does not focus on the runoff contribution from rock glaciers.

13. Line 43: Please provide reference and definition of an ice-cored rock glacier.

Re: We realized that the terminology, ice-cored rock glacier, may cause confusion as to the classification of rock glaciers. We have removed this phrase.

14. Line 44: You write " However, there lacks modelling studies to test these postulations and to assess the likelihood of glacier- rock glacier transition and the hydrological implications of this process." I agree with this statement, but I feel that you do not keep this in mind while wording some of your text. Many of the wording is written as if it was a fact, but in essence it isn't, such as freshwater from rock glaciers.

Re: Concur. We shall be more cautious about the wording and clarify that the hydrological role of rock glaciers as a contributor to surface runoff is beyond the scope of this work.

15. Line 54: what exactly is "extremely"? Such qualifying words must not be used in a scientific publication unless clearly quantifiable.

Re: We have deleted the inappropriate modifier in this sentence.

16. Line 57: Please clarify that Arenson and Springman (you can find details in Arenson 2002), emphasize that the deformation is not related to an "average" ground ice content, because such an average does not really exist, but rock glaciers do show quite complex internal structures. The deformations are often limited to a shear horizon (Arenson et al., 2002), where the ground ice content is high. Concluding from the ground ice content in the share zone to the ground ice content of the whole rock glacier is something that has not yet been confirmed and is associated with significant errors (orders of magnitude).

Re: At line 57, we do not introduce the "average" ground ice assumption or indicate that Arenson and Springman relate rock glacier deformation to "average" ice content. Furthermore, inserting an additional clarification might break the logic flow.

To avoid possible misunderstanding, in the revised manuscript we have added necessary clarifications in Sect. 3.1 (model design and assumptions): "Here we neglected the presence of shear horizon where deformation is enhanced and ground ice content is high, as discovered from borehole investigations."

17. Line 74: Discontinuous permafrost has no altitudinal boundary. The whole concept of continuous and discontinuous permafrost, which has been developed for polar regions, should not be used in mountainous environments. That's why the term Mountain permafrost had originally been coined.

Re: We followed the convention developed in recent years that researchers started to use the concept of continuous and discontinuous to classify permafrost occurring all over the Northern Hemisphere, e.g., Obu et al. (2019). In the context of permafrost study focusing on the Tibetan Plateau, this classification scheme is widely adopted (e.g., Zhao et al., 2021; Zou et al., 2017), and altitude is commonly used to describe the distribution of the continuous and discontinuous permafrost, because it is the primary factor

controlling the environmental conditions on the Tibetan Plateau. We have changed the wording (altitudinal boundary) to "the lower limit of permafrost" to be consistent with the literature.

18. Figure 1: What is the year of the image? What was the scale used for mapping?

Re: The background image was taken in the year of 2019. We have added the information in the caption. The scalebar is plotted at the top left corner of Fig. 1.

19. Line 91: Are you using Ascending and/or descending imagery?

Re: Most are ascending imagery. Information of the satellite data is shown in Table 1 (at Line 115).

20. Line 93: What exactly is high? Can you be quantitative as this is another relative term.

Re: We set coherence = 0.3 as the threshold, as detailed at line 108.

21. Line 99: Please provide more details on the analysis methodology used for the InSAR assessment. E.g. did you use PS or any other method? There are many aspects unclear on the InSAR assessment.

Re: We used the Differential InSAR (DInSAR) method in this study.

22. Line 100: relative term, what do you mean by "near"?

Re: In practice, the selected reference points are located within 300 m from the landform.

23. Line 101: what was the landform coverage? How much topographic shade did you experience for the landforms?

Re: The areal extent of the five rock glaciers ranges from ~0.07 $km^2$ to ~0.2 $km^2$, as detailed in Table 4. We do not experience the shade issue for our study objectives, because the slopes of the landforms are gentle ($< 20°$), smaller than the incidence angles of the SAR images we used (38.7° and 36.3°).

24. Line 105: I assume that this was done using SRTM topography and not by combining ascending and descending stacks. This can result in significant errors in deformation due to the significant differences in the resolution between SRTM and InSAR imagery. How much are the errors in your evaluation?

Re: Yes, we reprojected the results using the SRTM DEM. Considering the uncertainties introduced both from the DEM and interferometric quality (assessed by coherence), for one pixel, the velocity error mostly ranges from 1 cm/yr to 10 cm/yr, and the relative uncertainty is between 5% and 25%. The absolute and relative errors are reduced to <1 cm/yr and <5% when considering the spatial mean velocity of all pixels covering one rock glacier. We have reported the errors in the revised manuscript.

25. Line 108: you mean less than half? Is this still representative? Can you quantify that using only 40% of the area is representative for the assessment presented? Also, you probably are biased towards the flatter sections of a rock glacier where there is less topography, but where you likely would have more compressive flow.

Re: Here the value of 40% serves only as a threshold for selecting interferograms of good measurement quality from many InSAR observations. The interferograms adopted after applying the criteria presented here have a mean spatial coverage of 61%. From the velocity distribution maps (Fig. 6), we do not observe obvious biases towards flatter regions, because surface topography does not necessarily

cause decorrelation in the interferometric processing, though it contributes to the uncertainty of the results (as detailed in the former response to comment #24).

26. How did you address differences between compressive and extensive flow sections in rock glaciers? Are you implying that the ice content in the compressive areas are representative?

Re: The comment seems to be follow-up questions to the previous one (comment #25). At line 108, we did not separate sections of a rock glacier or differentiate compressive and extensive flow patterns.

27. Line 109: The values presented are averaged over the Dec. 2007 to Feb 2020 stack, is that correct? And I assume it is based on the 40% area coverage.

Re: No, the value is not a temporal average, but the spatial average of all pixels with valid measurements on each rock glacier. The 40% is the threshold value.

28. Table 1: How come you only have one interferogram? What is your level of confidence to use just that one interferogram in your assessment?

Re: By using the conventional DInSAR methodology (unlike the SBAS method for instance), only one interferogram can be used to obtain the displacement occurred during the period between the two SAR acquisitions.

29. Line 144: 1. you assume that the base of the rock glacier equals the base of the permafrost, correct? 2. You assume homogeneous conditions, which I haven't seen in any rock glacier, i.e. this is a huge simplification. I'm not saying there is no value in doing this, but you must be aware of what the consequences of such a simplification are when you draw your conclusions.

Re: 1. No, we define the base of a rock glacier as the depth where no deformation occurs beneath. We divided the rock glacier body into the active layer and the permafrost core. Beneath the bottom of the landform, permafrost may still be present but beyond our research scope.

2. Concur, rock glaciers do not develop a homogeneous internal structure in the nature, and that assuming a homogeneous structure is a huge simplification made in our model setting. The rationale behind this simplification (and many others made in this work) is that we are aware of the reality that knowledge of rock glaciers obtained from direct observations is valuable yet limited, so that at the current stage, we aim to take advantage of the existing data to explore some empirical relationships. Such practice requires us to be rigorous and careful about the level of uncertainties in our results and not oversell our findings. The importance of gathering data based on direct observations should be emphasized. In the future, the accuracy of the modelling results will be improved with more ground truth data obtained and used for method validation.

30. Line 145: Talus derived rock glacier show very variable thickness. Potential generalization that may lead to misleading results / conclusions.

Re: We have added discussion in Sect. 5.3 of the revised manuscript, when discussing the uncertainty introduced by the thickness derivation (at line. 430): "In addition, rock glaciers, especially the talus-derived ones, tend to develop very variable thicknesses across the landform, the distribution of which cannot be inferred using the existing empirical approaches."

31. Line 146: You also assume constant temperature conditions within the permafrost body, which is often not the case. Again, a rock glacier is not a special type of a glacier and can't be compared to a temperate glacier.

Re: We agree that constant temperature conditions are rare in reality (one example is reported in Monnier and Kinnard (2013)), but a simplified scenario is assumed in this work.

32. Line 184: I am very surprised that the single most important parameter, the temperature, is simply ignored.

Re: We discussed this aspect in Sect. 5.2.2, where we first reviewed the effect of ground temperature in controlling the creep parameter, then we justified our method of assuming a homogeneous warm profile and using the effective viscosity to represent the effects of many factors, finally we explained why this ground temperature condition is likely to be realistic in our study area.

In the revised manuscript, we moved these justifications to the methodology part (Sect. 3.1).

33. Line 187: What is the error range? Geophysics w/o calibration may have significant errors.

Re: According to Monnier and Kinnard (2015), the mean ice fraction is 0.66±0.101. The magnitude of error is ~10%.

34. Line 220: This correlation should simply not be used (See Arenson and Jakob, 2010). Using such a simplified correlation does not account for the complex geomorphic background of why a rock glacier exists. Hence, utilizing a glaciological, mass balance inspired approach to describe a periglacial, topo-geologically driven process will not provide accurate results.

Re: We acknowledge it is another simplification made to the complex structure of rock glaciers. We discussed this aspect in Sect. 5.3 of the revised manuscript. Justifications of the simplified assumptions made in this work are summarized in our response to comment #29-2.

35. Line 224: SRTM resolution is not 30 m, but varies geographically as it is in arc degrees.

Re: We described the SRTM DEM resolution at Line 95: "We estimated and removed the topographic phase with the 1-arcsec digital elevation models (DEM) produced by the Shuttle Radar Topography Mission (SRTM) (spatial resolution ~30 m)." We have modified the wording at Line 224: "…a spatial resolution of ~30 m."

36. Table 4: These active layer thicknesses are extremely thin. Please look at some of the rock glacier active layer thicknesses in the Alps (e.g. PERMOS reports) where you will find that rock glacier active layer thicknesses are often several meters thick. Hence the major thermal protection and the lack of contribution to any runoff, even as the permafrost degrades. The energy available for ground ice thaw below an active layer thickness of several meters, is low.

Re: We derived the active layer thickness from the ESA CCI Product (at line 253), and yes, the values seem to be very small for a real rock glacier. Another rock glacier on the Tibetan Plateau, where we have conducted in-situ measurements, develops an active layer of ~2 m thick. We agree that underestimating the active layer thickness would lead to significant errors when considering the thermal regime of a rock glacier.

However, in this work, we do not take the thermal evolution into account or estimate future runoff contribution as ground ice thaws (detailed in our response to the main comment). In addition, the

sensitivity test shows that the active layer thickness exerts the least effect on the model result (Fig. 10), which justifies our use of the ESA CCI data provided that the direct measurements are lacking.

37. Line 358: Rock glaciers do not show a uniform creep. Most rock glaciers have an area that is faster and another that is slower. For example, large rock glacier may no longer advance because the lower part lost too much ice to allow creep. However, the upper part is still creeping, Your approach will completely overestimate the ice content as it does not take the actual rock glacier kinematics into consideration.

Re: We agree that rock glaciers do not creep uniformly at the landform scale, so that instead of considering the landform as a whole, "we defined and outlined the coherently moving part of the landform by considering the time series of downslope velocity of each pixel acquired during all the observational periods. If the InSAR-measured velocity is higher than 5 cm yr$^{-1}$ in more than half of the periods at a given pixel, it was included into the coherently moving part of the landform." (At line 110–112.) In the given example, the lower part of a rock glacier which no longer moves is not counted in our estimates.

38. Line 367: importance relative to what?

Re: The importance is relative to a larger region: the entire Himalayas. Here we compared the ratio (ice storage in rock glaciers vs. in glaciers, 1:17) over the Nepalese Himalaya with the ratio reported in previous research focusing on the entire Himalayas, where the Nepalese Himalaya is a sub-region.

We have changed the wording: "…than across the entire Himalayas (1:24)."

39. Line 378: You state "This premise indicates that our method is applicable to rock glaciers currently moving at a relatively stable rate." For one, based on data from the Alps, we know that this is likely not the case, and more importantly, you use data from rock glaciers that do not show stable deformation to develop your model, which should then be only valid for stable deformation? This does not sound logical to me.

Re: By stating the premise of "moving at a relatively stable rate", we aim to exclude the destabilized rock glaciers from the scope of method application. In the strict sense, an object in the steady-state creep exhibits constant strain rate, which is not the case for any rock glaciers. Many rock glaciers are reported to experience motion fluctuations at the inter-annual scale, including the ones we used for developing the model. To simulate deformation behavior of these non-destabilized rock glaciers, Glen's flow law is still widely accepted. However, in the revised manuscript, we have excluded the data of Gruben rock glacier from the validation set, because it is losing internal ice and changing its morphology rapidly (Gärtner-Roer et al., 2021; R. Delaloye, personal communication, July 21, 2021), which does not align with our model design.

40. Line 385: Call the "clean" (what ever that actually means) as an uncovered or covered glacier. Or simply call it glacier because rock glaciers are not special glaciers, as I've been mentioning severel times already.

Re: Concur, rock glaciers are not glaciers. We have removed the misleading word "clean" from the sentence.

41. Line 419: You are citing Cicoira et al. (2020) to support your statement. Are you sure, since the publication of Cicoira et al. (2020) had a completely different objective and it seems to me that your referencing is taking out of the appropriate context.

Re: The work presented by Cicoira et al. (2020) indeed has a different research objective, yet shares the same technical issue of expressing the rock glacier creep without known internal structure data, and in their work, the shear horizon cannot be considered either. We cited this paper to show this common issue when studying rock glacier kinematics over a regional extent (in contrast to the case studies with sufficient data). To avoid inappropriate comparison between the two pieces of work, we have removed this citation.

**42. Line 424: I suggest that you read Arenson and Jakob (2010) and revisit your statement.**

Re: We have read through the discussion and reply articles by Arenson and Jakob (2010) and Brenning (2010), respectively. Regarding the empirical derivation of rock glacier thickness presented by Brenning (2005a), we think it is statistically valid but likely to have limited level of accuracy. In the revised manuscript, we have rewritten the discussion on the uncertainty in deriving rock glacier thickness (Sect. 5.3).

**43. Line 426: I am not at all surprised by the large bias that you found, however, I do not see this bias be further developed, for example using error propagation theories, to illustrate what that means for your end result.**

Re: The uncertainty introduced by the error of thickness can be represented by the model sensitivity to the area variable, as we derived the thickness based on an area–thickness relationship. Fig. 10 shows that the error range caused by the varying area parameter is within the 5% range, given that the area-thickness relationship is applicable to the landform.

When this empirical derivation is invalid, the propagated error can be significant but difficult to quantify due to the limited available data. Therefore, in Sect. 5.2.4, we only acknowledge that the uncertainty introduced by thickness derivation when applied to rock glaciers without known information of structure cannot be eliminated with the existing empirical methods.

**44. Table 7: What is Tref?**

Re: We missed the symbol in the table caption and have added at line 432: "…and the corresponding bias relative to in situ measured thickness ($T_{ref}$) (Barsch et al., 1979; Cicoira et al., 2019a; Arenson et al., 2002; Hoelzle et al., 1998)." We have modified and moved the table to the supplement materials.

**45. Table &: How confident are you that these 5 (!) rock glaciers, which all have very specific features, are representative so that a correlation, such as the one you present, can be developed and reasonably be applied for hundreds of rock glaciers in very different settings?**

Re: Sect. 5.2.4 does not aim to illustrate the wide applicability of the thickness–area relationship we adopted. Instead, we hope to point out that the reliability of thickness derivation remains to be an issue at the current stage.

We have rewritten the discussion on the rock glacier thickness derivation method (Sect. 5.3).

**46. Line 459: Based on my review I do not support this statement and it is my very strong impression that this approach is not yet ready and specifically I would not call the uncertainties "well-quantified". In fact, the uncertainties are unknown.**

Re: We agree that this is an inappropriate statement. We have removed this sentence. We've also assigned an uncertainty level according to the average bias derived from the model validation (see also in the response to comment #7).

47. Line 460: I completely agree with the final statement and encourage the authors to put their effort in getting more field data so that can provide a better estimate for rock glacier thicknesses.

Re: Concur. Field data are fundamental for deriving generalized empirical rules, we would like to contribute to the data gathering and improve the approach presented in this work accordingly.

48. Line 468: The authors indicate that they are measuring active layer from remote sensing. First, they have not discussed this aspect in the paper, which means that this should not just pop-up in the conclusion, and secondly, I am not aware of a method on how to measure rock glacier active layer thicknesses from space. Or maybe the authors mean geophysics, which has its own challenges for block rock glaciers.

Re: We introduced the ALT data source in the methodology section at line 253: "Active layer thickness was determined as the mean value over the extent of each rock glacier during 2006–2017 from the European Space Agency Permafrost Climate Change Initiative Product (ESA CCI)." The ALT dataset is produced based on remotely sensed datasets of Land Surface Temperature (LST), Snow Water Equivalent (SWE) and landcover, so that we refer to it as a remote sensing product in the manuscript.

49. Line 472: Level of accuracy implied is unrealistic.

Re: We have modified the structure of the result presentation (see also the response to comment #7).

50. In summary, this manuscript is not ready for publication, and I strongly encourage the authors to re-evaluate their scientific basis and if there is even any merit in the approach presented considering the significant uncertainties that exists because of the assumptions used.

Re: Again, we thank Dr. Arenson for his informative and inspiring comments, especially the clarification of rock glacier hydrological value, the reliability of rock glacier thickness derivation, and the appropriate format of result presentation. We have addressed these aspects in the response letter and made changes accordingly to the manuscript.

Moreover, we agree with Dr. Arenson's emphasis on the field data of rock glaciers, which is essential for improving the accuracy of modelling approaches. In the current manuscript, we aim to take advantage of the existing observational data and build a framework for inferring ice content with remote sensing-based input. In the response letter, we attempt to provide more detailed justifications to the necessary assumptions made in this work and avoid possible exaggerating statements. To further improve the performance of the approach, as stated in the final paragraph of the discussion section, "…more data obtained from field and geophysical investigations, especially detailed data of rock glacier composition, can be integrated in the future to calibrate and validate the empirical rheological model. More reliable methods for estimating rock glacier thickness will also improve the accuracy of the modelling results." And it is a research path we are following at present.

In the revised manuscript, we have summarized the limitations and merits in the discussion section and concluded: "In summary, the lack of ground truth data essentially hinders our approach from achieving high-level accuracy in quantifying ice content of rock glaciers. Nonetheless, the proposed model makes a first attempt to build a framework for inferring ice content with remote sensing-based input by taking advantage of the existing observational data. With the likely emergence of more data to be integrated

for model calibration and validation, it forms a promising approach to improve the accuracy of modelling results and application to mountain permafrost regions where rock glaciers are widespread for preliminary water storage evaluation."

**References**

Arenson, L. U., and Jakob, M. (2010). The significance of rock glaciers in the dry Andes – A discussion of Azócar and Brenning (2010) and Brenning and Azócar (2010). *Permafrost and Periglacial Processes, 21*(3), 282-285. https://doi.org/10.1002/ppp.693

Ballantyne, C. K. (2018). *Periglacial geomorphology*. John Wiley & Sons.

Brenning, A. (2005a). *Climatic and geomorphological controls of rock glaciers in the Andes of Central Chile* Humboldt-Universität zu Berlin, Mathematisch-Naturwissenschaftliche Fakultät II].

Brenning, A. (2005b). Geomorphological, hydrological and climatic significance of rock glaciers in the Andes of Central Chile (33-35 degrees S). *Permafrost and Periglacial Processes, 16*(3), 231-240. https://doi.org/10.1002/ppp.528

Brenning, A. (2010). The significance of rock glaciers in the dry Andes – reply to L. Arenson and M. Jakob. *Permafrost and Periglacial Processes, 21*(3), 286-288.

Cicoira, A., Marcer, M., Gärtner-Roer, I., Bodin, X., Arenson, L. U., and Vieli, A. (2020). A general theory of rock glacier creep based on in-situ and remote sensing observations. *Permafrost and Periglacial Processes*. https://doi.org/10.1002/ppp.2090

Corte, A. (1976). The Hydrological Significance of Rock Glaciers. *Journal of Glaciology, 17*(75), 157-158. https://doi.org/10.3189/s0022143000030859

Duguay, M. A., Edmunds, A., Arenson, L. U., and Wainstein, P. A. (2015). Quantifying the significance of the hydrological contribution of a rock glacier–A review. *GeoQuébec 2015*.

Gärtner-Roer, I., Brunner, N., Delaloye, R., Haeberli, W., Kääb, A., and Thee, P. (2021). Glacier-permafrost relations in a high-mountain environment: 5 decades of kinematic monitoring at the Gruben site, Swiss Alps. *The Cryosphere Discuss., 2021*, 1-30. https://doi.org/10.5194/tc-2021-208

Geiger, S. T., Daniels, J. M., Miller, S. N., and Nicholas, J. W. (2014). Influence of Rock Glaciers on Stream Hydrology in the La Sal Mountains, Utah. *Arctic, Antarctic, and Alpine Research, 46*(3), 645-658. https://doi.org/10.1657/1938-4246-46.3.645

Harrington, J. S., Mozil, A., Hayashi, M., and Bentley, L. R. (2018). Groundwater flow and storage processes in an inactive rock glacier. *Hydrological Processes, 32*(20), 3070-3088. https://doi.org/10.1002/hyp.13248

Jones, D. B., Harrison, S., Anderson, K., and Whalley, W. B. (2019). Rock glaciers and mountain hydrology: A review. *Earth-Science Reviews, 193*, 66-90. https://doi.org/10.1016/j.earscirev.2019.04.001

Krainer, K., and Mostler, W. (2002). Hydrology of Active Rock Glaciers: Examples from the Austrian Alps. *Arctic, Antarctic, and Alpine Research, 34*(2), 142-149. https://doi.org/10.1080/15230430.2002.12003478

Monnier, S., and Kinnard, C. (2013). Internal structure and composition of a rock glacier in the Andes (upper Choapa valley, Chile) using borehole information and ground-penetrating radar. *Annals of Glaciology, 54*(64), 61-72. https://doi.org/10.3189/2013AoG64A107

Monnier, S., and Kinnard, C. (2015). Internal structure and composition of a rock glacier in the Dry Andes, Inferred from ground-penetrating radar data and its artefacts. *Permafrost and Periglacial Processes, 26*(4), 335-346. https://doi.org/10.1002/ppp.1846

Obu, J., Westermann, S., Bartsch, A., Berdnikov, N., Christiansen, H. H., Dashtseren, A., Delaloye, R., Elberling, B., Etzelmüller, B., Kholodov, A., Khomutov, A., Kääb, A., Leibman, M. O., Lewkowicz, A. G., Panda, S. K., Romanovsky, V., Way, R. G., Westergaard-Nielsen, A., Wu, T., Yamkhin, J., and Zou, D. (2019). Northern Hemisphere permafrost map based on TTOP modelling for 2000–2016 at 1 km2 scale. *Earth-Science Reviews, 193*, 299-316. https://doi.org/10.1016/j.earscirev.2019.04.023

Pruessner, L., Huss, M., Phillips, M., and Farinotti, D. (2021). A Framework for Modeling Rock Glaciers and Permafrost at the Basin-Scale in High Alpine Catchments. *Journal of Advances in Modeling Earth Systems, 13*(4), e2020MS002361.

Winkler, G., Wagner, T., Pauritsch, M., Birk, S., Kellerer-Pirklbauer, A., Benischke, R., Leis, A., Morawetz, R., Schreilechner, M. G., and Hergarten, S. (2016). Identification and assessment of groundwater flow and storage components of the relict Schöneben Rock Glacier, Niedere Tauern Range, Eastern Alps (Austria). *Hydrogeology Journal, 24*(4), 937-953. https://doi.org/10.1007/s10040-015-1348-9

Zhao, L., Zou, D., Hu, G., Wu, T., Du, E., Liu, G., Xiao, Y., Li, R., Pang, Q., Qiao, Y., Wu, X., Sun, Z., Xing, Z., Sheng, Y., Zhao, Y., Shi, J., Xie, C., Wang, L., Wang, C., and Cheng, G. (2021). A synthesis dataset of permafrost thermal state for the Qinghai–Tibet (Xizang) Plateau, China. *Earth Syst. Sci. Data, 13*(8), 4207-4218. https://doi.org/10.5194/essd-13-4207-2021

Zou, D., Zhao, L., Sheng, Y., Chen, J., Hu, G., Wu, T., Wu, J., Xie, C., Wu, X., Pang, Q., Wang, W., Du, E., Li, W., Liu, G., Li, J., Qin, Y., Qiao, Y., Wang, Z., Shi, J., and Cheng, G. (2017). A new map of permafrost distribution on the Tibetan Plateau. *The Cryosphere, 11*(6), 2527-2542. https://doi.org/10.5194/tc-11-2527-2017

---

## Referee Report (RR1)

Third review to the article from Hu et al. 'Modelling rock glacier ice content based on InSAR-derived velocity, Khumbu and Lhotse Valleys, Nepal':

I would like again to acknowledge the extensive work performed by the authors to address my comments from my previous review. Although I do agree with some critical comments from the other reviewers regarding the methodology, I believe the current version is now acceptable for publication after minor adjustments. The research raised several questions about the feasibility/relevance of coupling remote sensing observation and modelling, quantifying the ice content based on this approach and extrapolating the results to an entire region. However, these questions are now well discussed, and the limitations are acknowledged. Independently of if we agree or not with the assumptions, I believe the procedure and resulting outputs are now clearly described and therefore worth publication. Generating debates in the community is also what open access dissemination is about.

I have focused the work for this third review on the main parts that I commented in my second review, suggesting some minor corrections to facilitate the understanding of the content. Note that the line numbers refer to the version without track changes.

l.232-234: "… we present our method to measure surface velocities of rock glaciers with InSAR for constraining the model (Sect. 3.5.1) and use complementary remote sensing products to derive geometric and structural parameters (Sect 3.5.2)."

l.248-249: "multi-looking operation and adaptive Goldstein filter (8x8 pixels) were applied using the open-source software…"

l.250: "The final georeferenced interferograms have a ground resolution…"

l.252: "Ice-debris landforms"? Is there a reason not using "rock glaciers" here? If it also considers debris-covered glaciers, just say it so.

l.261: "For each pixel, we found the velocity error is < 10 cm/yr". Rather used the way you explained it your response to my questions, it is much clearer: "For each interferogram, we quantified the uncertainty at the pixel-level. Among all the high coherent pixels, the largest uncertainty is 9.8 cm/yr. The velocity error is therefore considered as < 10 cm/yr."

l.262-265: Still unclear to me if the criteria are applied to discard interferograms, entire or part of the landforms. Based on what you explained in your response, I would suggest writing: "… we selected the interferograms and documented rock glacier parts meeting the following criteria… (1) only pixels showing acceptable coherence (> 0.3) are kept; (2) the coherent pixels must cover more than 40% of the landform surfaces; (3) the mean velocity must be larger than 5 cm/yr (Wang et al., 2017). We set this empirical threshold considering the typical noise level from atmospheric delays (5 cm/yr). The interferograms and landforms that do not meet these criteria were discarded." If I misunderstood: please adjust the content and make it clear.

l.271: "After the procedure described in Step 2, for each selected landform, …"

l.274-276: "… in more than half of the interferograms, the pixel was included in the coherently moving part… Otherwise, the pixel is discarded, i.e. not included in the coherently moving part. The area is considered as inactive or in a transitional kinematic status."

l.279: Again, it is better explained in the response to my question: "the mean velocity error is the square root of the quadratic sum of all the velocity errors, which is limited to < 1 cm/yr". You could btw consider referring here to your github codes (basically referring to the code and data availability section) to make it simple to look for answers about how it has been calculated.

l.280: "… the range of the spatially averaged velocities within the coherently moving parts… By doing so, isolated patterns are neglected assuming that they may be related to short-term kinematic fluctuations, not representative of the multi-annual kinematic behaviour of the whole landform."

l.291: "…the abnormal value in 2015 has been removed from the range"

l.350: "… (Sect 4.4.1) and present the modelled ice content of the five rock glaciers…"

l.418: "consistent with the fact that rock glaciers are currently not a major contribution to surface runoff in the study area"

l.450: "the uncertainty in deriving rock glacier thickness remains ambiguous" could be rephrased. I don't think "ambiguous uncertainty" means anything.

l.455: One thing that is missing as limitation in this part: when considering your definition of coherently moving area, you discard quite big parts of the morphologically delineated landforms, which raises the question: how representative is the resulting InSAR average when it is based on data covering less than a quarter the rock glacier surface (for ex. Fig.8b)? As you acknowledge in your answer to my previous questions, it can be both due to low-coherence (potentially due to high-velocity) or low-velocity. It basically means that you have a big uncertainty in both directions: the results may be underestimated or overestimated. Good to mention the problem, I think.

l.458: Avoid using "stable" / "stability" when speaking about moving landforms. "at a relatively constant rate."

l.461: Same here "the stability of the motion". Maybe "to consider an average rate and avoid misleading conclusions based on unrepresentative short-term patterns".

l.463: "… or behave differently from the coherently moving parts"

l.467: "Third, rock glaciers affected by significant subsidence (instead or in addition to downslope creep) cannot be…"

l.471: "Finally, …"

l.496: "The likely emergence of… will likely allow for improving the accuracy of the approach". "We expect the improved model can be valuably applied to…"

l.505: "(2) Mean downslope velocities in the coherently moving part of the rock glaciers in…" I think it is important to specify it here – that is the mean of what you define as coherent, not the entire morphologically-delineated rock glaciers.

l.507: "remained constant" – wise to do a "find&replace" function for the entire document cause using "stable" is quite misleading.

l.516: Maybe "confirms" instead of "highlights"?

---

## Referee Report (RR2)

**Modelling rock glacier ice content based on InSAR-derived velocity, Khumbu and Lhotse Valleys, Nepal**

**By Hu Yan et al.**

*General remarks*

The authors present a novel method to estimate the ice volume in rock glaciers based on a modified ice flow model and InSAR derived surface velocity. The model is calibrated using literature data from a rock glacier in the Andes, validated at rock glaciers in the Swiss Alps where detailed information is available and then applied to five rock glaciers in Khumbu Himal. Finally the authors upscale their results to the whole Nepalese Himalaya based on an existing inventory using a scaling relationship. The topic is of high importance as the ice contained in rock glacier could potentially be of hydrological importance, measurements can only be done on a very limited number of rock glaciers and therefore a modelling approach could provide valuable information.

This paper has gone through few round of reviews and even though all reviewers are in line that work is in general if high interest there seems still to be some concerns which could not be fully addressed. I won't recall the available reviews but provide an independent opinion based on the current version but acknowledging that the manuscript has clearly improved.

*General comments:*

- The main focus of the study is to estimate the ice content of few rock glaciers in Khumbu Himal using a novel methodology. I know that the authors cannot change now this setup, but it is in general questionable to develop a new method based on one rock glacier (which is quite long and narrow) calibrate it on others and then apply is to rock glaciers with different characteristics in a region with different climate and topographic settings. Hence, it remains unclear how well this information can be transferred. The authors must more convincingly show this, e.g. by providing more detailed information about topographic, climatic and ground temperature conditions of the different regions. I understand that the authors maim focus is the Himalaya, but why not first develop, calibrate and validate the model on rock glaciers and regions in the Alps or Andes where much more information and also in-situ measurement of rock glaciers are available? If the authors decide to keep this set-up then at least a rational for choosing these rock glaciers need to be given. Las Liebres rock glacier is measured by quite thoroughly by GPR. However, to get better information about the ice content it is in general recommended to combine different geophysical techniques.

- The authors should also more clearly present how they define rock glaciers in their study. When the authors first present a definition of rock glaciers it is very general without mentioning of permafrost. But at line 100ff at the beginning of the model they refer to ice-rich permafrost. The also mention the transition to rock glaciers with glacier melt (L40). Is this also true for glacier flowing into regions where permafrost is unlikely? This is important to consider are the presence of permafrost influences melt and the ice flow. In this sense more information about the ground thermal conditions and possible permafrost presence (e.g. by considering available climate measurements and permafrost modelling results) is required for all considered regions.

- The flow modelling seems to be suitable as applied in a similar way by different other studies. One issue as also mentioned in the text is the seasonal variation occurring mainly at

the shear horizon which is not captured in the model. The seasonal is according to the available measurements 60-90% of the surface velocity. The authors assume that they neglect the short term variations by taking "the range of the spatial mean velocities of the coherently moving parts". This needs to be more convincingly shown. E.g. Can't the authors generate a time series of the velocity and show them?

- In case I understood correctly they transfer the derived velocity for the part they obtained values to the whole landform (L280ff). This is questionable as variable parts of the rock glaciers might have different ice contents and for some rock glaciers they get results for clearly less than 50% of the rock glacier. Moreover, the part of which the authors obtain suitable results varies strongly, e.g. it is the upper part of Kala-Patar, only one side Kongma and the lower part for Tobuche. The only rock glacier for which the approach is reasonable is Nuptse rock glacier.

- This is also true for the scaling. The authors use a formula based on area which was developed for the Andes by Brenning (2005) and apply it without adjustments to the rock glaciers in the study region and the Nepal Himalaya. This is equally questionable as it is well known that scaling parameters vary and should be calibrated for the specific regions. Also the characteristics of the identified rock glaciers varies clearly. While the Nuptse Rock Glacier has a clearly identifiable tongue Kala-Patar not and has a depression with small lakes (see Figure 1). It is hence likely that the different parts pf the rock glaciers have different ice contents. I suggest that the authors analyse the topography of the rock glaciers and adjust the scaling accordingly or use other suggested approaches.

[Figure]

[Figure]

Figure 1: Nuptse and Kala-Patar rock glaciers (Image source: Pleiades from Google Earth)

- Another issue is the rock glacier delineation. I know this is quite difficult and also subjective (see e.g. the cited study by Brardionini et al. ) and the delineation is fine for a regional study, but for this localised rock glaciers the authors should make more effort to provide the most precise outlines possible.

- What about the terminus of Khumbu glacier? Knight, Harisson and Jonas (2019) argue it might be a future rock glacier. Even though I am not fully in line with the argumentation it would be highly valuable to also model the ice content and have some comparison to the rock glaciers (the authors state that they also calculated the velocity of debris-covered glaciers so it can be easily done). The ice core taken close to the more rock glacier part as identified by Knight et al. (2019) might provide some valuable data (Miles et al. (2021).

- This value could then also nicely compared to the modelled ice content of the identified rock glaciers.

- It is not fully clear to me how the uncertainty of the final result was calculated. The uncertainty ranges are much too low considering all the uncertainties. Provide an own section for clarification.

- These issues make the regional extrapolation highly uncertain and there is no advance in knowledge compared to the first rough estimate presented by Jones et al. (2018) as cited in the study. If the authors really want to extrapolate they should do so by applying their velocity-based approach to the large region or a subset (e.g. the whole Khumbu Himal) and then compare to the available data.

Several of these issues are discussed in the discussion section acknowledging the uncertainty. This is well appreciated, but does not really make the results more accurate. At minimum I ask the authors to provide the most important information affecting the accuracy in the methods section to that the reader clearly knows limitation before knowing the results and can then better interpret them.

**Specific comments**

Abstract, general comment: The abstract is missing the information about how the volume the rock glacier and the of ice of the water equivalent was estimated. The volume of the rock glacier is estimated on an existing scaling approach.

L22/23: If the model is easily applicable why didn't the authors do so for upscaling? An important prerequisite is also a rock glacier inventory.

L25ff: The authors need to extend their definition of rock glaciers, state the relation to permafrost and also move the information given in L100 to here.

L28: The authors might want to include one or two more citations about other mountain regions where rock glaciers store a significant amount of water.

L31ff: Can the authors please a bit more specific about debris-covered glacier to rock glacier transition. What about the distal part of Khumbu (Knight et al. 2019)? Please also cite a reference from another research group not only one from the authors.

L35f: This is in theory correct, but it is well known that the debris-covered glaciers in the Himalaya lost at least as much mass as debris-free glaciers (due to manyfold reasons incl. reduced ice flux, supraglacial ponds, ice cliffs etc.). Hence, this argument is not valid. Please revise. My general recommendation is to omit this entire paragraph and really focus on rock glaciers and not debris-covered glaciers.

L41: I agree that the ratio can be higher if the glaciers melt, but disagree with statement with the transition to rock glaciers as this ice already existed and was considered in models.

L49ff: These are way to many citations in a row. Please be more specific about the cited papers or remove some.

L56: Would be good to mention all relevant factors controlling the movement and then specific the most important ones and then provide more details about the ice content.

L70: Please indicate in Fig. 1 the ones which transition to rock glaciers according to the cited references.

L74f: The authors should also consider Fukui et al. (2007)

Fig. 1: The digitisation of the debris-covered glaciers is quite poor. Even though not the focus of the study, this needs improvement.

L126f: Please be more specific about the permafrost core, in particular about the water occurrence.

L130: Please check the statement about the high ice content. The shear horizon is mentioned in the cited studies, but no information about the ice content (but maybe I have overseen this). The shear horizon is nicely presented by Cicoira et al. (2021) and they also state here that the ice content is lower than in the ice-rich core.

L133: The seasonal variation are first presented by Wirz et al. (2016).

Fig. 3: The figure is quite similar to the one by Monnier & Kinnard (2016) apart from the fact that the deformation at the front is not shown. I suggest to show also the deformation and also include the shear horizon. I would then also refer to the reference (but add adjusted or similar) as the idea seems to be taken from it.

L205f: The authors apply the empirical formula established by Brenning (2005). See my general comment about applying the formula to other regions. How do the derived volumes of the Las Liebres rock glacier and the three rock glaciers in the Swiss Alps compare to measured volumes? Provide this information here.

Table 1: Where do the values given in the table come from? I recommend to show only three decimals for the area. The delineation is not so precise.

L236: Consider to cite also more classical papers which introduced the approach, e.g. Strozzi et al. (2004).

L261: I would rather call it uncertainty as no validation measurements are available.

L298: Provide the information about the source of the ice content of the glaciers.

Fig. 5a: Should be Murtèl-Corvatsch (or only Murtèl)

Model-sensitivity: I have no time to think through the model sensitivity in detail. As the model seems to be quite incentive to the different input parameters the authors want to provide more details about the possible reasons.

L349f: This and similar kind of sentences are from my point of view not needed as this is evident from the headings.

L353: The information about the glacier velocity should either be better integrated in the study and compared to the rock glaciers or omitted. I suggest the latter as it distracts from the general topic of this study.

Fig. 7: Indicate in this figure which velocity was used to calculate the ice content.

L.378: What is the area of the coherently moving parts and what is the water equivalent (w.e.) of the moving part?

L381: Provide the information how much water is stored in all the rock glaciers was derived.

Table 4 and 5: I suggest to combine and also include the area and the w.e. of the coherently moving parts.

Section 4.5: As written above this is a very rough approximation. Simply extrapolation the values from 5 rock glaciers does not really add to our knowledge considering the uncertainty. If the authors aim to upscale then using their approach and including the velocity information.

Section 5 Discussion: As written the discussion about the uncertainties is appreciated, but also highlights the large uncertainties and hence sheds many questions on the approach. I recommend the authors also to highlight the advances of the presented approach in relation to the literature and, hence, better justify their presented approach. A relevant paper to consider here or maybe already when presented the method is Hartl et al. (2016)

L403ff: As written above: This is basically a listing of the headings of the subchapters and can therefore be removed.

L410: Suggest to wright "reader" instead of "user"

L470ff: Either provide more detailed information about the investigated rock glaciers (the Tien Shan is larger) and an accessible reference or omit this paragraph.

Overall, quire difficult to judge the overall value of the study. One hand the study is highly interesting and important on other hand contains many shortcomings leading to highly uncertain results. My suggestion would be to split the paper into two: One which focusses on model development (in a region with suitable in-situ measurements) and one which applies an improved method to the larger region.

*Additional references not cited in the manuscript:*

Fukui, K., Fujii, Y., Ageta, Y., Asahi, K., 2007. Changes in the lower limit of mountain permafrost between 1973 and 2004 in the Khumbu Himal, the Nepal Himalayas. Global Planet. Change 55, 251–256.

Hartl, L., Fischer, A., Klug, C., Nicholson, L., 2016. Can a simple numerical model help to fine-tune the analysis of ground-penetrating radar data? Hochebenkar rock glacier as acase study. Arct. Antarct. Alp. Res. 48, 377–393. https://doi.org/10.1657/AAAR0014-081.

Miles, K.E., Hubbard, B., Miles, E.S., Quincey, D.J., Rowan, A.V., Kirkbride, M., Hornsey, J., 2021. Continuous borehole optical televiewing reveals variable englacial debris concentrations at Khumbu Glacier, Nepal. Communications Earth & Environment 2, 12. https://doi.org/10.1038/s43247-020-00070-x.

Strozzi, T., Kääb, A., Frauenfelder, R., 2004. Detecting and quantifying mountain permafrost creep from in situ inventory, space-borne radar interferometry and airborne digital photogrammetry. Int. J. Remote Sens. 25, 2919–2931.

Wirz, V., Gruber, S., Purves, R.S., Beutel, J., Gärtner-Roer, I., Gubler, S., Vieli, A., 2016. Short-term velocity variations at three rock glaciers and their relationship with meteorological conditions. Earth Surf. Dynam. 4, 103–123. https://doi.org/10.5194/esurf-4-103-2016.

---

## Editor Decision (ED1)

[revised manuscript text omitted]

Table S1, S2, and S3; Figures S1, S2, S3, S4, S5, S6, S7, and S8

[Figure]

**Figure S1: Relationship between debris fraction ($\theta_{d,core}$) and water fraction ($\theta_{w,core}$). The observational data are derived from the**
**GPR and DGPS measurements in Monnier and Kinnard (2015b & 2016).**

**Table S1. Parameters of the sensitivity experiments. Scn-1.0 is the reference scenario that adopts the parameters of Murtèl-Corvatsch rock glacier. The other scenarios are designed by multiplying the reference value of each variable with the corresponding factor in their scenario labels.**

| Scenario | $A_{rg}$ (km²) | W (m) | $\alpha$ (°) | $h_{al}$ (m) | $\rho_d$ (kg/m³) | $\theta_{d,al}$ (%) | $\theta_{a,core}$ (%) |
|---|---|---|---|---|---|---|---|
| Scn-0.2 | 0.01297 | 40 | 3.2 | 0.6 | 1450 | 13 | 1.5 |
| Scn-0.4 | 0.02594 | 80 | 6.4 | 1.2 | 1700 | 26 | 3.0 |
| Scn-0.6 | 0.03892 | 120 | 9.6 | 1.8 | 1950 | 39 | 4.5 |
| Scn-0.8 | 0.05189 | 160 | 12.8 | 2.4 | 2200 | 52 | 6.0 |
| Scn-1.0 | 0.06487 | 200 | 16 | 3.0 | 2450 | 65 | 7.5 |
| Scn-1.2 | 0.07784 | 240 | 19.2 | 3.6 | 2700 | 72 | 9.0 |
| Scn-1.4 | 0.09081 | 280 | 22.4 | 4.2 | 2950 | 79 | 10.5 |
| Scn-1.6 | 0.10379 | 320 | 25.6 | 4.8 | 3200 | 86 | 12.0 |
| Scn-1.8 | 0.11677 | 360 | 28.8 | 5.4 | 3450 | 93 | 13.5 |

[Figure]

**Figure S2: Modelled relationships (grey shaded areas) between the ice fraction ($\theta_{i,core}$) and the surface velocity ($u_s$) of 95% confidence intervals for the three RGs monitored in the PERMOS network with model parameterisation Scheme 1. The ranges of the observed velocities (yellow bands) are used as velocity constraints for inferring ice content from the modelled relationships. Also shown are the reference ice content obtained from previous field-based surveys (blue lines). The inference ice contents are the mean values (solid black lines) with the estimated ranges (dash-dotted black lines).**

[Figure]

**Figure S3: Similar to Fig. S4, but showing results derived from model parameterisation Scheme 3. The grey shaded areas outline the modelled relationships between the ice fraction ($\theta_{i,core}$) and the surface velocity ($u_s$) with 95% confidence intervals. The yellow bands show the observed surface velocities, and the blue lines denote the reference ice contents. For each rock glacier, the intersection between the simulated $\theta_{i,core}$- $u_s$ relationship (grey shaded area) and the observed velocity (yellow band) gives the estimated range of ice content, as marked by the dash-dotted black lines. The inferred ice content is taken as the average value of the estimated range and indicated by the solid black line.**

~~The debris–covered glaciers (DCGs) investigated in this study show a discontinuous and inconsistent velocity field due to its heterogeneous and fast motion which is unfavourable for InSAR measurements (Fig. S4). Figure S5 presents that the DCGs exhibit similar and stable kinematic features in mean and median velocities as rock glaciers, yet Chola DCG shows a standard deviation in mean velocity as 15.4 cm yr⁻¹, much higher than that of the rock glaciers (3.4 cm yr⁻¹). The maximum velocity is variable with the largest value of 215.9±29.7 cm yr⁻¹, as shown in Chola DCG (Fig. S5-b). However, those statistics of velocity~~

[Figure]

**Figure S4: Velocity field map showing the discontinuous and inconsistent surface velocity distribution of Lobuche DCG.**

[Figure]

**Figure S5: Time series of the InSAR-derived downslope velocities of the two debris-covered-glaciers. The spatial mean velocities and uncertainties during each period are shown (black squares and error bars) as well as the median (blue) and maximum (red) velocities.**

[Figure]

**Figure S6: Modified after Fig. 11a in the manuscript. The yellow shading shows the observed surface velocity (~0.1 m yr$^{-1}$) and the vertical solid black line denotes the modelled ice content (71%). The red shading marks an assumed surface velocity (1 m yr$^{-1}$) and the estimated ice fraction is shown by the vertical dotted line (60%).**

**Table S2: Estimated rock glacier thickness ($T_{area}$) derived from the thickness–area relationship used in this study, and the corresponding bias relative to in situ measured thickness ($T_{ref}$) (Barsch et al., 1979; Cicoira et al., 2019a; Arenson et al., 2002; Hoelzle et al., 1998). The rock glacier thickness ($T_{slp}$) derived from thickness–slope angle relationship proposed by Cicoira et al. (2020), and the associated bias. The last row gives the mean absolute error (MAE) derived from the two methods.**

| Rock glacier | $T_{area}$ (m) | $T_{slp}$ (m) | $T_{ref}$ (m) |
|---|---|---|---|
| Murtèl-Corvatsch | 29 | 26.2 | 27 |
| Muragl | 24 | 19 | 20 |
| Schafberg | 24 | 20.8 | 25 |
| MAE | 2.3 | 2 | – |

[Figure]

**Figure S7: Modelled relationships (grey shaded areas) between the ice fraction ($\theta_{i,core}$) and the surface velocity ($u_s$) of 95% confidence intervals for the three RGs monitored in the PERMOS network assuming a thickness error of 6 m. The ranges of the observed velocities (yellow bands) are used as velocity constraints for inferring ice content from the modelled relationships. Also shown are the reference ice content obtained from previous field-based surveys (blue lines). The inference ice contents are the mean values (solid black lines) with the estimated ranges (dash-dotted black lines).**

[Figure]

**Figure S8: Similar to Fig. S7, but showing results with a thickness error of 10 m.**

**Table S3. Summary of the reference and inference ice contents derived from two scenarios assuming different thickness errors, namely 6 m and 10 m. The values in brackets following the inference ice contents give the corresponding bias from the reference ice contents. The last row presents the root mean square error (RMSE) of the two scenarios.**

| Rock glacier | Reference (%) | Inference and bias (%) | |
| --- | --- | --- | --- |
| | | 6-m thickness error | 10-m thickness error |
| Murtèl-Corvatsch | 85 | 66 (-19) | 66 (-19) |
| Muragl | 50 | 60 (10) | 61 (11) |
| Schafberg | 65 | 62 (-3) | 63 (-2) |
| RMSE | – | 12 | 13 |

---

## Author Response (AR2)

Second review to the article from Hu et al. 'Modelling rock glacier ice content based on InSAR velocity, Khumbu and Lhotse Valleys, Nepal':

I would like to acknowledge the extensive work performed by the authors to address my comments from the first review. The structure is much better and it is therefore easier to follow the workflow and understand the main findings. Assuming the second reviewer will cover the modelling part, this review mostly focuses on 1) suggestions to clarify InSAR elements and acknowledge limitations that were partly explained in the response to my review but not necessarily well included in the manuscript; 2) a list of detailed comments.

Note that the line numbers refer to the version without track changes (except for 4.1 section that is missing in the final version – cut by mistake I guess – but I found it in the version with track changes).

Re: We thank the reviewer for the constructive suggestions regarding a more effective presentation of the InSAR methodology, and the appreciable effort devoted to improving the manuscript in various ways. We consider these comments carefully and provide our point-by-point responses given below. The line numbers refer to the ones in the revised manuscript with track changes, aiming to help the reviewer and editors locate the revisions made correspondingly.

---InSAR method/results/discussion---

1.  I understand from your explanation in the response to my previous review that the threshold of 5 cm/yr is both based on an uncertainty estimate and an assumption that everything < 5 can be assumed as not significantly moving and therefore not relevant for the study of active landforms. Sounds still weird to me cause it neglects areas that are inactive (or transitional) but may still contain ice, e.g. some of the areas that were inventoried based on morphological criteria but discarded in your InSAR analysis. But anyway, as long it is well explained, it is surely ok. I still find it hard to follow at l.244-262, so please just try to be very structured in this part (step by step / criterion by criterion). A real InSAR section in the discussion would also be nice (now you are mostly focusing on modelling).

    Re: We agree with the reviewer that we might underestimate the ice storage by discarding the inactive parts of rock glaciers. We have highlighted that our model is applied to the "coherently moving part" and discussed this issue in *Sect. 5.4 Limited application to the coherently moving parts of rock glaciers in quasi-steady-state motion* (L730):

    "Second, our model is suitable to be applied to the coherently moving part. However, the parts of rock glaciers are in a transitional kinematic status (practically defined as velocities < 5 cm yr-1) or move as an individual portion from the coherently moving parts. Moreover, the 1-D InSAR method may fail to detect some moving areas of the landforms creeping nearly along the satellite's flight direction due to the lack of sensitivities of the LOS geometry. These parts may also contain ice but are excluded from our estimation, causing possible underestimation of ground ice as well."

    Regarding the analyzing procedure of InSAR results, we have enriched the InSAR section and divided into three steps: *Step1: Interferometric processing*, *Step 2: Calculating downslope velocities from high-quality interferograms*, and *Step 3: Determining the velocities of the coherently moving parts as the model constraint*. Specific modifications are presented in the following replies to the example questions.

• l.245-246: 'Uncertainties were quantified…': Without mentioning the values you calculated, this sentence is a bit useless. Is it the 10 cm/yr error you are mentioning later (l.259)? And if yes, why to choose 5 cm/yr as threshold based on another reference (Wang et al., 2017)?

Re: This sentence was written to introduce the uncertainty sources (both interferometric and DEM-relevant) considered in estimating uncertainties. We changed the sentence a bit to highlight the main information we would like to convey at L292:

"The projection was conducted considering the satellite's flight direction, the local incidence angle, and the landform topographic parameters including the aspect and slope angles (Massonnet and Feigl, 1998; Bechor and Zebker, 2006). We considered the propagation of errors introduced by the InSAR measurements and DEM data which were used to determine the associated topographic parameters (Hu et al., 2021)."

For each interferogram, we quantified the uncertainty at the pixel-level; among all the high-coherence pixels, the largest uncertainty is 9.8 cm/yr, so we claimed that "for one pixel, the velocity error is < 10 cm/yr". Here 10 cm/yr is the maximum uncertainty instead of the minimum so we do not regard it as the measuring capability of our method.

In fact, the velocity threshold (5 cm/yr) is introduced because we aim to reduce the influence of atmospheric delays, which is supposedly removed by subtracting the phase values of nearby inactive pixels from the landform pixels (Step 1 in Sect. 3.5.1). This operation performs well provided that the detected displacement is significant compared with the reference area. Otherwise, the correction may not be effective due to poor signal-to-noise ratios. The specific value was determined by considering the phase distribution in the interferogram and the morphologic parameters of the landform used in calculating the downslope velocity.

We have provided the reason at L299:

"…the mean velocity of the landform is larger than 5 cm $yr^{-1}$ (Wang et al., 2017). We set this empirical threshold considering the typical noise level (5 cm $yr^{-1}$) as we observed in most interferograms."

• l.248-249: Is the sentence 'the remaining pixels cover more than 40% of the surface' an additional criterion, i.e. did you discard landforms than have < 40% coverage? Not clear. If it is simply a fact/information, it should not be listed here.

Re: Yes, it is. We have listed as a separate criterion at L298:

"(2) the remaining pixels cover more than 40% of the landform surface;"

• l.250-252: 'Next, we defined and outlined… If the InSAR measured velocity is higher than 5 cm/yr in more than half of the periods…'. This sounds to me like an additional criterion and in that case why not listing it at l.249?

Re: The criteria listed at L.249 are used for selecting interferograms that cannot reliably represent the overall kinematics of a landform. In other words, we excluded some interferograms from further velocity analyses. Here at L.250–252, we listed another criterion to find the pixels in the coherently moving part from the series of observations constituted by the remaining interferograms. The two

sets of criteria were put forward out of different purposes: one for selecting interferograms and the other for selecting pixels.

We have separated the two selections into Steps 1 and 2 in Sect. 3.5.1.

• l.255-256: 'an area actively in motion with the landform as a whole': that is a poor phrasing, and one could argue that by discarding areas within the morphological delineation, you don't consider the landform as a whole.

Re: We have re-written this part at L302:

"Field observations have revealed that multiple areas moving differentially can occur on rock glaciers and exhibit complex kinematic patterns (e.g., Buchli et al., 2018), which violates the assumption of a continuously moving body (Sect. 3.1, Fig. 3). Therefore, we aim to identify the coherently moving part of the landform that corresponds with our assumption and is thus suitable for model application.

After the data-refining procedure in Sect. 3.5.1.2, for each landform, the remaining interferograms constituted a series of observations spanning multiple years. Then we defined and outlined the "coherently moving part" of each landform by considering the time series of downslope velocity of each pixel acquired during the observational periods. If the InSAR-measured velocity is higher than 5 cm yr$^{-1}$ in more than half of the interferograms at a given pixel, it was included in the coherently moving part of the landform. Otherwise, the pixel cannot be regarded as actively in motion with the coherently moving area but in an inactive or transitional kinematic status."

• l.258-259: 'For one pixel, the velocity error is < 10 cm/yr, and the error of the mean velocity is limited to < 1 cm/yr.' What is the difference between the mean velocity and the velocity error? If 10 cm/yr of error, why using 5 as criterion (l.249)?

Re: The first value (< 10 cm/yr) is associated with the velocity measured at each pixel, which has two sources of error, i.e., interferometric processing and DEM. The second error (< 1 cm/yr) is derived by error propagation when we calculated the mean velocity, which is the arithmetic mean of all pixel values covering a landform. In this case, the mean velocity error is the square root of the quadratic sum of all the velocity errors. In addition, the equations we used to derive the uncertainties can be found in our open-access codes (l.351–365 and l.401): https://github.com/cryoyan/DeeplabforRS/blob/master/read_raster_for_shapefile.py

We have defined the first error at L294:

"We considered the propagation of errors introduced by the InSAR measurements and DEM data which were used to determine the associated topographic parameters (Hu et al., 2021). For each pixel, we found the velocity error is < 10 cm yr$^{-1}$."

And second error at L329:

"The error of the mean velocity can be derived by error propagation of all the pixels taken into account, which is limited to < 1cm yr$^{-1}$"

We have explained the choice of threshold in the response to the first example question.

2. Figure 10: The color scale is confusing (0-30 cm/yr with discrete classes). If you discarded everything < 5, the scale should start at 5. I would also suggest showing differently areas with low

coherence and those that are moving under the threshold. It is not at all the same criterion, it can in some cases mean the complete opposite: e.g. loss of coherence due to too fast movement. It looks for example likely for the upper part of Fig.10e (and generally likely when using 70-92 days interferograms). An alternative is to add an explanation in the caption mentioning that the transparent pixels (areas without InSAR coverage) are either due to low coherent or low velocity. And maybe add a discussion about this in Section 5?

Re: We did not show the low-coherence or low-velocity area because this figure was primarily plotted to present the average velocities of the coherently moving parts which were determined from a series of interferograms spanning different observational periods, making it practically difficult to show the status of the masked-out areas. There can be various situations. For example, one pixel could show a velocity larger than 5 cm/yr in one interferogram out of five interferograms, and in the other four interferograms, this pixel lost coherence due to its fast motion. In the latter case, this pixel can be regarded as an individual moving part from the coherently moving part. It is common that one rock glacier has more than one moving areas. More importantly, decorrelation is not necessarily caused by fast movement. It can occur due to changes of surface conditions, such as the soil moisture, the shift between freeze and thaw status. Therefore, we tend not to over-interpret the kinematics of the decorrelated areas.

Therefore, we have added an explanation in the caption (L598) and mentioned this issue in the discussion (L715, presented in our response to the 1# comment), according to the reviewer's suggestion.

We have also changed the color bar to start from 5 cm/yr. In addition, the discrete classes are only one way to present the magnitude of velocity, as a commonly used format in the open-source software QGIS that we adopted for plotting this figure, The coloring scheme is continuous so that the velocities values presented are not discrete.

[Figure]

**Figure 8: Velocity field maps show the average velocities of the coherently moving parts of the five rock glaciers (blue outlines) in the study area. The boundaries of the landforms delineated in Jones et al. (2018b) are in red. The transparent areas between the red and blue boundaries are due to low coherence or low velocity during the observational periods.**

3. Another issue is the LOS measurements and the downslope projection (quickly mentioned at l.245 and partly discussed in 5.4 and 5.6). You now cover the question of potential subsidence, but I think it is also important to mention that LOS measurements can lead to significant underestimation on slope facing the radar or on N-S slopes (with creep direction orthogonal to the LOS), as in the cases shown in Figure 10a-c (a: because you mainly used ascending data; b-c: N-S facing). I believe it is no coincidence that the coverage is better for both cases facing away from the radar (d-e). You may have areas that fall under your 5 cm/yr threshold although that are in reality moving at cm-dm level. I am aware there is not much to do to solve this issue (except discarding these areas or applying multi-geometry InSAR methods as mentioned in 5.6), but the limitation must at least be clearly acknowledged.

Re: We would like to provide additional explanations to the issue the reviewer pointed out here. We acknowledged that our method may fail to perform effectively on rock glaciers creeping nearly along the satellite's flight direction and lead to weak signals in the interferograms recording the LOS phase shifts. Fundamentally, LOS measurements from InSAR are insensitive to any movements perpendicular to the LOS direction, resulting in completely omission of that particular component and an underestimation of the magnitude of the 3D velocity vector. If the true 3D velocity is exactly along the slope direction, the LOS velocity can be restored without an underestimation issue. In other words, the velocities presented in Fig 10a–c are not underestimated under the downslope motion assumption. In reality, the reprojected velocity might be underestimated or even overestimated depending on the geometry (1D downslope vs. 3D real downslope vs. LOS).

However, as the reviewer observed, the phenomena that data coverage is smaller for landforms moving nearly perpendicular to LOS direction, is likely to happen. This is because the very small phase shift (the phase difference between the subject pixel and the reference pixel, L286) may not be successfully recognized from the interferogram due to the poor signal-to-noise ratio (source of the noise is mainly atmospheric errors). Theoretically, the omitted part might move slower than the detected part, though may still lie above the threshold value.

We have extended the discussion to include this aspect at L732:

"Moreover, the 1-D InSAR method may fail to detect some moving parts of the landforms creeping nearly along the satellite's flight direction due to the lack of sensitivities of the LOS geometry. These parts may contain ice but are excluded from our estimation, causing possible underestimation of ground ice as well."

4. In general, your answer to my comments contains several elements/references to other studies that can be added to the article. I believe some of my questions may also be raised by other readers. Due to your choice of introduction structure and limited discussion regarding InSAR you have almost no reference to other studies using InSAR for mountain permafrost applications (except Wang et al., 2017). I think it will strengthen the article if you show you have read what other have done (either in the intro – as background/state of the art; in the method – to justify your choices; or in the discussion – to relate with the way others deal with similar challenges).

Re: We did not present a very detailed introduction of InSAR to avoid a very lengthy methodology section, but a more comprehensive list of relevant bibliography, as the reviewer suggested, would improve the quality of discussion. We have added the citations to place our InSAR work in context in method at L264:

"InSAR has been widely applied to quantifying surface velocities of rock glaciers (e.g., Bertone et al. 2022; Reinosch et al. 2021; Rouyet et al. 2019; Zhang et al. 2021). In this study, we adopted the conventional Differential InSAR method to derive the surface velocities by assuming rock glaciers creep along the slope direction (Brencher et al., 2021; Hu et al., 2021; Liu et al., 2013; Wang et al., 2017)."

And in the discussion at L755:

"In addition, a more accurate 2-D surface velocity can be obtained by using multi-track InSAR data (e.g., Bertone et al., 2022; Zhang et al., 2021), allowing us to apply the model to rock glaciers with a complex velocity field."

Reference

Bechor, N. B. D. & Zebker, H. A. (2006). Measuring two-dimensional movements using a single InSAR pair. *Geophysical Research Letters*, 33(16). https://doi.org/10.1029/2006gl026883.

Bertone, A., Barboux, C., Bodin, X., Bolch, T., Brardinoni, F., Caduff, R., Christiansen, H. H., Darrow, M. M., Delaloye, R., Etzelmüller, B., Humlum, O., Lambiel, C., Lilleøren, K. S., Mair, V., Pellegrinon, G., Rouyet, L., Ruiz, L. & Strozzi, T. (2022). Incorporating InSAR kinematics into rock glacier inventories: insights from 11 regions worldwide. *The Cryosphere*, 16(7), 2769–2792. https://doi.org/10.5194/tc-16-2769-2022.

Liu, L., Millar, C. I., Westfall, R. D. & Zebker, H. A. (2013). Surface motion of active rock glaciers in the Sierra Nevada, California, USA: inventory and a case study using InSAR. *The Cryosphere*, 7(4), 1109–1119. .

Massonnet, D. & Feigl, K. L. (1998). Radar interferometry and its application to changes in the Earth's surface. *Reviews of Geophysics*, 36(4), 441–500. https://doi.org/10.1029/97rg03139.

Reinosch, E., Gerke, M., Riedel, B., Schwalb, A., Ye, Q. & Buckel, J. (2021). Rock glacier inventory of the western Nyainqêntanglha Range, Tibetan Plateau, supported by InSAR time series and automated classification. *Permafrost and Periglacial Processes*, 32(4), 657–672. https://doi.org/10.1002/ppp.2117.

Rouyet, L., Lauknes, T. R., Christiansen, H. H., Strand, S. M. & Larsen, Y. (2019). Seasonal dynamics of a permafrost landscape, Adventdalen, Svalbard, investigated by InSAR. *Remote Sensing of Environment*, 231, 111236. https://doi.org/10.1016/j.rse.2019.111236.

Zhang, X., Feng, M., Zhang, H., Wang, C., Tang, Y., Xu, J., Yan, D. & Wang, C. (2021). Detecting Rock Glacier Displacement in the Central Himalayas Using Multi-Temporal InSAR. *Remote Sensing*, 13(23), 4738. https://doi.org/10.3390/rs13234738.

---Detailed comments---

1. - l.15 and l.17: you use a mix of terminologies 'creep rate' / 'velocity' / 'kinematics' all along the manuscript. It would be easier to follow if you are consistent. Creep rate is ok when referring the mechanisms of the landforms, but InSAR only gives an information about the surface velocity that we infer being representative of the creep rate. Kinematics is a more generic terminology often referring to all spatio-temporal patterns of rock glacier creep rate. So when referring to InSAR products, I would suggest using 'velocity' only (see also Fig.2). Creep rate/kinematics can still be used when more generally referring to the processes.

Re: Thanks for the clarification. We have changed the terminology throughout the manuscript according to the reviewer's suggestion. Many of these modifications are mentioned in the response to the following detailed comments.

We have also changed the label in Fig.2:

[Figure]

2. l.15-16: …the Chilean Andes and the Swiss Alps.

   Re: Modified.

3. l.19-20: Based on previous inventories and extrapolating our findings to the entire…

   Re: Modified.

4. l.34: I have commented it in my previous review: what is the different between rock slope failure and mountainside collapse? Sounds very unclear to me. If you mean 'well-delineated rockslides' compared to 'large deep-seated gravitational slope deformation', maybe just say it so.

   Re: We have changed it to "rock slope failures such as rockslides and rock avalanches" and added a reference paper focusing on rock slope failures.

   Jarman, D. & Harrison, S. (2019). Rock slope failure in the British mountains. *Geomorphology*, 340, 202–233. https://doi.org/10.1016/j.geomorph.2019.03.002.

5. l.35: Could add here a reference to paraglacial studies, e.g. Ballantyne, or more recent references.

   Re: Please refer to the response above.

6. l.40: …and/or undergo transitions to rock glaciers (not all degrading glaciers transition to rock glaciers).

   Re: The reviewer is right. Now modified.

7. l.53: …approaches to quantify (or estimate) the likely ice content…

   Re: Added.

8. l.57-58: twice rheological in that sentence, you can probably remove the first.

   Re: Removed.

9. l.64: …in Nepalese rock glaciers….

   Re: Added.

10. l.65: …freshwater reservoirs…

Re: Changed.

11. l.72: …region. Previous seismic…

Re: Changed.

12. l.82: I think 'critical condition' is a bit vague. Just as a suggestion: 'critical limit' instead?

Re: We have changed the phrase as suggested.

13. l.84: Same here about 'warm and unstable condition'. What about 'warm and unstable state' instead?

Re: Replaced.

14. l.94: …the method to extrapolate the results at the regional scale (Sect.3.6).

Re: Changed.

15. l.99: 'feature distributed in ice-rich alpine permafrost': sounds a bit upside-down to me. It is ice-rich permafrost at these locations because of favorable conditions to accumulate debris and water that accumulates as ice due to permafrost conditions. Anyway, this sentence should probably come at the very beginning of the introduction and could be simplified as: 'Active rock glaciers are ice-rich permafrost landforms creeping downslope in alpine environments' (for example).

Re: We have changed the sentence to "Active rock glaciers are viscous flow features embodying ice-rich permafrost." We do not move it to the beginning of the introduction because the manuscript starts with introducing the ice storage in intact rock glaciers. Active rock glaciers are not the focus.

16. l.104: (mainly temperature and pressure)

Re: Modified.

17. Eq (2) and l.112-115: Tau / Tau-threshold is not clear to me. Probably because Tau is not defined, independently of the threshold stress. Good if it can be slightly clarified.

Re: We defined Tau at an earlier place when introducing Eq (1). We have added a definition here as well at L126:

"τ is the driving stress…"

18. l.120-122: long/heavy phrasing: to develop… to describe… in a… based on… Two sentences?

Re: We have re-written the sentence at L141:

"…we used a constant effective viscosity (B) to develop an empirical formula to describe the deformation behaviour of rock glaciers in a warm permafrost environment (> -3°C). The empirical formula was developed based on existing observational data and laboratory findings."

19. l.123: One could argue that it is wrong applying a glaciological model to rock glacier. The phrasing of that sentence is at least a bit unfortunate if not better explained. Consider rephrasing or explain better the choice of applying a glaciological setup.

Re: We have rephrased the sentence to mention the rationale of doing so at the beginning (L144):

"We assume a homogeneous structure and consider each rock glacier as a slab with uniform width and thickness and a semi-elliptical cross-section, resting on a bed of constant slope, which is a common setup in glaciology (Cuffey and Paterson, 2010)."

20. l.132-133: …the short-term rock glacier kinematic patterns are irrelevant to this study focusing on modelling the relationship between…

Re: Changed.

21. l.138: Here Tau is defined. It could come before (and potentially repeated here as well). See comment regarding l.112-115.

Re: Added. See the previous response.

22. l.166. 'glaciological studies': see my previous comment. To be clear, I am aware it is usual to adapt glaciological ice flow models to rock glacier flow calculation, but there is maybe a more elegant way to phrase it?

Re: We have rephrased it at L195:

"… we first used an empirical average value as assumed in modelling pure ice creep:"

23. l.167-171: You are performing two tests with different calculations of n, right? It is not well highlighted, and I think the conclusion of this comparison is not mentioned further in the results. Which one did you finally choose and why? If not so relevant to the paper, I would suggest to just mention the selected n calculation you finally used for the presented results.

Re: Yes, we used both relationships and mentioned this in model calibration (Sect. 3.2). The results and our selection are presented in Sect. 4.1 and 4.2. (We realized that Sect. 4.1 were missing in the clean version, now added back.)

24. l.180: Eq. 12 or 13: see previous comment: missing info on where/when you use each and why.

Re: We have added back the missing Sect. 4.1 where we show the results. The reasons were mentioned at L217:

"First, we adopted the exponential $B-\theta_{i,core}$ relationship estimated by Monnier & Kinnard (2016) with the same dataset and a constant creep parameter n (Eq. 12). Then by integrating the relationship between n and ice content (Eq. 13) …"

25. l.201: …can be extract based on optical imagery (xxx). Nice to specify which images and which acquisition dates are used for this step.

Re: We have added the imagery information at L229:

"…we first outlined the boundaries of the three rock glaciers from Google Earth images (September of 2018), from which their shapes and areal extents can be extracted using Geographic Information System tools."

26. l.204 and Table 1: A (Area) is the actual parameter used in Eq. 14. You explain the way W is measured (and list the values in the table) but you don't specify the length. Looks weird, considering

that A is probably a simple product of both. Also based on the envelop rectangle? If yes, you can just write 'the length and width of each rock glacier' (l.204) and add the length in Tab.1.

Re: We don't use length as an input parameter. Area (A) can be directly obtained based on the boundary polygon using QGIS. Only width (W) is derived based on the envelop rectangle. And the W parameter is used for calculating the shape factor ($S_f$) to consider the frictional effect between the rock glacier and surrounding bedrock.

27. l.217: …with variable ice fractions…

Re: Changed.

28. l.219-222: Heavy/unclear. Suggestion: …naming each scenario after a multiplication factor which indicate the ratio between the applied parameter and the reference scenario. For two parameters, we applied a value range according to the known natural variability based on observations (…).

Re: We have re-written the sentence at L252 based on the suggestion.

29. l.227: 'surface velocities' instead of 'surface kinematics'? See my comment at l.15/17.

Re: Yes, we have checked and changed the terminology throughout the manuscript.

30. l.266: See comment at l.201: specify which data are used to measure the geometry.

Re: We have added the data and tools used in Sect. 3.3 as the reviewer suggested.

31. l.230: See my main comment

Re: We have extended the InSAR section as presented in the response to the main comment.

32. l.234-235: why different baselines for ALOS and ALOS-2? Regarding the temporal baseline, if applying a simple calculation (24 cm (wavelength) divided by 4, divided 92 days, multiplied by 365 days), you get 24 cm as a theoretical limit of phase ambiguity. This could be documented somewhere, and used to discuss the related limitation. You may induce that you miss metric velocities due to decorrelation (for ex. in the upper part of Fig.10e?).

Re: The different maximum temporal baselines for ALOS and ALOS-2 we adopted were empirical based on the coherence level of interferograms generated. And the temporal baselines we actually used are 46 days and 14 days, respectively, as listed in Table 2.

The theoretical limit of phase ambiguity as derived and suggested by the reviewer is not very meaningful in our practical case for two reasons. First, we used interferograms with 46-day and 14-day time spans, thus were able to resolve faster movements than a 92-day-span interferogram. Second, we projected LOS velocities to downslope directions and thus scaled up the upper limit. For example, when a pixel moves faster than 24 cm/yr along LOS, in our real data, it is 47 cm/yr for 46-day pairs and 156 cm/yr for 14-day pairs, allowing us to resolve relatively fast movement. Therefore, we have chosen not to add a quantitative assessment of detection limit.

33. l.236 and 239: …were applied to the interferograms, which… In general: applying an 8x8 multi-looking on ALOS data, I doubt you get a 30m ground resolution, so does it mean you oversampled the final product and simply georeferenced it according to the DEM? If yes, good to say it.

Re: Yes. We have added this information at L285:

"The final georeferenced interferogram has a ground resolution of ~30 m according to the DEM."

34. l.260: I guess you simply mean 'Finally, we averaged the velocities over the entire observational period and used the results as constraint for modelling ice content'.

Re: Not exactly. We meant to say that we consider the range of the spatial mean velocity of each observational period and use the range as the constraint. For example, we measured the velocity of Kala-Patthar rock glacier during seven periods, which produced seven spatial mean velocities (the average of all pixels covering the landform), namely 10.3cm/yr, 11.4 cm/yr, 12.0 cm/yr, 10.5 cm/yr, 11.7 cm/yr, 13.4 cm/yr, and 13.6 cm/yr. Instead of taking the average of the seven values, we take the range of the values as the constraint, which is 10.3–13.6 cm/yr (as shown by the yellow band in Fig. 11).

We introduced how to get the mean velocity at L328:

"Then, we analysed the velocity values of all pixels within the coherently moving part of the landform and selected the mean, median, and maximum values for each observation to characterise the surface kinematics of the landforms."

We have specified that it is the spatial mean at L331:

"Finally, we take the range of the spatial mean velocities…"

35. l.261: partly copied-pasted from l.132-133. See my previous comment.

Re: We have noticed this repetition and changed the sentence at L332:

"By doing so, the short-term feature of rock glacier kinematics is neglected in our study."

36. l.262: …averaged velocity in our study.

Re: We have changed the sentence. See the previous response.

37. l.268-269: …The empirical relation for calculating rock glacier thickness used in the validation processed (ref Section?) was again applied here to…

Re: We have changed the sentence as suggested at L339:

"The empirical relation for calculating rock glacier thickness used in the validation procedure (Sect. 3.3) was applied here to obtain the thickness parameter."

38. l.270. …the averaged InSAR-derived downslope mean annual velocity based on the entire observation period, except…

Re: We keep the phrasing here. Please refer to our response to comment #34.

39. l.273-276: …the modelled ice contents and the volumetric extent of the… Then, we used average water equivalents to represent the water storage in a typical rock glacier in this region. … that reported 4226 intact rock glaciers over the Nepalese Himalayas.

Re: Re-phrased as suggested.

40. Section 4.1 (l.550 according to version with track changes – missing in the version without track changes): By applying the different regression models…

Re: We have restored the missing section and corrected the typo.

41. l.286: Uncertainties from the statistical analysis (dashed lines in Fig.4) have been considered in the simulation.

Re: Changed.

42. l.288: …mean annual surface velocities…

Re: Modified.

43. l.291-292: …the reference ice content, i.e. the average value of the…

Re: Modified.

44. l.294: …we see that Scheme 2 is the optimal model for the…

Re: Modified.

45. Figures 5-7: It would be easier to have similar caption texts for all figures. You could also consider moving Figures 5 and 7 in Supplementary, considering that you conclude that 2 is the best, without detailing much of the results from the other Schemes. Table 3 anyway summarizes the main points. That would reduce the length of the paper while still providing necessary info.

Re: We have moved Fig. 5 & 7 to the supplementary document as suggested.

46. l.322: The results of sensitivity experiments are normalised … (Fig.8).

Re: Changed.

47. l.325: …slope angle. In the extreme…

Re: Changed.

48. l.336-337: Surface velocities of the nearby… were also measured...

Re: Modified.

49. l.343: 'downslope rate'

Re: Modified.

50. l.347-348: 'The acceleration of Tobuche cannot be confidently revealed by our data and 2015 acquisition was therefore discarded to calculate the average of the entire period.' If I understood correctly, you did not use that value in the following, correct?

Re: Correct. The value was not used, but we did not calculate the average velocity (detailed in response to comment #34). We have re-written the sentence at L579:

"The acceleration of Tobuche cannot be confidently revealed by our data and 2015 acquisition was therefore discarded from the velocity series used as the modelling constraint."

51. l.349: 'the observation period'

Re: Modified.

52. Fig.10: See main comment. Caption: …are shown in red (or …are the red polygons).

Re: We have modified the caption as suggested here and in main comment.

53. l.361-362: …ice fraction of the landforms / …for individual landforms. Is it not more correct to write here: …of the coherently moving part of the landforms? In general: good to add a discussion about this limitation (discrepancy between morphological delineation and InSAR-based polygons).

Re: We have specified the coherently moving part in the sentence, and in the abstract (L19) and conclusions (L787) as well. The discussion has been added in Sect. 5.4 as detailed in the response to main comment.

54. l.369-370: The ranges of the InSAR-derived velocities (yellow bands) are used as velocity constraints…

Re: Modified.

55. l.377: Based on the estimated water… the extrapolated amount of water…. The smaller estimate compared to Jones is most likely due to the smaller considered extent of the landforms. That is maybe something to add in the discussion.

Re: The sentence has been modified.

We do not add the suggested comparison because Jones et al. (2018b and 2021) used a completely different method to estimate ice content of individual landform, making it difficult to assess whether the gap is primarily caused by the conservative areal extent used in this work. The uncertainty of extrapolation result is discussed in Sect. 5.5.

56. l.383: See main comment: missing an InSAR section in the discussion.

Re: We have extended the discussion in Sect. 5.4, as detailed in the response to main comment.

57. l.397-398: …within the timescale of our study…

Re: Modified.

58. l.399: '…which is consistent with the fact that rock glaciers are currently not a major contribution to surface runoff.' This is just a phrasing comment but in addition I don't really see the link with the start of the sentence.

Re: We have re-phrased the sentence as suggested.

The link between this part and the start of the sentence is as such: the fact that rock glaciers are not major sources of surface runoff indicates that the ice stored in rock glaciers are not melting rapidly, so that we can assume the amount of ice remains constant within the timescale of our study.

59. l.403: Upside-down sentence: it is not the lack of knowledge that limits the field data but the opposite. 'amount of data' instead of 'size'

Re: We have re-written the sentence at L666:

"Currently, the amount of field data is limited for deriving a statistical relationship with a low degree of uncertainty since detailed knowledge of rock glacier composition is largely lacking."

60. l.405: we relied on geophysical data (n = 14): quite cryptic explanation for a discussion. No need to use the parameter letter and value, but what it actually means.

Re: We have removed the mathematical expression and added the information at L684:

"However, due to the limited amount of calibration data (14 measurements in total)…"

61. l.406: …hypothesized that the empirical…

Re: Modified.

62. l.408: 'amount of data' instead of 'size'

Re: Modified.

63. l.418: …has generated discussions…

Re: Modified.

64. l.428: …introduced by thickness derivation cannot be eliminated when applied to rock glaciers without known information of structure.

Re: Modified.

65. l.435: …we measured surface velocities…

Re: Modified.

66. l.438: 1-D InSAR method. We converted the LOS measurements… by assuming the rock moves downslope without… (or along the slope direction). See also main comment, I believe the LOS measurements are not only an issue when dealing with subsidence. Something about N-S facing slopes (and slope facing towards radar) could be added somewhere.

Re: We have modified the sentence and extended the discussion. See the response to the main comment.

67. l.446: "Errors may arise" sounds like an understatement to me. Why not clearly saying: 'The simple extrapolation method was not designed for an accurate quantification of… but for an order of magnitude estimation of the potential water storage…' Here it could also be said more clearly that the dimensions of the rock glaciers are the main constraints to the ice content according to your results (due to relatively similar velocities), so as a prospect: the extent of the landforms could be used to extrapolate the results.

Re: We have re-written the first sentence to state the limitation more clearly as suggested.

68. l.450-451: Upside-down phrasing, I think. 'The average velocity of five rock glaciers is not able to represent the ice content of all rock glaciers in the entire mountain range' is what you want to say, I guess?

Re: We meant to say that not all rock glaciers have similar velocities as the five landforms, and the other rock glaciers could have ice contents different from our study objects as well. We have re-phrased the sentence:

"In reality, rock glaciers typically creep at rates ranging from decimetre to several metres per year (RGIK, 2020), thus the average ice content of the five rock glaciers with similar velocities may not be able to represent that of all rock glaciers with various velocities in the entire mountain range."

69. l.456: No, due to polar orbits, combining asc+desc does not provide an accurate 3-D info in most cases. Rather 2-2.5D. In general, it is good to mention it of course but a bit weird considering that you have not fully introduced the problem. Add references?

Re: We have rephrased the sentence and added citations. We have also better introduced the limitation of 1-D InSAR according to the major comment.

70. l.460-463: make two sentences. The end does not work.

Re: We have re-written the sentence into two at L782:

"With the likely emergence of more data to be integrated for model calibration and validation, it is promising to improve the accuracy of the approach. We expect the improved model to be applied to mountain permafrost regions where rock glaciers are widespread for preliminary water storage evaluation."

71. l.465: past tense in this sentence?

Re: Modified.

72. l.477: …from our inferred results…

Re: Modified.

73. l.479: the ratio between… is 1:17.

Re: Modified.

74. l.483: final point is missing ;)

Re: Added.

75. Reference list: Wang et al., 2017 is missing.

Re: Added.

Again, we sincerely thank the reviewer for the great effort to improve the quality of the manuscript considerably.

Report #2

Dear authors and Editor,

Thank you very much for allowing me to act as a referee for the present manuscript. I read the manuscript with great interest and I want to congratulate the authors for their efforts in presenting this manuscript, which is well written and developed.

The study from Hu et al. entitled "Modelling rock glacier velocity and ice content, Khumbu and Lhotse Valleys, Nepal" intends to propose a model to infer rock glacier ice content based on InSAR velocities. The model parameters were calibrated based on observational data on "Las Liebres rock glaciers, in Chilean Andes" and validated using data from four-rock glaciers in the Swiss Alps, because no field observations are available on the current study site. Then, they applied it to the five-rock glacier in Nepal intending to estimate ice/water storage on the studied rock glaciers and at regional scale.

Re: We thank the reviewer for the valuable comments regarding the velocity measurement, the uncertainty analysis, and the extrapolation method. We consider these comments carefully and provide our point-by-point responses given below. The line numbers refer to the ones in the revised manuscript with track changes, aiming to help the reviewer and editors locate the revisions made correspondingly.

From a general point of view, the manuscript is well written, this approach is quite novel. However, I think that there are some major drawbacks in this manuscript that from my point of view could question their acceptance. I would not recommend that the manuscript be published on The Cryosphere as presented for the following reasons:

1. A better assessment of surface velocities must be presented:

i. Even with the new changes on the "Track changes version", there is still unclear (at least for me) how the authors have obtained surface velocities. For example, why do they use 5 cm yr-1 as a threshold instead of that inner intrinsic value for each interferogram? It is not clear to the reader what does mean "coherent moving parts" and how they have been selected. More details are needed to clearly understand why the authors selected these values.

Re: We have re-written the InSAR section (Sect. 3.5.1) according to the comments from Reviewers #1 and #2.

The velocity threshold is introduced considering the typical level of noise in the interferograms based on our experience. We used this threshold for two purposes: first to select high-quality interferograms; second to find the percentage of reliable measurements at each pixel across all interferograms. It is a practically effective way for us to outline the "coherently moving part" of each rock glacier. For example, assuming that one pixel shows reliable velocities (>5 cm/yr) in four out of ten interferograms, meanwhile it is decorrelated and less active in four and two interferograms, respectively, it is difficult to decide whether this pixel is moving along with the main part of the rock glacier, or moving as a separate part, or in transitional status. In other word, we regard the typical noise level (5 cm/yr) as a more reliable and conservative threshold than the "inner intrinsic value" for obtaining velocity data.

We have added the reason at L299:

"…the mean velocity of the landform is larger than 5 cm yr$^{-1}$ (Wang et al., 2017). We set this empirical threshold considering the typical noise level (5 cm yr$^{-1}$) as we observed in most interferograms."

Regarding the coherently moving part, we have provided more details at L301:

"Step 3: Determining the velocities of the coherently moving parts as the model constraint

Field observations have revealed that multiple areas moving differentially can occur on rock glaciers and exhibit complex kinematic patterns (e.g., Buchli et al., 2018), which violates the assumption of a continuous moving body (Sect. 3.1, Fig. 3). Therefore, we aim to identify the coherently moving part of the landform that corresponds with our assumption and thus suitable for model application.

After the data-refining procedure in Sect. 3.5.1.1, for each landform, the remaining interferograms constituted a series of observations spanning multiple years. Then we defined and outlined the "coherently moving part" of each landform by considering the time series of downslope velocity of each pixel acquired during the observational periods. If the InSAR-measured velocity is higher than 5 cm yr$^{-1}$ in more than half of the interferograms at a given pixel, it was included in the coherently moving part of the landform. Otherwise, the pixel cannot be regarded as actively in motion with the coherently moving area but in an inactive or transitional kinematic status."

ii. InSAR uncertainties need revision. The authors mentioned that they took "three random pixels" within a buffer of 300 m around each rock glacier. This selection is a bit tricky because, with Sentinel-1, you have a large scene from which it is possible to identify stable areas, which are not susceptible to change. More details are needed to understand why only three pixels? Are these three pixels statistically representative of the area? Then, the authors argued that they used the methodology of Hu et al., 2020, but even in this publication, is very hard to understand how they obtain an averaged uncertainty value. For example, later on, in section 3.5.1 (around L485), the authors mentioned that for one pixel, the velocity error is < 10 cm yr-1, why do not mention how much is the uncertainty precisely? Have been the uncertainties quantified?

Re: First we would like to clarify that the objective of using the average phase of the "three random pixels" is to represent the background signal detected in the near surroundings of a rock glacier, which should contain the phase shift caused by atmospheric delay. Then we correct this error by subtracting the reference phase value from each landform pixel. The reference pixels are not required to show large-scale representativeness.

Second, we have precisely quantified the uncertainty of all the rock glacier pixels in all the interferograms. We did not mention the exact value (the maximum error is 9.8 cm/yr) because here we aim to show the level of uncertainties in general.

Then, regarding how we obtain an uncertainty value based on our previous publication, we followed the rules of error propagation. The equations are not listed in the paper but can be found at l.351–365 and l.401 in our published code:
https://github.com/cryoyan/DeeplabforRS/blob/master/read_raster_for_shapefile.py

iii. I wonder if the SRTM DEM is the adequate DEM to correct the topographic effect on InSAR interferograms. It is partially well known that better quality of DEM used for this purpose, better results could be obtained on the InSAR interferograms and later on in the unwrapped products. However, I wonder, why the authors do not consider the highest resolution DEM like "8m High Mountain Asia DEM" (free available at https://doi.org/10.5067/0MCWJJH5ABYO)? I mention that because, in the European Alps, many surprises have been found when SRTM DEM is used to correct the topography effect.

Re: We conducted comparison experiments between the SRTM DEM and HMA DEM for our previous study focusing on periglacial landforms on the Tibetan Plateau and didn't find noticeable differences between the interferograms. Then we returned to SRTM DEM due to a preference of radar imagery-generated DEM.

We thank the reviewer for the sharing research experience in the European Alps. We should systematically compare different DEMs for topography correction in the future work.

iv. The delimitation of moving/coherent parts. There is no clear how the authors define "coherent moving parts" and how they obtain velocity fields. For example, for "Kalaa-Patthar and kongma rock glacier", there is only a small path of movement (i.e. it is very surprising because the velocities plotted in Figure 10 are an average of several unwrapped velocity products). However, on the rock glacier inventory made by jones et al., 2018b, a bigger polygon has been delineated. How do the authors assess the average velocity of the rock glacier by considering a single small patch? Is this patch statistically representative of the entire landform? If the other sectors of the rock glaciers do not move (still within the Jones et al 2018b delineation), does it means that these parts can be considered relict rock glaciers? Another example, for the case of "Tobuche rock glacier", the authors did not identify coherent velocities in the upper sector of the rock glacier (this could be probably due to relatively rapid movements of this sector or simply because there are no good enough results in the interferograms given the complex topography) which supposed to have more important ice concentration than the lower sector, how the authors deal with this problem? No explanation has been done yet in the manuscript. For this specific rock glacier, is the lower sector surface velocity representative of the behaviour of the upper sector? From my experience, I would say no, but it should be demonstrated statistically for your case.

Re: We have extended our introduction to the "coherently moving part" to better explain our motivation for the defining this.

In Sect 3.5.1 at L301:

Step 3: Determining the velocities of the coherently moving parts as the model constraint

"Field observations have revealed that multiple areas moving differentially can occur on rock glaciers and exhibit complex kinematic patterns (e.g., Buchli et al., 2018), which violates the assumption of a continuous moving body (Sect. 3.1, Fig. 3). Therefore, we aim to identify the coherently moving part of the landform that corresponds with our assumption and thus suitable for model application.

After the data-refining procedure in Sect. 3.5.1.1, for each landform, the remaining interferograms constituted a series of observations spanning multiple years. Then we defined and outlined the "coherently moving part" of each landform by considering the time series of downslope velocity of each pixel acquired during the observational periods. If the InSAR-measured velocity is higher than 5 cm yr$^{-1}$ in more than half of the interferograms at a given pixel, it was included in the coherently moving part of the landform. Otherwise, the pixel cannot be regarded as actively in motion with the coherently moving area but in an inactive or transitional kinematic status."

In the modelling work, we only focus on the coherently moving part recognized by InSAR data, because it is in line with the model assumptions. We did not intend to assign the velocity of the coherently moving part to the entire rock glacier, nor to assume it is able to represent the velocity of other parts of the landform. The other parts can be either inactive or in very rapid motion, but we cannot draw conclusions since both situations result in poor InSAR data quality. Besides, this identification is not directly relevant to ground ice estimation.

We agree with the reviewer's (and reviewer #1's) insight that the rapidly moving or inactive sections could also store ice, which could lead to underestimation. We have highlighted that our model is applied to the "coherently moving part" and discussed this issue in *Sect. 5.4 Limited application to the coherently moving parts of rock glaciers in quasi-steady-state motion* (L730):

"Second, our model is suitable to be applied to the coherently moving part. However, the parts of rock glaciers that are in a transitional kinematic status (practically defined as velocities < 5 cm yr$^{-1}$) or move as an individual portion from the coherently moving parts, may also contain ground ice but are not taken into account following our homogeneous moving assumption. Moreover, the 1-D InSAR method may fail to detect some slow-moving parts of the landforms creeping nearly in parallel to the flight of satellite due to the poor signal-to-noise ratio, causing possible underestimation of ground ice as well."

Finally, we made modifications to the boundaries (the terminus part in particular) according to those delineated by Jones et al. (2018). We realized that our delineation of the rooting zone was not precise and have modified the rooting part in the revised Fig. 8. We have also changed the legend of Fig. 8 to specify this operation.

2. Uncertainty analysis is probably too optimistic: I partially agree with the second reviewer who said that field data must support this study. However, I will not criticize this fact because there is not always possible to collect field data. So, in this case, the authors must turn on a reliable and compressive uncertainties computation. Is in this part where I have my biggest concern because even if there is a section on how the authors have assessed uncertainties (i.e., Section 5.3 and 5.5) this section remains too vague and qualitative, instead of quantitative. By applying Azocar and Brenning, 2010 methodology, very high uncertainties are obtained from this empirical relationship and those, are not fully understood/analyzed in the manuscript. Later on, in the new version of the manuscript, you mention that "as suggested by Wagner et al., 2021, you subtract 10 meters on the initial computed thickness", but it seems very delicate to me to subtract "10 meters" (i.e. to avoid overestimation) knowing previously that average rock glacier thickness is "30 ± 3 meters" (Cicoira et al., 2020). By doing quick calculations using those mentioned values, you have an uncertainty of 1/3 of the rock glacier thickness (±30%) without considering the uncertainties on the physical assumptions (i.e. simplified model), and surface velocities (partially well known) and ice/water content (poorly known).

The ice and water storage will depend on rock glacier thickness, thus, this is a critical factor in the equation, which is assessed with ambiguity. The authors do not explain how they deal with this source of uncertainty and assume an error of ±2m as coherent, but observations in the European Alps have shown that sometimes is even more (i.e., ±5-7m), so the question is, how much will impact when water storage is computed? As I said before, as no field data is available, a clear, replicable and coherent uncertainty analysis must be present to strengthen the analysis and support the results.

Re: We understand the concern the reviewer raised about the uncertainty of thickness derivation. First, we would like to clarify the method we adopted for thickness derivation. In Sect. 5.3 (L698), we stated that "Wagner et al. (2021) suggested an adapted relationship by subtracting 10 m from the derived thickness to remove the likely overestimation effect" only for reviewing previous discussion on this uncertainty, yet we did not use their adapted relationship. We specified that we used the "classical thickness-area relationship" at L701.

We followed the classical relationship because we did not observe significant overestimation in thickness of the validation rock glaciers. Table S2 presents the comparison results and explains why we

estimated an error of ~2 m. The suggested uncertainty level (~30%) based on Wagner et al. (2021) is not applicable to this work.

**Table S2. Estimated rock glacier thickness ($T_{area}$) derived from the thickness–area relationship used in this study, and the corresponding bias relative to in situ measured thickness ($T_{ref}$) (Barsch et al., 1979; Cicoira et al., 2019a; Arenson et al., 2002; Hoelzle et al., 1998). The rock glacier thickness ($T_{slp}$) derived from thickness–slope angle relationship proposed by Cicoira et al. (2020), and the associated bias. The last row gives the mean absolute error (MAE) derived from the two methods.**

| Rock glacier | $T_{area}$ (m) | $T_{slp}$ (m) | $T_{ref}$ (m) |
|---|---|---|---|
| Murtèl-Corvatsch | 29 | 26.2 | 27 |
| Muragl | 24 | 19 | 20 |
| Schafberg | 24 | 20.8 | 25 |
| MAE | 2.3 | 2 | – |

As the thickness is derived from surface area, the different bias obtained, i.e., 10 m (Wagner et al., 2021), 5–7 m (suggested by the reviewer), 3 m (Cicoira et al., 2020), and 2 m (validation result in this work), could be attributed to the different area delineation, which is an issue raised in the next comment and also by the #3 reviewer. In this revision, we have reported more details about rock glacier delineation in Methodology at L229:

"To derive the input parameters, we first outlined the boundaries of the three rock glaciers from Google Earth images (September of 2018), from which their shapes and areal extents can be extracted using Geographic Information System tools. As Muragl and Schafberg rock glaciers consist of multiple or overlapping lobes, we focus on a single active lobe of each rock glacier where the borehole is present and composition data are available."

To facilitate a "clear, replicable and coherent uncertainty analysis" suggested by the reviewer, we have incorporated the uncertainty of the area parameter into the error propagation, as detailed in the response to the next piece of comment.

Here we have also analysed the uncertainty of ice estimation given that the absolute error of thickness is 6 m (as the average of 5–7 m) or 10 m (Fig S7 and S8; Table S3). The uncertainties associated with the two scenarios are 12% and 13%, respectively. The codes will be open-access at GitHub upon publication of the manuscript. We have described the uncertainty experiment in Sect. 5.3 at L703:

"In the validation part, we estimated the thickness-related error by considering the uncertainty involved in delineating the rock glacier area based on Google Earth images, which derives from the occurrence of different image quality and the contrasting interpretations by different operators due to the complex morphology of rock glaciers (Brardinoni et al., 2019; RGIK, 2020; Schmid et al., 2015; Way et al., 2021). We assumed a 40% uncertainty in the area parameter, leading to a ~10% error (or an absolute error of 2–4 m) in thickness. In addition, we conducted analysis assuming a more significant thickness error according to previous studies (Cicoira et al., 2020; Wagner et al., 2021), i.e., 6 m and 10 m, and obtained errors in ice content of 12% and 13%, respectively, which are greater than the 8% uncertainty in our results (Fig. S7 and S8; Table S3)."

[Figure]

**Figure S7: Modelled relationships (grey shaded areas) between the ice fraction ($\theta_{i,core}$) and the surface velocity ($u_s$) of 95% confidence intervals for the three RGs monitored in the PERMOS network assuming a thickness error of 6 m. The ranges of the observed velocities (yellow bands) are used as velocity constraints for inferring ice content from the modelled relationships. Also shown are the reference ice content obtained from previous field-based surveys (blue lines). The inference ice contents are the mean values (solid black lines) with the estimated ranges (dash-dotted black lines).**

[Figure]

**Figure S8: Similar to Fig. S7, but showing results with a thickness error of 10 m.**

**Table S3. Summary of the reference and inference ice contents derived from two scenarios assuming different thickness errors, namely 6 m and 10 m. The values in brackets following the inference ice contents give the corresponding bias from the reference ice contents. The last row presents the root mean square error (RMSE) of the two scenarios.**

| Rock glacier | Reference (%) | Inference and bias (%) | |
| --- | --- | --- | --- |
| | | 6-m thickness error | 10-m thickness error |
| Murtèl-Corvatsch | 85 | 66 (-19) | 66 (-19) |
| Muragl | 50 | 60 (10) | 61 (11) |
| Schafberg | 65 | 62 (-3) | 63 (-2) |
| RMSE | – | 12 | 13 |

Another problem that I have seen is the fact that Azocar and Brenning, 2010 methodology is based on the rock glacier area delineation. However, a lot of ambiguity is present when delineating rock glacier borders. This is visible in your results, for example, the Kala-Patthar, kongma rock glacier appears to be better delineated (i.e. clear distinction between headwall and rock glacier) than the Tobuche rock glacier (with a straight line in the rooting zone). This is a recurrent problem when rock glacier area is

estimated even in the European Alps (please refer to the new IPA guidelines; https://www.unifr.ch/geo/geomorphology/en/research/ipa-action-group-rock-glacier/). The authors have not analyzed how much will impact if a different area is taken into account.

Re: In fact, we have investigated the impact of different areas on the ice estimation by analysing the model sensitivity, and found that the varying parameter of surface area does not significantly affect the prediction result (Sect. 4.3).

Considering the importance of thickness in controlling the rock glacier movement, we have calculated the thickness error introduced by the area parameter. The codes will be open-access at GitHub upon publication of the manuscript.

In Sect. 3.3 at L236:

"We assigned a relative uncertainty of 40% to the area parameter and considered the propagated error to the final modelling result."

We have updated the validation and prediction results (Fig. 5, S2, S3, 9; Table 3 and 5):

[Figure]

**Figure 5: Modelled relationships (grey shaded areas) between the ice fraction ($\theta_{i,core}$) and the surface velocity ($u_s$) of 95% confidence intervals for the three RGs monitored in the PERMOS network with model parameterisation Scheme 2. The yellow bands show the observed surface velocities, and the blue lines denote the reference ice contents. For each rock glacier, the intersection between the simulated $\theta_{i,core}$- $u_s$ relationship (grey shaded area) and the observed velocity (yellow band) gives the estimated range of ice content, as marked by the dash-dotted black lines. We take the estimated average as the inferred ice content and show the value by the solid black line.**

[Figure]

**Figure S2: Modelled relationships (grey shaded areas) between the ice fraction ($\theta_{i,core}$) and the surface velocity ($u_s$) of 95% confidence intervals for the three RGs monitored in the PERMOS network with model parameterisation Scheme 1. The ranges of the observed velocities (yellow bands) are used as velocity constraints for inferring ice content from the modelled relationships. Also shown are the reference ice content obtained from previous field-based surveys (blue lines). The inference ice contents are the mean values (solid black lines) with the estimated ranges (dash-dotted black lines).**

[Figure]

**Figure S3: Similar to Fig. S4, but showing results derived from model parameterisation Scheme 3. The grey shaded areas outline the modelled relationships between the ice fraction ($\theta_{i,core}$) and the surface velocity ($u_s$) with 95% confidence intervals. The yellow bands show the observed surface velocities, and the blue lines denote the reference ice contents. For each rock glacier, the intersection between the simulated $\theta_{i,core}$-$u_s$ relationship (grey shaded area) and the observed velocity (yellow band) gives the estimated range of ice content, as marked by the dash-dotted black lines. The inferred ice content is taken as the average value of the estimated range and indicated by the solid black line.**

**Table 3. Summary of the reference and inference ice contents derived from the three model parameterisation schemes. The values in brackets following the inference ice contents give the corresponding bias from the reference ice contents. The last row presents the root mean square error (RMSE) of the schemes.**

| Rock glacier | Reference (%) | Inference and bias | | |
| --- | --- | --- | --- | --- |
| | | Scheme 1 (%) | Scheme 2 (%) | Scheme 3 (%) |
| Murtèl-Corvatsch | 85 | 91 (6) | 74 (–11) | 79 (–6) |
| Muragl | 50 | 56 (6) | 59 (9) | 66 (16) |
| Schafberg | 65 | 79 (14) | 68 (3) | 76 (11) |
| RMSE | – | 9 | 8 | 12 |

[Figure]

**Figure 9: Modelled relationships between the ice fraction ($\theta_{i,core}$) and the surface velocity ($u_s$) of 95% confidence intervals for the five RGs in Khumbu Valley with model parameterisation Scheme 2 (grey shaded areas). The ranges of the InSAR-derived velocities (yellow bands) are used as the velocity constraints for inferring ice contents from the modelled relationships. The upper and lower bounds of the estimated ice contents are within the range outlined by the dash-dotted black lines and the solid black lines show the mean values representing the inference ice contents.**

**Table 5. Modelled average ice contents, as well as the minimum and maximum estimates (in brackets) of rock glaciers in Khumbu and Lhotse Valleys and the corresponding water volume equivalents.**

| Rock glacier | Inference ice content (%) | Water volume equivalent (million m$^3$) |
|---|---|---|
| Kala-Patthar | 70±8 | 1.4±0.2 |
| Kongma | 72±8 | 1.5±0.2 |
| Nuptse | 74±8 | 5.9±0.6 |
| Lingten | 74±8 | 2.0±0.2 |
| Tobuche | 74±8 | 2.7±0.3 |

3. The extrapolation of key parameters. For me, it is very delicate to extrapolate ice-content values from Andean mountain ranges to Asia mountain ranges on one hand, because no comparison between precipitation/temperature ranges has been done. Temperature and precipitation will play major roles in the ice content and water availability. These conditions are completely different in an arid region (i.e. the Andes) and high mountain Asia.

Re: In fact, we considered the temperature factor, i.e., the thermal status of permafrost in the Las Liebres rock glacier and the rock glaciers in our study area. We drew the comparison because both are in the warm permafrost status. We introduced this precondition at L141:

"In this study, we used a constant effective viscosity (B) to develop an empirical formula to describe the deformation behaviour of rock glaciers in a warm permafrost environment (> -3°C) … This warm ground condition is likely to be realistic in our study area (Sect. 2)."

The analysis of ground conditions in our study area was given at L88:

"…we infer that these rock glaciers develop in a warm permafrost environment for the following reasons: (1) the landforms are located near or below the altitudinal limit of permafrost distribution in Nepal (Fujii and Higuchi, 1976; Jakob, 1992), indicating that the local environment is at the critical limit of permafrost occurrence; (2) based on empirical relationships between mean annual ground temperature (MAGT), mean annual air temperature, latitude, and altitude, the estimated MAGT is >0.5°C, which suggests that permafrost in this area is in a warm and unstable state (Nan et al., 2002; Zhao and Sheng, 2015)."

Second, we agree that precipitation plays an important role in controlling the ice content and water availability of rock glaciers. The water input mainly contributes to the seasonal variations of rock glacier velocities (Kenner et al., 2017; Cicoira et al., 2019), yet our method focuses on the multi-annual kinematic status and neglected the short-term behavior. We claimed our focus in the Methodology section at L153:

"However, the short-term rock glacier kinematic patterns are irrelevant to this study focusing on modelling the relationship between ice content and multi-annual average movement velocity in our study."

Regarding the ice content, the Andes rock glacier indeed has different ice content (42%-82%, according to Monnier and Kinnard (2016)) from the rock glaciers in our study area (70%–74%), which might be attributed to the different precipitation condition.

**Reference**

Cicoira, A., Beutel, J., Faillettaz, J. & Vieli, A. (2019). Water controls the seasonal rhythm of rock glacier flow. *Earth and Planetary Science Letters*, *528*, 115844. https://doi.org/10.1016/j.epsl.2019.115844

Kenner, R., Phillips, M., Beutel, J., Hiller, M., Limpach, P., Pointner, E. & Volken, M. (2017). Factors controlling velocity variations at short-term, seasonal and multiyear time scales, Ritigraben rock glacier, Western Swiss Alps. *Permafrost and Periglacial Processes*, *28*(4), 675–684. https://doi.org/10.1002/ppp.1953

Monnier, S. & Kinnard, C. (2016). Interrogating the time and processes of development of the Las Liebres rock glacier, central Chilean Andes, using a numerical flow model. *Earth Surface Processes and Landforms*, *41*(13), 1884–1893. https://doi.org/10.1002/esp.3956

4. From my experience, I do not think that the surface velocity is the best parameter to determine ice/water content. Following your stated hypothesis, the velocity increase has a direct relationship with the ice content. However, a generalized increase in creep rates has been observed recently in the European Alps, but it does not imply an increase in ice content.

Re: The same concern has been raised by Dr. Lukas Arenson in the previous round of review. In brief, the phenomena (increased velocity indicates increased ice content) described here cannot be deduced from our work, because the relationship between velocity and ice content in our model in non-linear. Moreover, our approach is designed for estimating current ice content by assuming the amount of ground ice remain constant within the past 1–2 decades.

We presented a discussion in Sect. 5.1 with an example for better illustration in Fig. S6:

"**5.1 Incapability of predicting ground ice evolution**

Our results were presented in the form of a modelled relationship between the ice content and surface velocity (as shown by the grey shading in Fig. 5, S2, S3 and 9), which might mislead the users to interpret the ground ice evolution from rock glacier kinematic variations. For instance, assuming the surface velocity of Kala-Patthar rock glacier reaches 1 m yr$^{-1}$, the corresponding ice fraction would be approximately 60% (detailed in Fig. S6 in the supplement material). However, we cannot draw the conclusion that ground ice stored in Kala-Patthar rock glacier would decrease by 10% if it accelerated to 1 m yr$^{-1}$, because the geometric parameters of the landform would change accordingly, particularly the thickness of the permafrost core and the active layer, making the current modelled relationship no longer valid.

In the proposed approach, we assume that the amount of ice stored in rock glaciers remain constant within the timescale of our study (1–2 decades, constrained by InSAR data), which is consistent with the fact that rock glaciers are currently not a major contribution to surface runoff (Duguay et al., 2015; Jones et al., 2019b). Predicting ground ice changes from kinematic variations is beyond the applicability of our model."

[Figure]

**Figure S6: Modified after Fig. 11a in the manuscript. The yellow shading shows the observed surface velocity (~0.1 m yr$^{-1}$) and the vertical solid black line denotes the modelled ice content (71%). The red shading marks an assumed surface velocity (1 m yr$^{-1}$) and the estimated ice fraction is shown by the vertical dotted line (60%).**

Finally, I strongly encourage the authors to re-evaluate this approach. Estimate ice/water content in a rock glacier is a very difficult task, which uncertainties should be estimated properly and supported with field observations if it is possible.

Best regards

Re: We thank the reviewer again for taking time to review this manuscript and providing valuable comments, especially regarding an enriched uncertainty presentation, which greatly helps improve our work.

Report #3

The manuscript deals with the modelling of rock glacier ice content based on InSAR-derived surface velocity in 5 active rock glaciers of Khumbu and Lhotse Valleys, northeastern Nepal. These estimates rely on empirical ice content and kinematic data drawn from three rock glaciers in the Swiss Alps. Modelled ice content in the five rock glaciers of interest are then applied to an existing inventory of active rock glaciers in Nepalese Himalaya.

I have read with interest the reviewers' comments, the authors' responses and the revised manuscript. The authors have done an excellent job in their point-by-point replies. The revised manuscript shows an extensive effort made to address all of the reviewers' concerns. Although some of the main objections raised by Dr. Arenson remain unsolved, the revised/rewritten discussion acknowledges most of the limitations adequately. In this regard, the upscaling procedure to estimate water storage from the five study rock glaciers to the entire Nepalese Himalaya represents quite a leap, and therefore inherent uncertainties could be described in a more explicit and systematic way.

Re: We thank the reviewer for the constructive suggestions, especially regarding an extended description of the delineation-related uncertainties, which we believe considerably help improve the quality of the manuscript. We consider these comments carefully and provide our point-by-point responses given below. The line numbers refer to the ones in the revised manuscript with track changes, aiming to help the reviewer and editors locate the revisions made correspondingly.

1.  In particular, the authors could enrich their state-of-the-art by adding reference to recent work on the uncertainties involved in the compilation of rock glacier inventories on optical imagery, and on Google Earth (GE) in particular: (e.g., Schmid et al., 2015; Jones et al., 2018b; Brardinoni et al., 2019; Way et al., 2021). Uncertainty derives from: (i) the spatial resolution of optical imagery and cloud cover, which in GE vary greatly across a given region; (ii) the mapper (experience, training and personal interpretation); (iii) rock glacier typology (e.g., lobate, tongue-shaped, and multilobe polymorphic).

    Uncertainty applies to: (1) identification of rock glaciers; (2) delineation of rock glacier outline, whose inter-operator variability will affect the rock glacier area, hence the estimated ice/water content; and (3) dynamic classification of the rock glacier (active, inactive and relict), which will affect the number of rock glaciers for which ice/water content is estimated (Brardinoni et al., 2019; Way et al., 2021). Variability in point 2 between mappers has been shown to vary greatly depending on rock glacier type. Uncertainty in point 3, including inter-operator variability, can be reduced greatly by incorporation of InSAR-based kinematic attribute, following a protocol tested on 11 regions across the globe (Bertone et al., 2021).

    Bertone, A, Barboux, C, Bodin, X, Bolch, T, Brardinoni, F, Caduff, R, Christiansen, H H, Darrow, M, Delaloye, R, Etzelmüller, B, Humlum, O, Lambiel, C, Lilleøren, K S, Mair, V, Pellegrinon, G, Rouyet, L, Ruiz, L, and Strozzi, T. Incorporating kinematic attributes into rock glacier inventories exploiting InSAR data: preliminary results in eleven regions worldwide. The Cryosphere Discuss. [preprint], https://doi.org/10.5194/tc-2021-342

    Brardinoni F, Scotti R, Sailer R, and Mair V. 2019. Sources of uncertainty and variability in rock glacier inventories. Earth Surface Processes and Landforms, 44, 2450-2466.

    Jones et al 2018b (already in reference list)

Schmid MO, Baral P, Gruber S, Shahi S, Shrestha T, Stumm D,Wester P. 2015. Assessment of permafrost distribution maps in the Hindu Kush Himalayan region using rock glaciers mapped in Google Earth. The Cryosphere 9(6): 2089–2099.

Way RG et al., 2021 Consensus-Based Rock Glacier Inventorying in the Torngat Mountains, Northern Labrador. American Society of Civil Engineers Proceedings. Regional Conference on Permafrost and the 19th International Conference on Cold Regions Engineering. https://doi.org/10.31223/X5C60W

Re: We have introduced the uncertainties associated with area delineation in the Methodology and Discussion. We have also considered the impact of the area uncertainty on the modelling result. More details are summarized in the response to comments from the #2 reviewer.

At L229:

"To derive the input parameters, we first outlined the boundaries of the three rock glaciers from Google Earth images (September of 2018), from which their shapes and areal extents can be extracted using Geographic Information System tools. As Muragl and Schafberg rock glaciers consist of multiple or overlapping lobes, we focus on a single active lobe of each rock glacier where the borehole is present and composition data are available. The three rock glaciers for validation have a tongue-shaped typology."

At L700:

"In the validation part, we estimated the thickness-related error by considering the uncertainty involved in delineating the rock glacier area based on Google Earth images, which derives from the occurrence of different image quality and the contrasting interpretations by different operators due to the complex morphology of rock glaciers (Brardinoni et al., 2019; Schmid et al., 2015; Way et al., 2021). We assumed a 40% uncertainty in the area parameter, leading to a ~10% error (or an absolute error of 2–4 m) in thickness. In addition, we conducted analysis assuming a more significant thickness error according to previous studies (Cicoira et al., 2020; Wagner et al., 2021), i.e., 6 m and 10 m, and obtained errors in ice content of 12% and 13%, respectively, which are greater than the 8% uncertainty in our results (Fig. S7 and S8; Table S3)."

2. With reference to the five rock glaciers in Khumbu and Lhotse Valleys, and the three rock glaciers from Switzerland, please consider adding an attribute in Tables 1 and 4 to characterize rock glacier typology (e.g., talus lobate, debris tongue-shaped, or others) so that the reader can compare area, width, slope, but also typology. Perhaps you could acknowledge briefly that the three rock glaciers in Switzerland (lines 60-61) are substantially smaller than the five selected in Nepal.

Re: We have included the typology information in the description, as all the features concerned are tongue-shaped. At L232:

"The three rock glaciers for validation have a tongue-shaped typology."

We have also mentioned the extent contrast between the two groups of rock glaciers at L601:

"The geometric and structural data used as input parameters are detailed in Table 4. The five rock glaciers are tongue-shaped features and their areal extents are substantially larger than the three validation rock glaciers (Table 1 and 4)."

3. Since you are extrapolating your modelling results to Nepalese Himalaya, please consider: (i) justifying briefly the selection of those valleys and the five rock glaciers in particular; (ii) describing where the average size of your five rock glaciers plots (percentile) within the size distributions of rock glaciers across Nepalese Himalaya. The latter would allow the reader to understand where the five sample rock glaciers stand compared to the regional population.

Re: We selected the Khumbu and Lhotse valleys as the study region mainly because the rock glaciers and frozen ground status in this area are among the best studied in the Nepalese Himalaya. We have also conducted field investigations in the Khumbu valley (see Knight et al., 2019), making it possible for in-situ investigations for validating our results in the future.

Knight, J., Harrison, S. & Jones, D. B. (2019). Rock glaciers and the geomorphological evolution of deglacierizing mountains. *Geomorphology*, *324*, 14–24. https://doi.org/10.1016/j.geomorph.2018.09.020

We have analyzed the size distribution of the rock glaciers in the Nepalese Himalaya (Fig. S7), and added descriptions at L746:

"Second, the dimensional extent of rock glaciers varies across the Nepalese Himalaya (Fig. S9). Considering the surface areas of rock glaciers and the thickness–area relationship, the volumes of the landforms lie between 0.08 million $m^3$ and 228 million $m^3$. The dimensions of Kala-Patthar, Kongma, Nuptse, Tobuche, and Lingten rock glaciers are at the 26th, 27th, 35th, 50th, and 72nd percentiles of the regional population (Jones et al., 2018), respectively, and cannot represent the sizes of all rock glaciers across the mountain range."

[Figure]

**Figure S9: Box and whisker plot showing the statistical distribution of rock glacier volumetric dimensions in the Nepalese Himalaya (Jones et al., 2018b).**

Jones, D. B., Harrison, S., Anderson, K., Selley, H. L., Wood, J. L. & Betts, R. A. (2018). The distribution and hydrological significance of rock glaciers in the Nepalese Himalaya. *Global and Planetary Change*, *160*, 123–142. https://doi.org/10.1016/j.gloplacha.2017.11.005

4. InSAR methodology: please describe how movement along LOS was projected to the line of maximum slope, adding relevant reference (e.g., Bechor, NB and Zebker, HA. 2006. Measuring two-dimensional movements using a single InSAR pair. Geophysical Research Letters, 33(16)). Please specify whether the projection was conducted systematically or was limited to pixels with

slope below a given threshold. When the angle between LOS and line of maximum slope is high (>60°), projecting may amplify InSAR related errors.

Re: We have added citations and described the factors considered in the reprojection procedure at L292:

"The projection was conducted considering the flight direction of satellite, the local incidence angle, and the landform morphologic parameters including the aspect and slope angles (Massonnet and Feigl, 1998; Bechor and Zebker, 2006)."

The reprojection equation is given below:

$$V_{slp} = \frac{V_{LOS}}{\sin(\theta_{asp} - \alpha)\sin\theta_{inc}\cos\theta_{slp} + \cos\theta_{inc}\sin\theta_{slp}}$$

where $\alpha$ is the flight direction of the SAR satellite; $\theta_{inc}$ is the local incidence angle; $\theta_{asp}$ and $\theta_{slp}$ are the aspect angle and slope angle of the lobe, respectively.

We applied the projection to all pixels given that no steep slope occurring on the five rock glaciers in our study area.

Other minor comments:

1. Line 434: please consider adding the following reference:

   Scotti R, Crosta G B, and Villa A. 2017. Destabilisation of Creeping Permafrost: The Plator Rock Glacier Case Study (Central Italian Alps): The Destabilised Plator Rock Glacier. Permafrost and Periglacial Processes, 28(1), 224–236.

   Re: Added.

2. Lines 439-440: please consider removing the following sentence: "Rock glaciers showing strong subsidence indicators from optical images, such as surface depressions or cracks, are not suitable for the current method". It defeats the purpose of using InSAR data. Interpretation of vertical surface deformation (e.g., subsidence) based on morphologic features observed on optical images is unreliable and potentially misleading.

   Re: We have removed this sentence as suggested.

3. Line 450: please consider modifying the citation of the IPA report, currently referred to as "Delaloye & Echelard, 2020", with the "How to cite" indication contained in the updated version of the document: "RGIK, 2021" and in the reference list as: "RGIK (2021). Towards standard guidelines for inventorying rock glaciers: baseline concepts (version 4.2.1). IPA Action Group Rock glacier inventories and kinematics (Ed.), 13 pp.". This effort involved a broad international working group.

   Re: We have updated the citation to the correct format.

I enjoyed reading the thread of revisions and look forward to seeing the paper published.

Re: We thank the reviewer again for the encouragement and the effort for making the manuscript better.

---

## Author Response (AR3)

**Report #1**

Dear Editor, dear authors,

I think the manuscript has improved substantially from the last version and I am happy with the changes made at this time. Please consider some minor changes aimed at improving the readability of the following sentences.

Re: We thank the reviewer for the encouraging comment and the substantial help we received to improve the quality of the manuscript through the revision process. Thank you.

In particular, please consider revising as follows:

1. L 209: "As Muragl and Schafberg rock glaciers consist of multiple and/or overlapping lobes, in each of them we focus on a single active lobe for which borehole and composition data are available. The three rock glaciers selected for validation are tongue shaped."

Re: Changed.

2. L 292: "We applied the projection to all pixels given that no steep slope occurs on the five rock glaciers of interest.

Re: We have added the sentence.

3. L 700: "... which derives from interactions among variable image quality, the operator's mapping style and interpretation, and the complexity of the rock glacier morphology (Brardinoni et al., 2019; Schmid et al., 2015; Way et al., 2021)."

Re: Changed.

**Report #2**

Third review to the article from Hu et al. 'Modelling rock glacier ice content based on InSAR-derived velocity, Khumbu and Lhotse Valleys, Nepal':

I would like again to acknowledge the extensive work performed by the authors to address my comments from my previous review. Although I do agree with some critical comments from the other reviewers regarding the methodology, I believe the current version is now acceptable for publication after minor adjustments. The research raised several questions about the feasibility/relevance of coupling remote sensing observation and modelling, quantifying the ice content based on this approach and extrapolating the results to an entire region. However, these questions are now well discussed, and the limitations are acknowledged. Independently of if we agree or not with the assumptions, I believe the procedure and resulting outputs are now clearly described and therefore worth publication. Generating debates in the community is also what open access dissemination is about.

Re: We are grateful to the reviewer for the valuable feedbacks we received to improve the manuscript in many aspects through the review process. Thank you.

I have focused the work for this third review on the main parts that I commented in my second review, suggesting some minor corrections to facilitate the understanding of the content. Note that the line numbers refer to the version without track changes.

1. l.232-234: "... we present our method to measure surface velocities of rock glaciers with InSAR for constraining the model (Sect. 3.5.1) and use complementary remote sensing products to derive geometric and structural parameters (Sect 3.5.2)."

Re: Changed.

2. l.248-249: "multi-looking operation and adaptive Goldstein filter (8x8 pixels) were applied using the open-source software..."

Re: Changed.

3. l.250: "The final georeferenced interferograms have a ground resolution..."

Re: Changed.

4. l.252: "Ice-debris landforms"? Is there a reason not using "rock glaciers" here? If it also considers debris-covered glaciers, just say it so.

Re: Yes, we did also apply the same procedure to the Chola debris-covered glacier. Relevant data is presented in the Supplementary Materials.

5. l.261: "For each pixel, we found the velocity error is < 10 cm/yr". Rather used the way you explained it your response to my questions, it is much clearer: "For each interferogram, we quantified the uncertainty at the pixel-level. Among all the high coherent pixels, the largest uncertainty is 9.8 cm/yr. The velocity error is therefore considered as < 10 cm/yr."

Re: Agree. We have changed the phrasing.

6. l.262-265: Still unclear to me if the criteria are applied to discard interferograms, entire or part of the landforms. Based on what you explained in your response, I would suggest writing: "... we selected the interferograms and documented rock glacier parts meeting the following criteria... (1) only pixels showing acceptable coherence (> 0.3) are kept; (2) the coherent pixels must cover more than 40% of the landform surfaces; (3) the mean velocity must be larger than 5 cm/yr (Wang et al., 2017). We set this empirical threshold considering the typical noise level from atmospheric delays (5 cm/yr). The interferograms and landforms that do not meet these criteria were discarded." If I misunderstood: please adjust the content and make it clear.

Re: We have changed the sentence according to the suggestion.

7. l.271: "After the procedure described in Step 2, for each selected landform, ..."

Re: Changed.

8. l.274-276: "... in more than half of the interferograms, the pixel was included in the coherently moving part... Otherwise, the pixel is discarded, i.e. not included in the coherently moving part. The area is considered as inactive or in a transitional kinematic status."

Re: Changed.

9. l.279: Again, it is better explained in the response to my question: "the mean velocity error is the square root of the quadratic sum of all the velocity errors, which is limited to < 1 cm/yr". You could btw consider referring here to your github codes (basically referring to the code and data availability section) to make it simple to look for answers about how it has been calculated.

Re: Changed.

10. l.280: "... the range of the spatially averaged velocities within the coherently moving parts... By doing so, isolated patterns are neglected assuming that they may be related to short-term kinematic fluctuations, not representative of the multi-annual kinematic behaviour of the whole landform."

Re: Changed.

11. l.291: "...the abnormal value in 2015 has been removed from the range"

Re: Changed.

12. l.350: "... (Sect 4.4.1) and present the modelled ice content of the five rock glaciers..."

Re: Changed.

13. l.418: "consistent with the fact that rock glaciers are currently not a major contribution to surface runoff in the study area"

Re: Changed.

14. l.450: "the uncertainty in deriving rock glacier thickness remains ambiguous" could be rephrased. I don't think "ambiguous uncertainty" means anything.

Re: We have changed the sentence: "…the uncertainty of in deriving rock glacier thickness remains challenging to accurately quantify…"

15. l.455: One thing that is missing as limitation in this part: when considering your definition of coherently moving area, you discard quite big parts of the morphologically delineated landforms, which raises the question: how representative is the resulting InSAR average when it is based on data covering less than a quarter the rock glacier surface (for ex. Fig.8b)? As you acknowledge in your answer to my previous questions, it can be both due to low-coherence (potentially due to high-velocity) or low-velocity. It basically means that you have a big uncertainty in both directions: the results may be underestimated or overestimated. Good to mention the problem, I think.

Re: We thank the reviewer for raising this point. It can lead to either underestimation or overestimation issue if we attempt to use the ice content (%) of the coherently moving area for representing the percentage of ice in the whole landform. However, we do not intend to do so. Here we only aim to discuss the bias when estimating the amount of ice stored in rock glaciers, which would always be an underestimation problem. As we stated in L. 574:

"This part may also contain ice but are excluded from our estimation, causing possible underestimation of ground ice storage as well."

16. l.458: Avoid using "stable" / "stability" when speaking about moving landforms. "at a relatively constant rate."

Re: Changed.

17. l.461: Same here "the stability of the motion". Maybe "to consider an average rate and avoid misleading conclusions based on unrepresentative short-term patterns".

Re: Changed.

18. l.463: "... or behave differently from the coherently moving parts"

Re: Changed.

19. l.467: "Third, rock glaciers affected by significant subsidence (instead or in addition to downslope creep) cannot be..."

Re: Changed.

20. l.471: "Finally, ..."

Re: Changed.

21. l.496: "The likely emergence of... will likely allow for improving the accuracy of the approach". "We expect the improved model can be valuably applied to..."

Re: Changed.

22. l.505: "(2) Mean downslope velocities in the coherently moving part of the rock glaciers in..." I think it is important to specify it here – that is the mean of what you define as coherent, not the entire morphologically-delineated rock glaciers.

Re: Agree. Information added.

23. l.507: "remained constant" – wise to do a "find&replace" function for the entire document cause using "stable" is quite misleading.

Re: We have changed the wording consistently throughout the manuscript.

24. l.516: Maybe "confirms" instead of "highlights"?

Re: Changed.

Review of the manuscript tc-2021-110

Modelling rock glacier ice content based on InSAR-derived velocity, Khumbu and Lhotse Valleys, Nepal

By Hu Yan et al.

**General remarks**

The authors present a novel method to estimate the ice volume in rock glaciers based on a modified ice flow model and InSAR derived surface velocity. The model is calibrated using literature data from a rock glacier in the Andes, validated at rock glaciers in the Swiss Alps where detailed information is available and then applied to five rock glaciers in Khumbu Himal. Finally the authors upscale their results to the whole Nepalese Himalaya based on an existing inventory using a scaling relationship. The topic is of high importance as the ice contained in rock glacier could potentially be of hydrological importance, measurements can only be done on a very limited number of rock glaciers and therefore a modelling approach could provide valuable information.

This paper has gone through few round of reviews and even though all reviewers are in line that work is in general if high interest there seems still to be some concerns which could not be fully addressed. I won't recall the available reviews but provide an independent opinion based on the current version but acknowledging that the manuscript has clearly improved.

Re: We thank the reviewer for the constructive and insightful comments which would significantly improve the quality of the work. We consider these comments thoroughly and revise the manuscript correspondingly. In this letter, we provide the point-by-point responses with line numbers referring to the ones in the revised manuscript with track changes, aiming to help the reviewer and editors locate the revisions we made correspondingly.

**General comments:**

1. The main focus of the study is to estimate the ice content of few rock glaciers in Khumbu Himal using a novel methodology. I know that the authors cannot change now this setup, but it is in general questionable to develop a new method based on one rock glacier (which is quite long and narrow) calibrate it on others and then apply is to rock glaciers with different characteristics in a region with different climate and topographic settings. Hence, it remains unclear how well this information can be transferred. The authors must more convincingly show this, e.g. by providing more detailed information about topographic, climatic and ground temperature conditions of the different regions. I understand that the authors maim focus is the Himalaya, but why not first develop, calibrate and validate the model on rock glaciers and regions in the Alps or Andes where much more information and also in-situ measurement of rock glaciers are available? If the authors decide to keep this set-up then at least a rational for choosing these rock glaciers need to be given. Las Liebres rock glacier is measured by quite thoroughly by GPR. However, to get better information about the ice content it is in general recommended to combine different geophysical techniques.

   Re: We understand the concern regarding the apparent differences among the rock glaciers in our study, such as their planar shape, the local climate, and the topographic settings. However, these factors are irrelevant in our model setup, but play a role by adjusting the ground temperature. Therefore, we have taken the reviewer's suggestion (in this comment and the next) to further elaborate the rationale behind the selection of rock glaciers by demonstrating their similar ground temperature conditions, i.e., a warm permafrost environment (> -3°C).

   This assumption was introduced in *Sect. 3.1 Model design and assumptions* (L.153):

   "In this study, we used a constant effective viscosity ($B$) to describe the deformation behaviour of rock glaciers in a warm permafrost environment (> -3°C). The empirical formula was developed based on existing observational data and laboratory findings. This warm ground condition is likely to be realistic in our study area (Sect. 2) and occurs in the rock glaciers in the Andes and Swiss Alps selected for model calibration and validation (Sect. 3.2 and 3.3)."

As quoted above, in *Sect. 2 Study area*, we illustrated that the rock glaciers in the Khumbu and Lhotse valleys are situated in a warm permafrost environment (L.108):

"…we infer that these rock glaciers develop in a warm permafrost environment for the following reasons: (1) the landforms are located near or below the altitudinal limit of permafrost distribution in Nepal (Fujii and Higuchi, 1976; Jakob, 1992), indicating that the local environment is at the critical limit of permafrost occurrence; (2) based on empirical relationships between mean annual ground temperature (MAGT), mean annual air temperature, latitude, and altitude, the estimated MAGT is >0.5°C, which suggests that permafrost in this area is in a warm and unstable state (Nan et al., 2002; Zhao and Sheng, 2015)."

We have then added information of the ground thermal conditions of the rock glaciers in the Andes and Swiss Alps of which in-situ data were used in our study for model calibration and validation in *Sect. 3.2 Model calibration* (L.219) and *Sect. 3.3 Model validation* (L.236), respectively:

"Las Liebres rock glacier was considered to have a near 0 °C permafrost temperature (Monnier and Kinnard, 2016), according to the borehole measurement of a nearby rock glacier (Monnier and Kinnard, 2013)."

"All of the selected rock glaciers have warm cores showing permafrost temperatures between -1 and 0 °C (PERMOS, 2019)."

2. The authors should also more clearly present how they define rock glaciers in their study. When the authors first present a definition of rock glaciers it is very general without mentioning of permafrost. But at line 100ff at the beginning of the model they refer to ice- rich permafrost. The also mention the transition to rock glaciers with glacier melt (L40). Is this also true for glacier flowing into regions where permafrost is unlikely? This is important to consider are the presence of permafrost influences melt and the ice flow. In this sense more information about the ground thermal conditions and possible permafrost presence (e.g. by considering available climate measurements and permafrost modelling results) is required for all considered regions.

Re: We have stated the definition of rock glacier adopted in our study. In *Sect. 1 Introduction* (L.25):

"Rock glaciers are valley-floor and valley-side landforms that commonly occur in the periglacial and glacial realm. Intact rock glaciers develop permafrost or glacial ice cores containing varying amounts of ground ice."

We have also introduced the definition at the beginning of the *Abstract* (L.10) according to the Specific Comment #3:

"Active rock glaciers are viscous flow features embodying ice-rich permafrost and other ice masses. They contain significant amounts of ground ice and serve as potential freshwater reservoirs as mountain glaciers melt in response to climate warming."

The reviewer questioned the existence of permafrost considering that glaciers flow into non-permafrost regions. In our opinion, there exists isolated permafrost in the bodies of the transitioning landforms, as long as the ground ice has not been entirely melted out. In that case, permafrost temperature stays at around 0 °C. However, our approach is not applicable to these actively transitioning features because they are probably experiencing rapid changes in ice content (detailed in *Sect. 5.2 Limitations of the model application*). None of the rock glaciers selected in this study are transitioning landforms either.

To clarify this point, we have omitted the discussion of glacier–rock glacier transition in the revised manuscript in response to this comment and Specific Comment #5. We have also provided information of the ground thermal conditions of the rock glaciers in different regions (detailed in response to General Comment #1).

3. The flow modelling seems to be suitable as applied in a similar way by different other studies. One issue as also mentioned in the text is the seasonal variation occurring mainly at the shear horizon which is not captured in the model. The seasonal is according to the available measurements 60-90% of the surface velocity. The authors assume that they neglect the short term variations by taking

*"the range of the spatial mean velocities of the coherently moving parts". This needs to be more convincingly shown. E.g. Can't the authors generate a time series of the velocity and show them?*

Re: Fig. 7 plots the time series of the velocities derived by InSAR, from which we can observe fluctuations in velocities in different seasons. Take Nuptse rock glacier as an example (Fig. 7a): in 2007, the mean velocity in Aug (20 cm/yr) was 30% larger than that in Dec (15 cm/yr), showing the occurrence of seasonal variations in surface movements. However, the absolute values of velocities lie within a narrow range during the multi-year observation window, as reported at L.440:

"…most rock glaciers, except for Tobuche, moved at a nearly constant rate, ranging from 5 cm yr$^{-1}$ to 30 cm yr$^{-1}$ during the observational period…"

In this study, we take the range of the spatially mean velocities within the coherently moving parts as model constraints (as shown by the yellow bands in Fig.7), because they represent the multi-annual kinematic behavior of the whole landform. We have further explained this point at L.349:

"By doing so, isolated patterns are neglected assuming that they may be related to short-term fluctuations, not representative of the multi-annual kinematic behaviour of the whole landform."

4. *In case I understood correctly they transfer the derived velocity for the part they obtained values to the whole landform (L280ff). This is questionable as variable parts of the rock glaciers might have different ice contents and for some rock glaciers they get results for clearly less than 50% of the rock glacier. Moreover, the part of which the authors obtain suitable results varies strongly, e.g. it is the upper part of Kala-Patar, only one side Kongma and the lower part for Tobuche. The only rock glacier for which the approach is reasonable is Nuptse rock glacier.*

Re: We might be unclear about this point. In fact, we do not "transfer the derived velocity…to the whole landform" but focus on the "coherently moving part" of the rock glacier. Some of the landforms indeed develop a small area of the coherently moving part, as the reviewer pointed out. However, we only model the ice content and calculate the corresponding water equivalent of these areas without any extrapolation to the entire rock glacier.

We gave the reasons for doing so at L.322:

"Field observations have revealed that multiple areas moving differentially can occur on rock glaciers and exhibit complex kinematic patterns (e.g., Buchli et al., 2018), which violates the assumption of a continuously moving body (Sect. 3.1, Fig. 3). Therefore, we aim to identify the coherently moving part of the landform that corresponds with our assumption and is thus suitable for model application."

5. *This is also true for the scaling. The authors use a formula based on area which was developed for the Andes by Brenning (2005) and apply it without adjustments to the rock glaciers in the study region and the Nepal Himalaya. This is equally questionable as it is well known that scaling parameters vary and should be calibrated for the specific regions. Also the characteristics of the identified rock glaciers varies clearly. While the Nuptse Rock Glacier has a clearly identifiable tongue Kala-Patar not and has a depression with small lakes (see Figure 1). It is hence likely that the different parts pf the rock glaciers have different ice contents. I suggest that the authors analyse the topography of the rock glaciers and adjust the scaling accordingly or use other suggested approaches.*

Re: Regarding the thickness derivation, firstly, the same empirical relationship has been adopted in previous publications focusing on study areas other than the Andes, such as the Austrian Alps (Wanger et al., 2021) and the Himalayas (Jones et al., 2018, 2021).

Furthermore, we adopted another approach, i.e., the thickness–slope relationship established based on field data gathered in the Alps (Cicoira et al., 2020), made comparisons between the two methods, and found that the two sets of results display the same level of errors (~2 m, Table S2).

We also agree upon the opinion that different parts of rock glaciers have different ice contents. Therefore, this study focuses on the coherently moving parts without any extrapolating to the entire landforms.

**Table S2: Estimated rock glacier thickness ($T_{area}$) derived from the thickness–area relationship used in this study, and the corresponding bias relative to in situ measured thickness ($T_{ref}$) (Barsch et al., 1979; Cicoira et al., 2019a; Arenson et al., 2002; Hoelzle et al., 1998). The rock glacier thickness ($T_{slp}$) derived from thickness–slope angle relationship proposed by Cicoira et al. (2020), and the associated bias. The last row gives the mean absolute error (MAE) derived from the two methods.**

| Rock glacier | $T_{area}$ (m) | $T_{slp}$ (m) | $T_{ref}$ (m) |
|---|---|---|---|
| Murtèl-Corvatsch | 29 | 26.2 | 27 |
| Muragl | 24 | 19 | 20 |
| Schafberg | 24 | 20.8 | 25 |
| MAE | 2.3 | 2 | – |

6. Figure 1: Nuptse and Kala-Patar rock glaciers (Image source: Pleiades from Google Earth)

   1) Another issue is the rock glacier delineation. I know this is quite difficult and also subjective (see e.g. the cited study by Brardionini et al. ) and the delineation is fine for a regional study, but for this localised rock glaciers the authors should make more effort to provide the most precise outlines possible.

   Re: We have consistently updated the boundaries of the landforms in Fig. 1 and Fig. 8.

[Figure]

**Figure 1: (a) Location of the study site; (b) Google Earth images (taken in 2019) showing the spatial distribution of the active ice–debris landforms, including rock glaciers (RG) in red outlines and debris-covered glaciers (DCG) in blue boundaries. The RGs are delineated by Jones et al. (2018) and the DCGs by the authors based on Google Earth images. The termini of Khumbu and Chola DCGs (outlined by dotted lines) are transitioning into rock glaciers (Knight et al., 2019).**

[Figure]

**Figure 8: Velocity field maps show the average velocities of the coherently moving parts of the five rock glaciers (blue outlines) in the study area. The boundaries of the landforms delineated in Jones et al. (2018b) are in red. The transparent areas between the red and blue boundaries are due to low coherence or low velocity during the observational periods.**

2) What about the terminus of Khumbu glacier? Knight, Harisson and Jonas (2019) argue it might be a future rock glacier. Even though I am not fully in line with the argumentation it would be highly valuable to also model the ice content and have some comparison to the rock glaciers (the authors state that they also calculated the velocity of debris-covered glaciers so it can be easily done). The ice core taken close to the more rock glacier part as identified by Knight et al. (2019) might provide some valuable data (Miles et al. (2021). This value could then also nicely compared to the modelled ice content of the identified rock glaciers.

[Figure]

Re: We agree with the reviewer that the terminus of Khumbu glacier is transitioning into a rock glacier. We have also outlined the transitioning part in Fig.1 as suggested in the Specific Comment #10. However, landforms in active transition are unlikely to fulfil one of the requirements of the model, i.e., that the amount of ice remains constant at the decadal timescale, and are therefore beyond the application of our approach.

We explained the reasons in *Sect. 5.2.1 Incapability of predicting ground ice evolution* (L.557):

"In the proposed approach, we assume that the amount of ice stored in rock glaciers remain constant within the timescale of our study (1–2 decades, constrained by InSAR data), which is consistent with the fact that rock glaciers are currently not a major contribution to surface runoff in the study area (Duguay et al., 2015; Jones et al., 2019b). Landforms undergoing rapid changes in ice content and corresponding morphology, such as transitional features from glaciers to rock glaciers, are beyond the applicability of our model."

7. It is not fully clear to me how the uncertainty of the final result was calculated. The uncertainty ranges are much too low considering all the uncertainties. Provide an own section for clarification.

These issues make the regional extrapolation highly uncertain and there is no advance in knowledge compared to the first rough estimate presented by Jones et al. (2018) as cited in the study. If the authors really want to extrapolate they should do so by applying their velocity-based approach to the large region or a subset (e.g. the whole Khumbu Himal) and then compare to the available data.

Several of these issues are discussed in the discussion section acknowledging the uncertainty. This is well appreciated, but does not really make the results more accurate. At minimum I ask the authors to provide the most important information affecting the accuracy in the methods section to that the reader clearly knows limitation before knowing the results and can then better interpret them.

Re: We estimated the uncertainties by comparing the observed and modelled ice content of the validation rock glaciers, as introduced in *Sect. 4.2 Model validation* (L.411):

"We used the RMSE (8%) derived from Scheme 2 to represent the uncertainty of our approach."

We have rewritten the *Discussion* to include an individual section on the *Uncertainties* (Sect. 5.1) where the two major sources of errors are carefully described.

"**5.1 Major uncertainty sources**

[revised manuscript text omitted]

As elaborated above, the most important factor affecting the accuracy of the approach is the amount of calibration data for establishing the viscosity–ice content ($B - \theta_{i,core}$) relationship. We have also taken the suggestion and provided this piece of information in the *Methods Section* at L.224 accordingly:

"The limited amount of calibration data plays an important role in the calculation of the uncertainty associated with our approach (detailed in Sect. 5.1.1)."

Finally, following the reviewer's advice, we have omitted the discussion of the regional extrapolation throughout the manuscript.

**Specific comments**

1. Abstract, general comment: The abstract is missing the information about how the volume the rock glacier and the of ice of the water equivalent was estimated. The volume of the rock glacier is estimated on an existing scaling approach.

   Re: We have added the information at L.20.

2. L22/23: If the model is easily applicable why didn't the authors do so for upscaling? An important prerequisite is also a rock glacier inventory.

   Re: We did not do so because this paper aims to propose a novel approach and providing exploratory findings, rather than to conduct a comprehensive large-scale investigation. We also agree upon the insight that a rock glacier inventory is a prerequisite. We have removed the adverb "easily" in this sentence and clarified the motivation of the study at L.14:

"This study proposes a novel approach for assessing the hydrological value of rock glaciers in a more quantitative way and presents exploratory results focusing on a small region."

3. L25ff: The authors need to extend their definition of rock glaciers, state the relation to permafrost and also move the information given in L100 to here.

Re: We added the definition of rock glaciers which also indicates their relationship to permafrost given in L.100 at the beginning of the abstract (L.10).

"Active rock glaciers are viscous flow features embodying ice-rich permafrost and other ice masses."

4. L28: The authors might want to include one or two more citations about other mountain regions where rock glaciers store a significant amount of water.

Re: We have named the geographical regions with citations at L.28:

"Recent research has suggested that they represent important hydrological reservoirs in areas where glaciers are undergoing recession in the face of climate change, such as South America (Azócar and Brenning, 2010; Rangecroft et al., 2014), North America (Munroe, 2018), and South Asia East (Jones et al., 2018a)."

5. L31ff: Can the authors please a bit more specific about debris-covered glacier to rock glacier transition. What about the distal part of Khumbu (Knight et al. 2019)? Please also cite a reference from another research group not only one from the authors.

Re: We have taken the suggestion in the next comment and removed the paragraph discussing the debris-covered glacier.

6. L35f: This is in theory correct, but it is well known that the debris-covered glaciers in the Himalaya lost at least as much mass as debris-free glaciers (due to manyfold reasons incl. reduced ice flux, supraglacial ponds, ice cliffs etc.). Hence, this argument is not valid. Please revise. My general recommendation is to omit this entire paragraph and really focus on rock glaciers and not debris-covered glaciers.

Re: We have removed the paragraph as suggested.

7. L41: I agree that the ratio can be higher if the glaciers melt, but disagree with statement with the transition to rock glaciers as this ice already existed and was considered in models.

Re: We are not sure if we understand this comment correctly. If glaciers transition to rock glaciers, certain amounts of ice transfer from glaciers to rock glaciers. In the ratio $\frac{rock\ glacier\ ice}{glacier\ ice}$, the denominator becomes smaller, and the numerator becomes larger, resulting in a higher ratio.

8. L49ff: These are way to many citations in a row. Please be more specific about the cited papers or remove some.

Re: We have removed some citations.

9. L56: Would be good to mention all relevant factors controlling the movement and then specific the most important ones and then provide more details about the ice content.

Re: We have listed the other first-order factors in this sentence:

"Ice content is one factor controlling the movement of rock glaciers by influencing the driving force and the rheological properties of materials which constitute the permafrost core (Arenson and Springman, 2005a; Cicoira et al., 2020), in addition to other first-order factors including ground temperature, sub-surface structure, debris content, and water pressure (Moore, 2014)…"

10. L70: Please indicate in Fig. 1 the ones which transition to rock glaciers according to the cited references.

Re: We have outlined the termini and specified them in the caption:

"The termini of Khumbu and Chola DCGs (outlined by dotted lines) are transitioning into rock glaciers (Knight et al., 2019)."

11. L74f: The authors should also consider Fukui et al. (2007)

Re: We have added the suggested citation.

12. Fig. 1: The digitisation of the debris-covered glaciers is quite poor. Even though not the focus of the study, this needs improvement.

Re: We have updated the figure.

13. L126f: Please be more specific about the permafrost core, in particular about the water occurrence.

Re: We have specified that the water is "unfrozen water", which is a normal constituent of permafrost.

14. L130: Please check the statement about the high ice content. The shear horizon is mentioned in the cited studies, but no information about the ice content (but maybe I have overseen this). The shear horizon is nicely presented by Cicoira et al. (2021) and they also state here that the ice content is lower than in the ice-rich core.

Re: The reviewer is right. The shear horizon has high debris content instead. We have corrected this.

15. L133: The seasonal variation are first presented by Wirz et al. (2016).

Re: We have added the suggested citation.

16. Fig. 3: The figure is quite similar to the one by Monnier & Kinnard (2016) apart from the fact that the deformation at the front is not shown. I suggest to show also the deformation and also include the shear horizon. I would then also refer to the reference (but add adjusted or similar) as the idea seems to be taken from it.

Re: We have updated the figure and added the reference in the caption.

[Figure]

**Figure 3: Schematic geometry, structure, stress status, and composition of rock glaciers (adapted from Monnier and Kinnard (2016)). The rock glacier consists of a permafrost core underlying the active layer. Parameters involved in the model include surface slope ($\alpha$), active layer thickness ($h_{al}$), thickness of permafrost core ($h_{core}$), driving stress at the base of the active layer ($\tau_0$), driving stress at depth z ($\tau_z$), surface velocity ($u_s$), velocity at depth z ($u_z$). $\theta_{d,al}$ and $\theta_{a,al}$ refer to the debris fraction and air fraction of the active layer. $\theta_{d,core}$, $\theta_{i,core}$, $\theta_{w,core}$, and $\theta_{a,core}$ are the fractions of debris, ice, water, and air in the permafrost core, respectively.**

17. L205f: The authors apply the empirical formula established by Brenning (2005). See my general comment about applying the formula to other regions. How do the derived volumes of the Las Liebres rock glacier and the three rock glaciers in the Swiss Alps compare to measured volumes? Provide this information here.

Re: We conducted a comparison between the measured thickness of the three Swiss rock glaciers and the derived thickness from two different empirical methods (the Andes and the Alps, respectively) in the Supplementary material (Table S2). Las Liebres is not included because we only used its measured thickness in this study. We have also presented a thorough discussion on thickness derivation in Sect. 5.1.2.

The volume can be calculated from thickness and area, so we did not directly validate the volume.

**Table S2: Estimated rock glacier thickness ($T_{area}$) derived from the thickness–area relationship used in this study, and the corresponding bias relative to in situ measured thickness ($T_{ref}$) (Barsch et al., 1979; Cicoira et al., 2019a; Arenson et al., 2002; Hoelzle et al., 1998). The rock glacier thickness ($T_{slp}$) derived from thickness–slope angle relationship proposed by Cicoira et al. (2020), and the associated bias. The last row gives the mean absolute error (MAE) derived from the two methods.**

| Rock glacier | $T_{area}$ (m) | $T_{slp}$ (m) | $T_{ref}$ (m) |
|---|---|---|---|
| Murtèl-Corvatsch | 29 | 26.2 | 27 |
| Muragl | 24 | 19 | 20 |
| Schafberg | 24 | 20.8 | 25 |
| MAE | 2.3 | 2 | – |

18. Table 1: Where do the values given in the table come from? I recommend to show only three decimals for the area. The delineation is not so precise.

Re: We introduced the way we obtained these values at L.241:

"To derive the input parameters, we first outlined the boundaries of the three rock glaciers from Google Earth images (September of 2018), from which their shapes and areal extents can be extracted using Geographic Information System tools."

We have changed the precision of the parameters. In addition, we assigned a relative uncertainty of 40% to the area parameter for considering the error propagation.

19. L236: Consider to cite also more classical papers which introduced the approach, e.g. Strozzi et al. (2004).

Re: We have added the suggested citation.

20. L261: I would rather call it uncertainty as no validation measurements are available.

Re: Changed.

21. L298: Provide the information about the source of the ice content of the glaciers.

Re: The amount of glacier ice was estimated based on Randolph Glacier Inventory 4.0. However, we have removed the discussion of regional extrapolation from the revised manuscript after considering the reviewer's suggestion.

22. Fig. 5a: Should be Murtèl-Corvatsch (or only Murtèl)

Re: We have updated the figure.

[Figure]

23. Model-sensitivity: I have no time to think through the model sensitivity in detail. As the model seems to be quite incentive to the different input parameters the authors want to provide more details about the possible reasons.

Re: We are not sure if we understand this comment correctly. We think the model has relatively low sensitivity to the varying parameters, which proves it suitable to use the model constraint, i.e., surface velocity, for deriving ice content.

24. L349f: This and similar kind of sentences are from my point of view not needed as this is evident from the headings.

Re: We thank the reviewer for the comment yet we kept these introductory sentences in the hope that the readers can quickly grasp the content and structure of the following section. In the particular example here, these sentences may seem redundant as the current section only consists of two sub-sections. In other long sections (e.g., Sect. 3.5), the short introductions at the beginning may be helpful and worth retaining. Therefore, for the sake of writing consistency, we keep the sentences here as well.

25. L353: The information about the glacier velocity should either be better integrated in the study and compared to the rock glaciers or omitted. I suggest the latter as it distracts from the general topic of this study.

Re: We have taken the suggestion and omitted this information.

26. Fig. 7: Indicate in this figure which velocity was used to calculate the ice content.

Re: We have highlighted the velocity range used in the model:

[Figure]

**Figure 7: Time series of the InSAR-derived downslope velocities of the landforms. The spatial mean velocities and uncertainties during each period are shown (red squares and error bars) as well as the median (blue) and maximum (orange) velocities. The yellow bands highlight the range of the mean velocities which were used as model constraints for estimating ice fractions.**

27. L.378: What is the area of the coherently moving parts and what is the water equivalent (w.e.) of the moving part?

Re: We defined the "coherently moving part" in the methodology section (Sect. 3.5.1–Step 3):

"Then we defined and outlined the "coherently moving part" of each landform by considering the time series of downslope velocity of each pixel acquired during the observational periods. If the InSAR-measured velocity is higher than 5 cm yr$^{-1}$ in more than half of the interferograms, the pixel was included in the coherently moving part of the landform. Otherwise, the pixel is discarded, i.e., not included in the coherently moving part. The area is considered as inactive or in a transitional kinematic status."

We have introduced the latter concept at L. 470:

"…the water volume equivalents of the moving parts of individual landforms, which are calculated based on the ice fractions and the volume of the moving parts…"

28. L381: Provide the information how much water is stored in all the rock glaciers was derived.

Re: We are not sure if we understand the comment correctly. We derived the total amount of water stored by simply adding up the amount of water stored in all the individual rock glaciers.

29. Table 4 and 5: I suggest to combine and also include the area and the w.e. of the coherently moving parts.

Re: In fact, the *area* and *water equivalent* in the tables correspond to the coherently moving parts. We have provided the area of rock glaciers in the combined table:

**Table 4. Summary of the geometric and structural parameters and the inferred ice content of the coherently moving part of rock glaciers in the study area.**

| Rock glacier | Area $(A_{rg})$ (km²) | Area of the coherently moving part $(A_{cmp})$ (km²) | Width (W) (m) | Active layer thickness $(h_{al})$ (m) | Surface slope $(\alpha)$ (°) | Inference ice content (%) | Water volume equivalent of the coherently moving part (million m³) |
|---|---|---|---|---|---|---|---|
| Kala-Patthar | 0.275 | 0.074 | 240 | 0.68 | 9 | 70±8 | 1.4±0.2 |
| Kongma | 0.384 | 0.077 | 300 | 0.83 | 13 | 72±8 | 1.5±0.2 |
| Lingten | 0.228 | 0.094 | 240 | 0.65 | 20 | 74±8 | 5.9±0.6 |
| Nuptse | 0.310 | 0.234 | 400 | 0.30 | 13 | 74±8 | 2.0±0.2 |
| Tobuche | 0.236 | 0.128 | 400 | 1.67 | 16 | 74±8 | 2.7±0.3 |

30. Section 4.5: As written above this is a very rough approximation. Simply extrapolation the values from 5 rock glaciers does not really add to our knowledge considering the uncertainty. If the authors aim to upscale then using their approach and including the velocity information.

Re: We have removed the discussion of regional extrapolation from the revised manuscript.

31. Section 5 Discussion: As written the discussion about the uncertainties is appreciated, but also highlights the large uncertainties and hence sheds many questions on the approach. I recommend the authors also to highlight the advances of the presented approach in relation to the literature and, hence, better justify their presented approach. A relevant paper to consider here or maybe already when presented the method is Hartl et al. (2016)

Re: We have taken the suggestion and rewritten the *Discussion* to present both an analysis of the uncertainties (Sect. 5.1) and the advances of the approach in relation to previous research (Sect. 5.3):

**"5.3 Contribution and prospect of the approach**

For the first time, we build a model framework to infer ice content with remote sensing-based input by taking advantage of the existing observational data. Previous research either relied on costly and labor-intensive in-situ methods, such as borehole drilling and geophysical surveys, to measure the ice content of individual rock glaciers (e.g., Haeberli et al., 1998; Hauck, 2013), or provided categorized estimates for regional scale studies (e.g., Jones et al., 2018 and 2021). The approach we have developed makes it possible to more conveniently and quantitatively assess the ground ice stored in individual or even region-wide rock glaciers.

The proposed approach can be further improved. The likely emergence of more data to be integrated for model calibration and validation, will allow for improving the accuracy of the method. A more accurate 2-D surface velocity can be obtained by using multi-track InSAR data (e.g., Bertone et al., 2022; Zhang et al., 2021), allowing us to apply the model to rock glaciers with a complex velocity field. We expect the improved model can be valuably applied to mountain permafrost regions where rock glaciers are widespread for preliminary water storage evaluation."

32. L403ff: As written above: This is basically a listing of the headings of the subchapters and can therefore be removed.

Re: We have rewritten the *Discussion* while keeping a short introduction at the beginning of the section. We have given reasons for this in our response to Comment #24.

33. L410: Suggest to wright "reader" instead of "user"

Re: Changed.

34. L470ff: Either provide more detailed information about the investigated rock glaciers (the Tien Shan is larger) and an accessible reference or omit this paragraph.

Re: We have omitted the paragraph from the discussion as suggested.

Overall, quite difficult to judge the overall value of the study. One hand the study is highly interesting and important on other hand contains many shortcomings leading to highly uncertain results. My suggestion would be to split the paper into two: One which focusses on model development (in a region with suitable in-situ measurements) and one which applies an improved method to the larger region.

Re: Again, we would like to thank the reviewer for providing many constructive and insightful feedbacks. We take the suggestion from the reviewer to omit the regional extrapolation from the revised manuscript.

This work is motivated to propose a novel approach for assessing the hydrological value of rock glaciers in a more convenient and quantitative way. Essentially, the methodology consists of both model development based on data from well-studied regions and model application to less-studied areas. Therefore, we maintain the manuscript as one piece for the sake of the integrity of the overall research objective and methodological framework.

Additional references not cited in the manuscript:

Fukui, K., Fujii, Y., Ageta, Y., Asahi, K., 2007. Changes in the lower limit of mountain permafrost between 1973 and 2004 in the Khumbu Himal, the Nepal Himalayas. Global Planet. Change 55, 251–256.

Hartl, L., Fischer, A., Klug, C., Nicholson, L., 2016. Can a simple numerical model help to fine-tune the analysis of ground-penetrating radar data? Hochebenkar rock glacier as acase study. Arct. Antarct. Alp. Res. 48, 377–393. https://doi.org/10.1657/AAAR0014-081.

Miles, K.E., Hubbard, B., Miles, E.S., Quincey, D.J., Rowan, A.V., Kirkbride, M., Hornsey, J., 2021. Continuous borehole optical televiewing reveals variable englacial debris concentrations at Khumbu Glacier, Nepal. Communications Earth & Environment 2, 12. https://doi.org/10.1038/s43247-020-00070-x.

Strozzi, T., Kääb, A., Frauenfelder, R., 2004. Detecting and quantifying mountain permafrost creep from in situ inventory, space-borne radar interferometry and airborne digital photogrammetry. Int. J. Remote Sens. 25, 2919–2931.

Wirz, V., Gruber, S., Purves, R.S., Beutel, J., Gärtner-Roer, I., Gubler, S., Vieli, A., 2016. Short-term velocity variations at three rock glaciers and their relationship with meteorological conditions. Earth Surf. Dynam. 4, 103–123. https://doi.org/10.5194/esurf-4-103-2016.